# Online RL in Linearly $q^\pi$-Realizable MDPs Is as Easy as in Linear MDPs If You Learn What to Ignore

**Gellért Weisz**
Google DeepMind, London, UK
University College London, London, UK

**András György**
Google DeepMind, London, UK

**Csaba Szepesvári**
Google DeepMind, Montreal, Canada
University of Alberta, Edmonton, Canada

## Abstract

We consider online reinforcement learning (RL) in episodic Markov decision processes (MDPs) under the linear $q^\pi$-realizability assumption, where it is assumed that the action-values of all policies can be expressed as linear functions of state-action features. This class is known to be more general than linear MDPs, where the transition kernel and the reward function are assumed to be linear functions of the feature vectors. As our first contribution, we show that the difference between the two classes is the presence of states in linearly $q^\pi$-realizable MDPs where for any policy, all the actions have approximately equal values, and skipping over these states by following an arbitrarily fixed policy in those states transforms the problem to a linear MDP. Based on this observation, we derive a novel (computationally inefficient) learning algorithm for linearly $q^\pi$-realizable MDPs that simultaneously learns what states should be skipped over and runs another learning algorithm on the linear MDP hidden in the problem. The method returns an $\varepsilon$-optimal policy after $\text{polylog}(H, d)/\varepsilon^2$ interactions with the MDP, where $H$ is the time horizon and $d$ is the dimension of the feature vectors, giving the first polynomial-sample-complexity online RL algorithm for this setting. The results are proved for the misspecified case, where the sample complexity is shown to degrade gracefully with the misspecification error.

## 1 Introduction

We consider reinforcement learning where an agent interacts in an online fashion with an environment modeled as a Markov decision process: The agent, observing a state, takes an action that results in a random next state and reward, the latter of which is to be maximized over time. To tackle large, possibly infinite state spaces, additional structure needs to be introduced to this problem. One such structure is a "feature-map" that maps state-action pairs to $d$-dimensional vectors (for some positive integer $d$) with the intention that a "good feature-map extracts important information from the state-action pairs so that learning with this extra information becomes tractable. An example is the case of *linear MDPs* [Jin et al., 2020a], where the assumption is that both the transition and reward functions are linearly factorizable and their left factors are given by the feature-map. In contrast, value-based approaches, such as $q^\pi$-*realizability* [Du et al., 2019, Lattimore et al., 2020] aim to model only the action-values with the features. In this work, we focus on the latter, a strictly more general setting than that of linear MDPs [Zanette et al., 2020, Proposition 4].

There are several sample-efficient algorithms discovering near-optimal policies in linear MDPs under various MDP access models and settings (online access: Jin et al. [2020a]; batch setting: Jin et al.

37th Conference on Neural Information Processing Systems (NeurIPS 2023).

| MDP class | Online RL | | Planning with simulator | |
|---|---|---|---|---|
| | poly($\cdot$) sample | poly($\cdot$) compute | poly($\cdot$) sample | poly($\cdot$) compute |
| Linear MDP | Jin et al. [2020a] | | | |
| $q^\pi$-realizable MDP | **This work** | Open problem | Yin et al. [2022] | |

**Table 1:** Comparison of efficiency results for linear MDPs and $q^\pi$-realizable MDPs under online RL and planing with a simulator. This work establishes that $q^\pi$-realizable MDPs are also sample efficiently solvable under online RL. The computational complexity of this problem remains open.

[2021]; reward-free setting: Wagenmaker et al. [2022]). The best known sample-complexity bound for the online access model is achieved by the computationally inefficient algorithm of Zanette et al. [2020], called ELEANOR, which serves as a starting point of our work.

In this work we consider the setting of linearly $q^\pi$-realizable MDPs. As opposed to linear MDPs, before this work, sample efficient solutions were only known for this case when the MDP is accessed through a simulator that implements some form of a state-reset function [Lattimore et al., 2020, Yin et al., 2022, Weisz et al., 2022] (Table 1). In this work we resolve an open problem by Du et al. [2019], and show that having access to a state-reset is not essential in this setting. To this end, we present SKIPPYELEANOR (Algorithm 1) and a corresponding theorem (Theorem 4.1) that shows that SKIPPYELEANOR, which uses online interactions only, is a provably sample-efficient solution to this problem. The rest of this paper is organized as follows. In Section 2 we introduce the basic definitions. In Section 3 we give an insight into the difference between linear $q^\pi$-realizability and linear MDPs, which motivates our approach. In Section 4 we describe our algorithm and the most important technical tools we discovered for its analysis. Notably, in Section 4.2 we establish a rich structure inherent in $q^\pi$-realizable MDPs, which acts as the technical foundation to this work, and may be of independent interest. Finally, Section 5 gives a summary of the proof of our main result (Theorem 4.1), before concluding with some notes on future work in Section 6.

## 2   Preliminaries

For a linear subspace $X$ of $\mathbb{R}^d$, let $\text{Proj}_X$ denote the orthogonal projection matrix onto $X$. Throughout we fix $d \in \mathbb{N}^+$. For $L > 0$, let $\mathcal{B}(L) = \{x \in \mathbb{R}^d : \|x\|_2 \leq L\}$ denote the $d$-dimensional Euclidean ball of radius $L$ centered at the origin, where $\|\cdot\|_2$ denotes the Euclidean norm. Let PD denote the set of positive definite matrices in $\mathbb{R}^{d \times d}$. We write $a \approx_\varepsilon b$ for $a, b, \varepsilon \in \mathbb{R}$ if $|a - b| \leq \varepsilon$. Let $\mathbb{I}\{B\}$ be the indicator function of a boolean-valued (possibly random) $B$ taking value 1 if $B$ is true and 0 if false. Let $\mathcal{M}_1(X)$ denote the set of probability distributions supported on set $X$. The rest of our notation is standard, but described in Appendix A for completeness.

For the setting of episodic finite horizon RL, with horizon $H$, a finite-action Markov decision process (MDP) describes an environment for sequential decision-making. It is defined by a tuple $(\mathcal{S}, [\mathcal{A}], P, \mathcal{R})$ as follows. The state space $\mathcal{S}$ is split across stages: $\mathcal{S} = (\mathcal{S}_t)_{t \in [H]}$ with $\mathcal{S}_1 = \{s_1\}$ for some designated initial state $s_1$. Without loss of generality, we assume the $(\mathcal{S}_t)_{t \in [H]}$ are disjoint sets. We define the function stage : $\mathcal{S} \to [H]$ as stage($s$) = $t$ if $s \in \mathcal{S}_t$. We consider finite action spaces of size $\mathcal{A}$ for some $\mathcal{A} \in \mathbb{N}^+$, and without loss of generality, define the set of actions to be $[\mathcal{A}] := \{1, \ldots, \mathcal{A}\}$. The transition kernel is $P : (\bigcup_{t \in [H-1]} \mathcal{S}_t) \times [\mathcal{A}] \to \mathcal{M}_1(\mathcal{S})$, with the property that transitions happen between successive stages, that is, for any $t \in [H-1]$, state $s_t \in \mathcal{S}_t$, and action $a \in [\mathcal{A}]$, $P(s_t, a) \in \mathcal{M}_1(\mathcal{S}_{t+1})$. The reward kernel is $\mathcal{R} : \mathcal{S} \times [\mathcal{A}] \to \mathcal{M}_1([0, 1])$. An agent interacts sequentially with this environment in an episode lasting $H$ steps by taking some action $a \in [\mathcal{A}]$ in the current state. The environment responds by transitioning to some next-state according to $P$, and giving a reward in $[0, 1]$ according to $\mathcal{R}$.[1]

We describe an agent interacting with the MDP by a *policy* $\pi$, which, to each history of interaction (including states, actions and rewards) assigns a probability distribution over the actions. Policies where this distribution only depend on the last state in the history are called *memoryless*, and these are identified with elements of the set $\Pi = \{\pi : \mathcal{S} \to \mathcal{M}_1([\mathcal{A}])\}$. Using a policy $\pi$, starting at some state $s$ in an MDP induces a probability distribution over histories, which we denote by $\mathcal{P}_{\pi,s}$. For

---

[1]Here, the reward and next-state are independent, given the current state and last action. Independence is nonessential and is assumed only to simplify the presentation.

any $a \in [\mathcal{A}]$, $\mathcal{P}_{\pi,s,a}$ is the distribution over the histories when first action $a$ is used in state $s$, after which policy $\pi$ is followed. $\mathbb{E}_{\bullet}$ is the expectation operator corresponding to a distribution $\mathcal{P}_{\bullet}$ (e.g., $\mathbb{E}_{\pi,s}$ is the expectation with respect to $\mathcal{P}_{\pi,s}$). The state- and action-value functions $v^{\pi}$ and $q^{\pi}$ are defined as the expected total reward within the first episode while $\pi$ is used:

$$v^{\pi}(s) = \mathbb{E}_{\pi,s} \sum_{u=\text{stage}(s)}^{H} R_u \quad \text{for } s \in \mathcal{S} \quad \text{and} \quad q^{\pi}(s,a) = \mathbb{E}_{\pi,s,a} \sum_{u=\text{stage}(s)}^{H} R_u \quad \text{for } s \in \mathcal{S}, \, a \in [\mathcal{A}].$$

Let $\pi^{\star} \in \Pi$ be an optimal policy, satisfying $q^{\pi^{\star}}(s,a) = \sup_{\pi \in \Pi} q^{\pi}(s,a) = \sup_{\pi \in \text{all policies}} q^{\pi}(s,a)$ for all $(s,a) \in \mathcal{S} \times [\mathcal{A}]$. Let $q^{\star}(s,a) = q^{\pi^{\star}}(s,a)$ and $v^{\star}(s) = \sup_{a' \in [\mathcal{A}]} q^{\star}(s,a)$ for all $(s,a)$.

# 3 From linear $q^{\pi}$-realizability to linear MDPs

As described in the introduction, we endow our MDP with a feature map $\varphi : \mathcal{S} \times [\mathcal{A}] \to \mathcal{B}(L_1)$ for some $L_1 > 0$. For reference, we start with a definition of linear MDPs with a parameter norm bound $L_2 > 0$, formalizing that the transition kernel and the expected rewards are approximately linear functions of the features:[2]

**Definition 3.1.** *[$\kappa$-approximately linear MDP] For any $\kappa \leq 1$, an MDP is a $\kappa$-approximately linear MDP if* (i) *there exists $\theta_1, \ldots, \theta_H \in \mathcal{B}(L_2)$ such that for any $h \in [H]$ and $(s,a) \in \mathcal{S}_h \times [\mathcal{A}]$, $\left| \mathbb{E}_{R \sim \mathcal{R}(s,a)} R - \langle \varphi(s,a), \theta_h \rangle \right| \leq \kappa$ and* (ii) *for any $f : \mathcal{S} \to [0,H]$ and $h \in [H-1]$, there exists $\theta'_h \in \mathcal{B}(L_2)$ such that for all $(s,a) \in \mathcal{S}_h \times [\mathcal{A}]$, $\left| \mathbb{E}_{S' \sim P(s,a)} f(S') - \langle \varphi(s,a), \theta'_h \rangle \right| \leq \kappa$.*

A key consequence of the linear MDP assumption is that the *inherent Bellman error*

$$\sup_{\theta_{h+1} \in \mathcal{B}(L_2)} \inf_{\theta_h \in \mathcal{B}(L_2)} \sup_{(s,a) \in \mathcal{S}_h \times [\mathcal{A}]} \left| \mathbb{E}_{R \sim \mathcal{R}(s,a), S' \sim P(s,a)} R(s,a) + \max_{a' \in [\mathcal{A}]} \langle \varphi(S',a'), \theta_{h+1} \rangle - \langle \varphi(s,a), \theta_h \rangle \right|,$$

scales with the misspecification $\kappa$. This property is also referred to as the *closedness to the Bellman operator*, and is a crucial component in the analysis of approximation errors for algorithms tackling linear MDPs.

In this work we consider a weaker linearity assumption where we only assume that the action-value functions are approximately linear:

**Definition 3.2** ($q^{\pi}$-realizability: uniform linear function approximation error of value-functions)**.** *Given an MDP, the uniform value-function approximation error (or misspecification) induced by a feature map $\varphi : \mathcal{S} \times [\mathcal{A}] \to \mathcal{B}(L_1)$, over a set of parameters in $\mathcal{B}(L_2)$ is*

$$\eta = \sup_{\pi \in \Pi} \max_{h \in [H]} \inf_{\theta^{(h)} \in \mathcal{B}(L_2)} \sup_{(s,a) \in \mathcal{S}_h \times [\mathcal{A}]} \left| q^{\pi}(s,a) - \left\langle \varphi(s,a), \theta^{(h)} \right\rangle \right|.$$

*For the MDP and the corresponding feature map, for all $h \in [H]$ fix any $\theta_h : \Pi \to \mathcal{B}(L_2)$ mapping each memoryless policy $\pi \in \Pi$ to its "parameter", such that*

$$q^{\pi}(s,a) \approx_{\eta} \langle \varphi(s,a), \theta_h(\pi) \rangle \qquad \text{for all } \pi \in \Pi, \, s \in \mathcal{S}_h, \text{ and } a \in [\mathcal{A}]. \tag{1}$$

*The set of all parameters $\Theta_h \subseteq \mathcal{B}(L_2)$ for a stage $h \in [H]$ is given by $\Theta_h = \{\theta_h(\pi) : \pi \in \Pi\}$.*

Note that $\theta_h$ satisfying Eq. (1) always exist [Weisz et al., 2022, Appendix C]. We focus on the feasible regime where $\eta$ is polynomially small in the relevant parameters. Specifically, we assume that $\eta$ is bounded according to Eq. (21). The main problem of interest in this work is the following:

**Problem 3.3** (informal)**.** *For any $\varepsilon, \zeta > 0$ and any MDP with corresponding uniform value-function approximation error $\eta$, derive an algorithm that, with probability at least $1 - \zeta$, will find an $\varepsilon$-optimal policy (i.e., a policy $\pi$ such that $v^{\pi}(s_1) \geq v^{\star}(s_1) - \varepsilon$) by interacting with the MDP online for $T$ steps with $T$ bounded by a polynomial function of $(d, H, \varepsilon^{-1}, \log \zeta^{-1}, \log L_1, \log L_2)$. That the interaction with the MDP is online means that it is only possible to observe the features corresponding to the current state, and to take an action and subsequently observe the resulting reward and next state, which then becomes the current state. We consider the fixed horizon episodic setting, that is, the next state is reset to the initial state $s_1$ after every $H$ steps.*

---

[2]Compared to the definition of Jin et al. [2020b], our definition does not require the existence of a vector-valued measure to represent the transition kernel. This is a generalization that is compatible with all existing algorithms for linear MDPs.

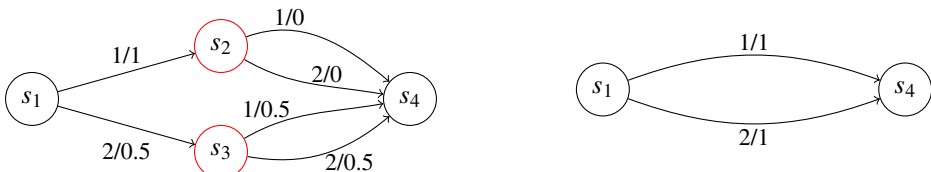

**Figure 1: Left:** MDP with deterministic transitions and rewards (edges are labeled with action/reward). **Right:** The same MDP with the red "low-range" states "skipped" over. $\varphi(s_1, \cdot) = (1), \varphi(s_3, \cdot) = (0.5), \varphi(\cdot, \cdot) = (0)$ otherwise. Both MDPs are $q^\pi$-realizable, but only the right MDP is linear.

Algorithms developed for linear MDPs are not directly applicable to Problem 3.3 when the MDP is only $q^\pi$-realizable: While a linear MDP is also $q^\pi$-realizable, a $q^\pi$-realizable MDP may be neither a linear MDP, nor one with a low inherent Bellman error [Zanette et al., 2020]. As an illustrative example, Fig. 1, left shows an MDP that is $q^\pi$-realizable but not linear. To see this, observe that the features for both actions in $s_1$ are identical, but their transitions and rewards are not. As illustrated in the figure however, if we *skip* over the red states (with identical actions) by taking the first action on them and summing up the rewards received until we reach a black state, we arrive at a linear MDP. This serves as the main intuition behind our work: the red states have no bearing on action-values, so they can be skipped, and the resulting MDP is linear.

More generally, we can define the *range* of any state as the maximum possible difference in action-value that the choice of action in that state can make:

$$\text{range}(s) = \sup_{\theta \in \Theta_{\text{stage}(s)}} \max_{i,j \in [\mathcal{A}]} \langle \varphi(s, i, j), \theta \rangle \text{ for all } h \in [H], s \in \mathcal{S}_h, \tag{2}$$

where $\varphi(s, i, j) = \varphi(s, i) - \varphi(s, j)$ is the notation for feature differences. Clearly, the choice of action in low-range states is not too important, as

$$v^\pi(s) - q^\pi(s, a) \leq \text{range}(s) + 2\eta \qquad \text{for any } \pi \in \Pi \text{ and all } a \in [\mathcal{A}]. \tag{3}$$

Not only are the action choices in low-range states unimportant for the task of finding a near-optimal policy for the MDP, these choices can affect transitions and rewards in a nonlinear way. Interestingly, the existence of low-range states is the reason why $q^\pi$-realizable MDPs are not necessarily linear, as shown by the next result (proved in Appendix C), which follows easily from Lemma 4.7.

**Proposition 3.4.** *Consider an MDP with uniform value-function approximation error $\eta \geq 0$. If there are no states $s \in \mathcal{S}$ with $\text{range}(s) < \alpha$ for some $\alpha > 0$, then the transitions and rewards of the MDP are linear (Definition 3.1) with misspecification scaling with $\eta$, and parameter norms scaling inversely with $\alpha$.*

**Our approach.** The above result immediately offers a strategy to learn under the (linear) $q^\pi$-realizability assumption. Assuming access to an oracle that can determine whether or not $\text{range}(s) < \alpha$ for any state $s$, the MDP could be "converted" to one that has no low-range states but has near-identical state and action-value functions of any policy (compared to the original MDP), by skipping over low-range states (by executing an arbitrary action) until a state with a range at least $\alpha$ is reached. We will call such a multi-state transition a *skippy step* and refer to such a policy as a *skippy policy*. The reward presented for a skippy step is the cumulative reward over the skipped states. When the oracle is correct, the new MDP is a linear MDP, allowing techniques such as ELEANOR to efficiently learn a near-optimal policy. This conversion argument is part of the intuition of our method, but it is not strictly part of the proof, so we defer the details to Appendix C. The only missing piece for solving the general case, Problem 3.3, is learning an oracle that can suggest when to skip over a state, and combining it with the learning algorithm for the linear MDP. This general approach leads to our algorithm, SKIPPYELEANOR, which runs a modified version of ELEANOR with guessed oracles. During the algorithm, we detect when an incorrect oracle leads to suboptimal results, and refine the oracle accordingly. The details of the algorithm are explained in the next section.

## 4 Algorithm

In this section we present our main results following our plan outlined above. We first give Algorithm 1, along with a high-level overview of the algorithm; the details are explained throughout the section. The parameters of the algorithm are presented in Appendix B.

---

**Algorithm 1** SKIPPYELEANOR

---

1: **Input:** accuracy $\varepsilon > 0$, failure probability $\zeta > 0$
2: Initialize $m \leftarrow 0$, $m' \leftarrow 0$, $Q_h = L_2 I$ for $h \in [H]$, $\pi^0 = (s \mapsto 1)$
3: **while** $m' \leq m'_{\max}$ **do**
4:  $\quad$ $m \leftarrow m + 1$, $m' \leftarrow m' + 1$ $\qquad\qquad\qquad$ ▷ $m'$ also counts iterations repeated due to Line 14
5:  $\quad$ Estimate optimistic problem parameters $\hat{G}, \bar{\theta}$ by solving Optimization Problem 4.10
6:  $\quad$ **for** $k \in [H]$ **do**
7:  $\quad\quad$ Let $\pi^{mk}$ be the policy defined by SKIPPYPOLICY$(\hat{G}, \bar{\theta}, k)$
8:  $\quad\quad$ Sample $n$ trajectories by executing $\pi^{mk}$ from $s_1$ for $n$ episodes
9:  $\quad\quad$ Record data $(S_h^{mkj}, A_h^{mkj}, R_h^{mkj})_{h \in [H], j \in [n]}$ and stage-mapping functions $(p^{mkj})_{j \in [n]}$
10: $\quad$ Solve Optimization Problem 4.12 with input $(\hat{G}, \bar{\theta})$, $\qquad\qquad$ ▷ Consistency check
    $\quad$ record its value $x$ (maximum discrepancy), and arguments $v$ (direction) and $i$ (stage).
11: $\quad$ Calculate useful component $w \leftarrow \text{Proj}_{Z(Q,i)} v$ $\qquad\qquad\qquad$ ▷ Definition 4.2
12: $\quad$ **if** $x > $ discrepancy_threshold **then**
13: $\quad\quad$ $Q_i \leftarrow (Q_i^{-2} + Q_i^{-1} w w^\top Q_i^{-1})^{-\frac{1}{2}}$ $\qquad$ ▷ append $Q_i^{-1} w$ to $C_i$ according to Eq. (4)
14: $\quad\quad$ $m \leftarrow m - 1$ $\qquad\qquad\qquad\qquad\qquad\qquad\qquad$ ▷ redo this iteration
15: $\quad\quad$ **continue**
16: $\quad$ **if** average_uncertainty $\leq$ uncertainty_threshold **then**
17: $\quad\quad$ **return** policy $\pi^{mH}$

---

---

**Algorithm 2** SKIPPYPOLICY

---

1: Input: $\hat{G}, \bar{\theta}, k$
2: Initialize $S_1 \leftarrow s_1$, $j \leftarrow 1$, $\pi^0 \leftarrow (s \mapsto 1)$, stage mapping $p$
3: **for** $i = 1$ to $H$ **do**
4: $\quad$ Compute skip probabilities $\tau_i \leftarrow \tau_{\hat{G}}(S_i)$ and non-skip action $a^+ \leftarrow \pi_{\bar{\theta}}^+(S_i)$ from Eq. (8)
5: $\quad$ Sample independently $B_i \sim \text{Bernoulli}(\tau_i)$
6: $\quad$ **if** $B_i = 0$ **then** $A_i \leftarrow 1$ $\qquad\qquad\qquad$ ▷ skip (follow $\pi^0$) with probability $1 - \tau_i$
7: $\quad$ **else**
8: $\quad\quad$ $p(j) \leftarrow i$, $j \leftarrow j + 1$
9: $\quad\quad$ **if** $j \leq k$ **then** $A_i \leftarrow a^+$ (Phase I) **else** $A_i \leftarrow 1$ (Phase II)
10: $\quad$ **if** $i = H$ **then**
11: $\quad\quad$ $p(j') = H + 1$ for $j' = j, \ldots, H$

---

For every stage $h \in [H]$, the algorithm keeps a progressively refined estimate of the geometry of the parameter space $\Theta_h$, by maintaining an ever shrinking ellipsoid enclosing $\Theta_h$. This ellipsoid is parametrized by an 'inverse covariance matrix'-like quantity $Q_h$, determined by $\tilde{O}(d)$ vectors, which guarantees $\max_{\theta_h \in \Theta_h} \|\theta_h\|_{Q_h^{-2}} = \tilde{O}(\sqrt{d})$. Looking at the definition of range in Eq. (2), it is clear that the smaller the ellipsoid becomes, the better estimate we can give for the ranges.

Given some data collected so far and $(Q_h)_{h \in [H]}$, SKIPPYELEANOR computes optimistic estimates of the action-values by calculating an optimistic policy parameter $\bar{\theta}$, as well as a guess $\hat{G}$ to a near-optimal design which is used to estimate the range for the states (due to technical reasons, $\hat{G}$ will guess a near-optimal design for the transformed parameter space $Q_h^{-1}\Theta_h$).

Data is collected by running stochastic versions of skippy policies on the MDP, where the states to be skipped over are determined based on the range estimates; when a state is skipped, an action is selected using a deterministic policy $\pi^0$ that always chooses the first action in every state. To ensure that the estimation problem is smooth in terms of $\hat{G}$, we use a smoothed version of skippy policies, where states are skipped randomly, and the probability of skipping is larger for states with lower ranges, while high-range states are never skipped. Similarly to ELEANOR, we aim to estimate the action-value function of a state-action pair by adding the estimated one-step reward to the estimated value-function of the next state. However, unlike ELEANOR, we would like to do this in the reduced MDP, where the low-range states that are skipped over are removed (and the corresponding transitions are replaced by skippy steps). Since we do not know these states in advance, we run exploratory policies that skip over next states starting from any state: namely,

we run SKIPPYPOLICY($\hat{G}, \bar{\theta}, k$) for all $k \in [H]$ with a maximum number of unskipped states $k$ (Phase I), and once this is skip budget is exhausted, all remaining states are skipped over by rolling out $\pi^0$ (Phase II), which ensures that we collect enough data at every stage of the MDP to be able to estimate the one-skippy-step reward of any skipping mechanism. Compared to ELEANOR, this introduces an additional loop in Line 6 of SKIPPYELEANOR; see Appendix D for additional details. For any execution, SKIPPYPOLICY maintains a stage-mapping function $p$, which, for any stage $h$ of the trajectory in the reduced MDP gives the stage index in the original MDP. In other words, $p(j)$ is the stage of the landing state of the $j^{th}$ skippy step.

Finally, we check if the data collected is consistent with our estimates $\hat{G}$ and $\bar{\theta}$, by calculating the maximal discrepancy of the estimates of the action-value difference at the last non-skipped state of $\pi^{mk} = $ SKIPPYPOLICY($\hat{G}, \bar{\theta}, k$) and that of the fixed skipping policy $\pi^0$ in different directions in the parameter space. If the discrepancy is too large for any $k$, we add the discrepancy-maximizing direction to $Q$ and throw away the data collected in this (i.e., the $m^{th}$) iteration; this is achieved by reducing the iteration counter $m$ by 1. On the other hand, if the discrepancy is small enough, we can guarantee that the gap between the value of $\pi^{mH}$ and $v^\star(s_1)$ scales with how much new information we collected, thus the algorithm can terminate returning this policy if this term is sufficiently small (which it eventually has to be).

The following theorem shows that with high probability, SKIPPYELEANOR finds a near-optimal policy after polynomially many interactions with the MDP. Rhe proof sketch is provided in Section 5, while our method and proof strategy is explained from the perspective of ELEANOR in Appendix D.

**Theorem 4.1.** *With probability at least $1 - \zeta$, SKIPPYELEANOR interacts with the MDP for at most $\tilde{O}\left(H^{11}d^7/\varepsilon^2\right)$ many steps, before returning a policy $\pi$ that satisfies $v^\star(s_1) \leq v^\pi(s_1) + \varepsilon$.*

## 4.1 Preconditioning: the enclosing ellipsoid

In this section we give the technical details about the effects of using the matrix $Q_h$ describing an enclosing ellipsoid for $\Theta_h$ (see Lemma 4.3) as preconditioning the features.

**Definition 4.2** (Valid preconditioning). *$Q = (Q_h)_{h \in [H]}$ is a valid preconditioning matrix sequence if for all $h \in [H]$*

$$Q_h = \left(L_2^{-2}I + \sum_{v \in C_h} vv^\top\right)^{-1/2} \tag{4}$$

*for some sequence $C_h = (v_1, \ldots, v_n)$ of vectors in $\mathbb{R}^d$ such that for all $1 \leq i \leq n$,*

$$\sup_{\theta \in \Theta_h} |\langle \theta, v_i \rangle| \leq 1 \quad and \quad \left\|\left(L_2^{-2}I + \sum_{j=1}^{i-1} v_j v_j^\top\right)^{-\frac{1}{2}} v_i\right\|_2^2 \geq \tfrac{1}{2} \quad and \quad \|v\|_2 \leq L_3, \tag{5}$$

*where $L_3$ is some fixed polynomial of the problem parameters $(d, H, \varepsilon^{-1}, \log \zeta^{-1}, \log L_1, \log L_2)$. (see Eq. (35) for its precise value).*

*For a valid preconditioning $Q$ and some $h \in [H]$, let $Z(Q, h)$ be the linear subspace spanned by those eigenvectors of $Q$ whose corresponding eigenvalues are at least $L_3^{-2}$. Let $\mathrm{Proj}_{Z(Q,h)}$ be the orthogonal projection matrix onto this subspace.*

Sometimes it will be convenient to *precondition* the features and parameters so that the enclosing ellipsoid is transformed to a ball of controlled radius (as Lemma 4.3 will show). To this end, introduce for all $h \in [H]$ and $(s, a, b) \in \mathcal{S}_h \times [\mathcal{A}] \times [\mathcal{A}]$ the following:[3]

$$\varphi_Q(s, a) = Q_h\varphi(s, a), \qquad \varphi_Q(s, a, b) = Q_h\varphi(s, a, b)$$

$$\theta_h^Q(\pi) = Q_h^{-1}\theta_h(\pi), \qquad \Theta_h^Q = \left\{\theta_h^Q(\pi) : \pi \in \Pi\right\} = \left\{Q_h^{-1}\theta : \theta \in \Theta_h\right\} \tag{6}$$

$$\hat{q}^\pi(s, a) = \langle \varphi(s, a), \theta_h(\pi) \rangle = \left\langle \varphi_Q(s, a), \theta_h^Q(\pi) \right\rangle \text{ for all } \pi \in \Pi.$$

The next lemma (proved in Appendix F) shows that for all $h \in [H]$, $Q_h$ defines an enclosing ellipsoid for $\Theta_h$; that is, $\Theta_h \subset \{\theta : \|\theta\|_{Q_h^{-2}} \leq \sqrt{d_1 + 1}\}$.

---

[3] Note that $Q_h, h \in [H]$ is invertible by construction.

**Lemma 4.3.** *Let $d_1 = 4d \log(1+16L_3^4 L_2^4) = \tilde{O}(d)$. Then, for any valid preconditioning $Q$ and $h \in [H]$,*

$$\sup_{\theta \in \Theta_h} \|\theta\|_{Q_h^{-2}} = \sup_{\theta \in \Theta_h^Q} \|\theta\|_2 \le \sqrt{d_1 + 1}.$$

Clearly, every time a new vector is added to $C_h$, the enclosing ellipsoid $\{\theta : \|\theta\|_{Q_h^{-2}} \le \sqrt{d_1 + 1}\}$ shrinks (as a positive semidefinite matrix is added to $Q_h^{-2}$). The following lemma (also proved in Appendix F) uses an elliptical potential argument to bound the number of times this can happen.

**Lemma 4.4.** *For any valid preconditioning $Q$, for all $h \in [H]$, the length of sequence $C_h$ corresponding to $Q_h$ according to Definition 4.2 is at most $d_1$.*

**Near-optimal design for $\Theta_h^Q$.** As $Q_h$ only provides an enclosing ellipsoid for $\Theta_h$, we introduce an (unknown) ellipsoid that aligns better with $\Theta_h^Q$. For all $h \in [H]$, fix a set $G_h^Q$ of policies of size $d_0 := 4d \log \log(d) + 16$, together with a probability distribution $\rho_h^Q$ on $G_h^Q$, such that $(G_h^Q, \rho_h^Q)$ is a near-optimal design for $\Theta_h^Q$ (i.e., satisfying Definition F.1). The existence of such a near-optimal design follows from [Todd, 2016, Part (ii) of Lemma 3.9].

We apply $G_h^Q$ to define a cruder version of range that depends only on a small set of policies, and can therefore be succinctly parametrized to inform SKIPPYPOLICY:

$$\text{range}_Q(s) = \max_{\pi \in G_h^Q} \max_{i,j \in [\mathcal{A}]} \langle \varphi(s, i, j), \theta_h(\pi) \rangle \qquad \text{for all } h \in [H], s \in \mathcal{S}_h. \tag{7}$$

$\text{range}_Q$ is easy to estimate, and can be used to bound the range function (proved in Appendix F):

**Proposition 4.5.** *For all $s \in \mathcal{S}$ and $Q \in PD^H$, $\text{range}(s) \le \sqrt{2d}\, \text{range}_Q(s)$.*

## 4.2 Linearly realizable functions

$q^\pi$-realizability (Definition 3.2) implies the linearity of many more functions than the action-value functions. In this section we characterize an interesting set of such functions, whose (approximate) linearity plays a crucial role in our algorithm and analysis, as their parameters can be conveniently estimated by least squares using the features. We rely on functions $f : \mathcal{S}_h \to \mathbb{R}$ (for some $h \in [H]$) being small for all states, relative to the states' $\text{range}_Q$-value:

**Definition 4.6.** *For any $h \in [H]$, $f : \mathcal{S}_h \to \mathbb{R}$ is $\alpha$-admissible for some $\alpha > 0$ if for all $s \in \mathcal{S}_h$, $|f(s)| \le \text{range}_Q(s)/\alpha$.*

The key observation is that expected (admissible) $f$ values are linearly realizable.

**Lemma 4.7** (Admissible-realizability)**.** *If $f : \mathcal{S}_h \to \mathbb{R}$ is $\alpha$-admissible then it is realizable, that is, for all $t \in [h-1]$ and $\pi \in \Pi$, there exists some $\tilde{\theta} \in \mathbb{R}^d$ with $\|\tilde{\theta}\|_2 \le 4d_0 L_2/\alpha$ such that for all $(s, a) \in \mathcal{S}_t \times [\mathcal{A}]$,*

$$\mathbb{E}_{\pi, s, a} f(S_h) \approx_{\eta_0} \langle \varphi(s, a), \tilde{\theta} \rangle \qquad \text{where } \eta_0 = 5d_0 \eta/\alpha.$$

The proof relies on constructing a set of policies that at states $s \in \mathcal{S}_h$ take a higher value action as opposed to a lower one with a certain probability, configured such that the expected action-value difference of some pairs within the set of policies is (approximately) proportional to $f(s)$. Thus, a linear combination of the action-values of policies in this set are also (approximately) proportional to $f(s)$. The statement of the lemma then follows from setting $\tilde{\theta}$ to the corresponding linear combination of the policies' parameters. The full proof is presented in Appendix G.

Next, we define matrix-valued functions with a special admissibility guarantee even when the underlying scalar-valued function does not satisfy any non-trivial admissibility criterion. We introduce a *guess* on the near-optimal design parameters that define $\text{range}_Q$ (Eq. (7)) for some valid preconditioning $Q$:

**Definition 4.8.** *For $h \in [2 : H]$, fix some arbitrary order of the policies in the set $G_h^Q$ (recall that this set is the support of the near-optimal design for $\Theta_h^Q$). Let the parameter of the $i^{th}$ policy in $G_h^Q$ be $\vartheta_h^i$ for $i \in [d_0]$. Call a "guess" of these parameters $\hat{G} = (\hat{G}_h)_{h \in [2:H]} = (\hat{\vartheta}_h^i)_{h \in [2:H], i \in [d_0]}$ "valid", if for all $h \in [2 : H], i \in [d_0]$, $\hat{\vartheta}_h^i \in \mathcal{B}(\sqrt{d_1 + 1})$. Let the set of valid guesses be $\mathbf{G}$.[4] By Lemma 4.3, $(\vartheta_h^i)_{h \in [2:H], i \in [d_0]} \in \mathbf{G}$, that is, it is a valid guess, and we call this the "correct" guess.*

---

[4] Note that while $G_h^Q$ contains policies, $\mathbf{G}$ and its elements (commonly denoted by $\hat{G}$) contain policy parameter vectors.

From a guess $\hat{G} = (\hat{\vartheta}_h^i)_{h \in [2:H], i \in [d_0]}$ we can calculate corresponding guesses of the range$_Q$-values:

$$\text{range}_Q^{\hat{G}}(s) = \max_{k \in [d_0]} \max_{i,j \in [\mathcal{A}]} \left\langle \varphi_Q(s,i,j), \hat{\vartheta}_{\text{stage}(s)}^k \right\rangle \qquad \text{for all } h \in [2:H], s \in \mathcal{S}_h .$$

Note that for any $h \in [2:H]$ and $s \in \mathcal{S}_h$, $\text{range}_Q^{\hat{G}}(s) = \text{range}_Q(s)$ if $\hat{G}$ is the correct guess for stage $h$.

Let $\bar{\varphi}_Q(s)$ be the unit vector in the direction of the largest feature difference between actions in $s$ and the zero vector if all feature vectors are the same (see Eq. (27) for a formal definition). Then, for any $\hat{G} \in \mathbf{G}$, $h \in [2:H]$, and $f : \mathcal{S}_h \to [-H, H]$, let

$$\mathbf{f}(s) = \bar{\varphi}_Q(s)\bar{\varphi}_Q(s)^\top \min \left\{ 1, \text{range}_Q^{\hat{G}}(s)\frac{\sqrt{2d}H}{\varepsilon} \right\} f(s) \quad \text{for } s \in \mathcal{S}_h .$$

For such $\mathbf{f} : \mathcal{S}_h \to \mathbb{R}^{d \times d}$, we adopt the notation $a^\top \mathbf{f} b$ for any $a, b \in \mathbb{R}^d$ to denote the function $s \in \mathcal{S}_h \mapsto a^\top \mathbf{f}(s)b$, and similarly, $\text{Tr}(\mathbf{f})$ to denote the function $s \in \mathcal{S}_h \mapsto \text{Tr}(\mathbf{f}(s))$.

Let $\text{Proj}_{\|(Q,h)}$ be the projection matrix onto the linear subspace spanned by those eigenvectors of the design matrix $V(G_h^Q, \rho_h^Q)$ (defined in Eq. (25)) whose corresponding eigenvalues are at least $\gamma$ (for some $\gamma > 0$ specified in Appendix B). Intuitively, this is the subspace where $\Theta_h^Q$ has a sufficiently large width. Let $\text{Proj}_{\perp(Q,h)}$ be the projection to the orthogonal complement subspace. For any $v \in \mathbb{R}^d$, we write $v_{\|(Q,h)}$ and $v_{\perp(Q,h)}$ for $\text{Proj}_{\|(Q,h)} v$ and $\text{Proj}_{\perp(Q,h)} v$, respectively.

We are now ready to state our special admissibility guarantee, which is proved in Appendix G. Let $\alpha = \tilde{O}(\varepsilon/(d^{1.5}H^2))$ be as in Eq. (16).

**Lemma 4.9.** *For any $h \in [2:H]$, $\hat{G} \in \mathbf{G}$, any function $\mathbf{f}$ constructed as above from some $f : \mathcal{S}_h \to [-H, H]$, and any $v, w \in \mathcal{B}(1)$, $v_{\|(Q,h)}^\top \mathbf{f} w$ is $\alpha$-admissible. Furthermore, if $\hat{G} = (\vartheta_h^i)_{h \in [2:H], i \in [d_0]}$ (the correct guess), $\text{Tr}(\mathbf{f})$ is also $\alpha$-admissible.*

## 4.3 Least-squares targets and Optimization Problem 4.10

Recall that SKIPPYELEANOR estimates action-values of states by first adding the estimated one-step reward and the estimated value-function of the next state in the reduced MDP (where low-range states are skipped). Due to the linearity of $q^\pi$-values, these can be used as target variables of a least-squares estimator to estimate the policy parameters. This estimator is only guaranteed to be accurate if the right (low-range) states are skipped; otherwise, we will argue in Section 4.4 that a discrepancy is detected and it is handled by changing the preconditioning $Q$. Finally, to ensure optimism, we select parameter estimates that lead to the largest estimated policy values. The whole estimation process leads to Optimization Problem 4.10, which we define in this section along with the functions that it uses as least-square targets. Each estimation is for a particular stage $h$ and may use the estimates $\bar{\theta}_i$ of Optimization Problem 4.10 for stages $i > h$. In this subsection, we consider the $m^{\text{th}}$ iteration of the optimization called by SKIPPYELEANOR, and consider $Q$ fixed. As a shorthand, we introduce the following notation for $l \in [m], j \in [n], k \in [H]$:

$\text{p}(lkj) = p^{lkj}(k)$ as recorded in Line 9 of Algorithm 1, and

$$S_{\text{p}(k)}^{lkj} = S_{p^{lkj}(k)}^{lkj}, \quad A_{\text{p}(k)}^{lkj} = A_{p^{lkj}(k)}^{lkj}, \quad R_{\text{p}(k)}^{lkj} = R_{p^{lkj}(k)}^{lkj}, \quad \varphi_t^{lkj} = \varphi(S_t^{lkj}, A_t^{lkj}), \quad \varphi_{\text{p}(k)}^{lkj} = \varphi(S_{\text{p}(k)}^{lkj}, A_{\text{p}(k)}^{lkj}) .$$

We collect the set of $(l, k, j)$ tuples for which the $k^{\text{th}}$ skippy step lands at stage $t$, for $t \in [H]$, as

$$\mathbf{I}^m(t) = \{(l, k, j) : l \in [m-1], j \in [n], k \in [H], \text{p}(lkj) = t\}$$

Note in particular that here $l \in [m-1]$, so $\mathbf{I}^m$ only considers data collected prior to iteration $m$.

To estimate the parameters $\hat{G}$ and $\bar{\theta}$, we consider (simulated) trajectories of SKIPPYPOLICY starting from stage $t$. For simplicity, we suppress the dependence of quantities on $\hat{G}$ and $\bar{\theta}$, which will be brought back later. The skipping probability $1 - \tau$, the policy $\pi^+$ (to be also used in SKIPPYPOLICY), and corresponding clipped action-value estimates are defined as

$$\tau(s) = \min \left\{ 1, \text{range}_Q^{\hat{G}}(s)\frac{\sqrt{2d}H}{\varepsilon} \right\} \qquad \text{if stage}(s) > 1, \text{ and } \tau(s_1) = 1;$$

$$\pi^+(s_i) = \arg\max_{a \in [\mathcal{A}]} \left\langle \varphi(s_i, a), \bar{\theta}_i \right\rangle, \qquad C(s_i) = \text{clip}_{[0,H]} \left\langle \varphi(s_i, \pi^+(s_i)), \bar{\theta}_i \right\rangle . \tag{8}$$

Let $s_i\rightarrow = (s_i, a_i, r_i, \ldots, s_H, a_H, r_H) \in \mathcal{S}_i \times [\mathcal{A}] \times [0,1] \times \cdots \times [0,1]$ be any ending of a trajectory. For $s_{t+1}\rightarrow$, let $I$ be the (random) index of the first state that is *not* skipped by SKIPPYPOLICY with the above $\tau$ (or $H+1$, if such an index does not exist). Then the estimated policy value of SKIPPYPOLICY from stage $t$ is

$$\mathbb{E}_I \left[ \sum_{u=t}^{I-1} r_u + \mathbb{1}\{I < H+1\} C(s_I) \right],$$

the sum of rewards along the skipped states plus the policy-value estimate from stage $I$. It follows from Corollary 4.11 below (proved based on Lemma 4.9) that if $\mathrm{range}_Q^{\hat{G}}$ is an accurate estimate of $\mathrm{range}_Q$, then this quantity decomposes into terms that are linearly expressible using the features. Therefore, we use such quantities as least-square targets. Indeed, writing out the expectation, we can re-express the estimated policy value as the sum of all rewards $\sum_{u=t}^{H} r_u$ plus a correction term $E^\rightarrow(s_{t+1}\rightarrow)$ defined as

$$E^\rightarrow(s_i\rightarrow) = \sum_{j=i}^{H} D(s_j\rightarrow)\tau(s_j) \prod_{j'=i}^{j-1} (1 - \tau(s_{j'})) \quad \text{where} \quad D(s_i\rightarrow) = C(s_i) - \sum_{u=i}^{H} r_u \quad \text{for } i > 1. \quad (9)$$

The next optimization problem aims to find optimistic parameters yielding the largest estimated action-value function for $s_1$, where $\bar{\theta}$ is in the confidence ellipsoid of the least-squares estimates $\hat{\theta}$.

**Optimization Problem 4.10** (for iteration $m$). *For input state $s$, with $\beta$ defined in Appendix B (emphasizing the dependence of functions defined above on $\hat{G}$ and $\bar{\theta}$ by adding them as subscripts):*

$$\underset{\hat{G} \in \mathbf{G}, \bar{\theta}_t \in \mathcal{B}(4d_0 H L_2/\alpha) \text{ for } t \in [H]}{\arg\max} C_{\hat{G}\bar{\theta}}(s_1) \qquad \text{subject to, for all } t \in [H]$$

$$X_{mt} = \lambda I + \sum_{lkj \in \mathbf{I}^m(t)} \varphi_t^{lkj} \varphi_t^{lkj\top}, \ \left\| \bar{\theta}_t - \hat{\theta}_t \right\|_{X_{mt}} \leq \beta H, \ \hat{\theta}_t = X_{mt}^{-1} \sum_{lkj \in \mathbf{I}^m(t)} \varphi_t^{lkj} \underbrace{\left( E_{\hat{G}\bar{\theta}}^\rightarrow(S_{t+1}^{lkj}, \ldots, R_H^{lkj}) + \sum_{u=t}^{H} R_u^{lkj} \right)}_{\text{least-squares target}}.$$

Since our realizability results in Section 4.2 only apply to functions defined at a given stage (as only memoryless policies are $q^\pi$-realizable), to be able to show that the least-squares targets are linearly realizable, we first decompose $E^\rightarrow(s_i\rightarrow)$ ($i \in [2:H]$) to directly express the effect of each stage in the trajectory (backwards): defining $E(s_i\rightarrow) = E^\rightarrow(s_i\rightarrow) - E^\rightarrow(s_{i+1}\rightarrow)$ (for convenience, we use the notation $E^\rightarrow(s_{H+1}\rightarrow) = 0$), we easily obtain

$$E^\rightarrow(s_i\rightarrow) = \sum_{j=i}^{H} E(s_j\rightarrow) \qquad \text{and} \qquad E(s_i\rightarrow) = \tau(s_i)\big(D(s_i\rightarrow) - E^\rightarrow(s_{i+1}\rightarrow)\big). \quad (10)$$

Next we define matrix-valued functions, whose trace equals $E(s_i\rightarrow)$, that have the same form as $\mathbf{f}$ in Section 4.2, for which Lemma 4.9 applies. This is crucial in establishing optimism of Optimization Problem 4.10, as well as learning from instances where we detect that $E^\rightarrow$ is not realizable in Optimization Problem 4.12. To this end, let

$$F(s_i\rightarrow) = \bar{\varphi}_Q(s_i)\bar{\varphi}_Q(s_i)^\top E(s_i\rightarrow) \qquad \text{and} \qquad \bar{F}(s_i) = \mathbb{E}_{\pi^0, s_i}[F(s_i, A_i, \ldots, R_H)] \quad \text{for } s_i \in \mathcal{S}_i.$$

Let $\bar{\Theta} = (\mathcal{B}(4d_0 HL_2/\alpha))^H$ denote the base set for the variables $\bar{\theta}_t$ in Optimization Problem 4.10. As $\bar{F}$ is of the same form as $\mathbf{f}$, we can apply Lemma 4.9 and then Lemma 4.7 to arrive at the following:

**Corollary 4.11.** *For any $\hat{G} \in \mathbf{G}$, $\bar{\theta} \in \bar{\Theta}$, $v, w \in \mathcal{B}(1)$, and for any $t \in [H-1]$, $i \in [t+1:H]$, there exists some $\tilde{\theta}_{ti} \in \mathbb{R}^d$ with $\left\| \tilde{\theta}_{ti} \right\|_2 \leq 4d_0 L_2/\alpha = 1/\sqrt{\lambda}$ such that for all $(s,a) \in \mathcal{S}_t \times [\mathcal{A}]$, where $\eta_0$ is defined in Lemma 4.7.*

$$\mathbb{E}_{\pi^0, s, a}\left[ v_{\|(Q,i)}^\top \bar{F}_{\hat{G}\bar{\theta}}(S_i)w \right] \approx_{\eta_0} \left\langle \varphi(s,a), \tilde{\theta}_{ti} \right\rangle. \quad (11)$$

*Furthermore, if $\hat{G}$ is the correct guess, there exists some $\tilde{\theta}'_{ti} \in \mathbb{R}^d$ with $\left\| \tilde{\theta}'_{ti} \right\|_2 \leq 4d_0 L_2/\alpha$ such that for all $(s,a) \in \mathcal{S}_t \times [\mathcal{A}]$, $\mathbb{E}_{\pi^0, s, a}[E_{\hat{G}\bar{\theta}}(S_i, \ldots, R_H))] = \mathbb{E}_{\pi^0, s, a}[\mathrm{Tr}(\bar{F}_{\hat{G}\bar{\theta}}(S_i))] \approx_{\eta_0} \left\langle \varphi(s,a), \tilde{\theta}'_{ti} \right\rangle.$*

## 4.4 Checking consistency

Considering the $m^{\text{th}}$ iteration of SKIPPYELEANOR, we want to verify if the estimated targets of Optimization Problem 4.10 are accurate (and learn if a discrepancy is detected), by using Corollary 4.11 on the targets' decomposition into $F$-functions. We filter the data collected in the $m^{\text{th}}$ iteration with the indicator $\tilde{c}_{ki}^j = \mathbb{1}\{\mathrm{p}(mkj) < i\}$ for $j \in [n]$, $k \in [H+1]$, $i \in [H+1]$, and further constrain

this by another indicator $c_{ki}^j$ (defined in Appendix B) that requires the data-point's least-squares uncertainty term to be sufficiently low, and the prediction non-negative (the contribution of the rest of the data will be analyzed separately). Next, we define the least-squares solution for estimating the matrix-valued $F$, as well as the empirical average prediction and realization of $F$ on the data collected in the $m^{\text{th}}$ round. For any $i \in [2 : H]$, $k \in [i-1]$ (recall that $\otimes$ denotes the tensor product):

$$\hat{\theta}_{\hat{G}\bar{\theta}}^{ti} = X_{mt}^{-1} \sum_{lkj \in \mathbf{I}^m(t)} \varphi_t^{lkj} \otimes F_{\hat{G}\bar{\theta}}(S_i^{lkj}, \ldots, R_H^{lkj}) \qquad \text{for } t \in [i-1]$$

$$y_{\hat{G}\bar{\theta}}^{ki} = \frac{1}{n} \sum_{j \in [n]} c_{ki}^j \varphi_{\mathrm{p}(k)}^{mkj\top} \hat{\theta}_{\hat{G}\bar{\theta}}^{\mathrm{p}(mkj),i} \qquad\qquad \hat{F}_{\hat{G}\bar{\theta}}^{ki} = \frac{1}{n} \sum_{j \in [n]} c_{ki}^j F_{\hat{G}\bar{\theta}}(S_i^{mkj}, \ldots, R_H^{mkj}) \tag{12}$$

In Appendix E.1, it is established via the usual least-squares analysis techniques and covering arguments, that with high probability the norm of the product of the matrix $y_{\hat{G}\bar{\theta}}^{ki} - \hat{F}_{\hat{G}\bar{\theta}}^{ki}$ and the projection matrix $\mathrm{Proj}_{\|(Q,i)}$ is small (Lemmas E.2 and E.3). The next optimization problem tests if this is true in arbitrary directions:

**Optimization Problem 4.12** (Consistency check). *Input: $(\hat{G}, \bar{\theta})$*

$$\underset{k \in [H-1],\, i \in [k+1:H],\, v \in \mathbb{R}^d : \|v\|_2 = 1}{\arg\max} \quad v^\top \left( y_{\hat{G}\bar{\theta}}^{ki} - \hat{F}_{\hat{G}\bar{\theta}}^{ki} \right) v$$

Lemma E.1 shows that the projection $w = \mathrm{Proj}_{Z(Q,i)} v$ is close to $v$, where $v$ is the outcome of Optimization Problem 4.12. Also, Lemmas E.1–E.3 imply that if the consistency check fails (i.e., Line 13 is executed because the value of Optimization Problem 4.12 is large), then $w$ aligns well with the subspace $\mathrm{Proj}_{\perp(Q,i)}$ projects to, and therefore $Q$ stays a valid preconditioning after appending $w$ to the list of values $Q$ is calculated from (Lemma E.4). Thus, $Q$ is always a valid preconditioning.

## 5 Proof overview

The proof of Theorem 4.1 is presented in Appendix E. It is composed of the following main steps: First, we bound the number of times the consistency check can fail (i.e., Line 13 is executed) by Lemma 4.4. Combining this with Lemma E.5, an elliptical potential argument bounding the number of times the average uncertainty can be large (these are the only two ways that the main iteration can continue) implies a sample-complexity result for SKIPPYELEANOR (Corollary E.6). Having limited the number of times the consistency check can fail, we derive guarantees regarding the performance of the policy returned by the algorithm: Via an induction argument (Lemma E.8) we show Corollary E.9, which shows that with high probability the difference between the optimization value of Optimization Problem 4.10, $C_{\hat{G},\bar{\theta}}(s_1)$ and $v^{\pi^{mH}}$ scales with the average uncertainty term $\sum_{i=1}^H \bar{\sigma}_k^m$. Thus, they are close when SKIPPYELEANOR returns in Line 17. This is complemented with the *optimism* property proved in Lemma E.10, stating that the optimization value $C_{\hat{G},\bar{\theta}}(s_1)$ is close to $v^\star(s_1)$. Combined, this proves Theorem 4.1.

## 6 Future work

Since we are not aware of a computationally efficient implementation of SKIPPYELEANOR, it remains an open question whether the problem of learning near-optimal policies from online interactions with a $q^\pi$-realizable MDP (Problem 3.3) is possible if the computational resources as well as the sample complexity are bounded by a polynomial in the relevant parameters. One approach is to replace ELEANOR with LSVI-UCB as the underlying algorithm, as the latter, despite having worse sample complexity, has a computationally efficient implementation [Jin et al., 2020b]. The challenge is to compute the optimal solution for the parameter $\hat{G}$ in Optimization Problem 4.10. This parameter interacts with the least-squares targets in a highly nonlinear way. We have been unable to derive a computationally efficient approximation that has an additive instead of a multiplicative approximation error (additive errors increase linearly in $H$, while multiplicative errors increase exponentially). Alternatively, it may be possible to show a computational hardness result for Problem 3.3 by e.g., reducing it to the satisfiability problem. These are left for future work. Our work on the realizability of auxiliary functions (Section 4.2) may be of independent interest for designing provably efficient algorithms for related problem settings, e.g., the setting of $q^\pi$-realizability in batch RL, where the data collection is not controlled.

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

# A    Notation

As usual, we use $\mathbb{R}$, $\mathbb{N}$, and $\mathbb{N}^+$ to denote the set of reals, non-negative and positive integers, respectively. For $i \in \mathbb{N}^+$, let $[i] = \{1, \ldots, i\}$; for another positive integer $j$, let $[i : j] = \{i, \ldots, j\}$ if $i \le j$, and $[i : j] = \{\}$ otherwise. For $a, b, x \in \mathbb{R}$, let $\text{clip}_{[a,b]}(x) = \min\{\max\{x, a\}, b\}$ and let $\lceil x \rceil$ denote the smallest integer i such that $i \ge x$. Let $\mathbf{0}$ be the all-0 vector in $\mathbb{R}^d$ and $I$ the $d$-dimensional identity matrix. For a (square) matrix $V$, let $V^\dagger$ denote its Moore-Penrose inverse, and $\text{Tr}(V)$ denote its trace. Let PD (and PSD) denote the set of positive definite (and positive semi-definite, respectively) matrices in $\mathbb{R}^{d \times d}$. For some $A \in$ PSD let $A^{\frac{1}{2}}$ denote the unique matrix $B \in$ PSD such that $A = BB$. For $V \in$ PD and $x \in \mathbb{R}^d$, let $\|x\|_V^2 = x^\top G x$. For matrices $A$ and $B$, we say that $A \ge B$ (or $A \le B$) if $B - A$ (or $A - B$, respectively) is positive semidefinite. $\text{Ker}(A)$ and $\text{Im}(A)$ are the kernel (or null space), and image, respectively, of matrix $A$. For compatible vectors $x, y$, let $\langle x, y \rangle$ be their inner product: $\langle x, y \rangle = x^\top y$. We write $y \otimes A$ for the tensor product between $y$ and matrix $A$, and then $\langle x, y \otimes A \rangle = \langle x, y \rangle A$. Where $Q$ and $h$ are obvious from the context, we write $v_\|$ and $v_\perp$ for $v_{\|(Q,h)}$ and $v_{\perp(Q,h)}$, respectively. Throughout the paper, we omit commas between quantities in subscripts or superscript for clarity of presentation, for example, by writing $A_{bc}$ for $A_{b,c}$.

For the big-Oh notation $O$, we introduce its counterpart $\tilde{O}$ that hides logarithmic factors of the problem parameters $(d, H, \varepsilon^{-1}, \zeta^{-1}, L_1, L_2)$.

# B    Parameters of Algorithm 1

$$n = \tilde{O}\left(d^5 H^6 / \varepsilon^2\right) \qquad\qquad \text{(for precise value see Eq. (42))}$$

$$\omega = 7(d_1 + 1) + 7/3 = \tilde{O}(d) \tag{13}$$

$$\gamma^{-1} = 8d = \tilde{O}(d) \tag{14}$$

$$\beta = \tilde{O}(H^{1.5}d) \qquad\qquad \text{(for precise value see Eq. (36))} \tag{15}$$

$$\alpha^{-1} = \frac{\sqrt{2d}\sqrt{d_1 + 1}H^2}{\sqrt{\gamma}\varepsilon} = \tilde{O}(d^{1.5}H^2/\varepsilon) \tag{16}$$

$$\lambda^{-1} = (4d_0 L_2/\alpha)^2 \tag{17}$$

$$m_{\max} = \beta^2 \log\left(1 + \frac{HmnL_1^2}{d\lambda}\right) + 1 = \tilde{O}\left(H^3 d^2\right)$$

$$m'_{\max} = m_{\max} + Hd_1 = \tilde{O}(H^3 d^2)$$

$$\bar{\sigma}_k^m = \frac{1}{n} \sum_{j \in [n]} \tilde{c}_{k,H+1}^j \min\left\{2(\beta\omega dH)^{-1}, \left\|\varphi_{\text{p}(k)}^{mkj}\right\|_{X_{m,\text{p}(mkj)}^{-1}}\right\} \qquad \text{for } k \in [H] \tag{18}$$

$$\tag{19}$$

$$c_{ki}^j = \mathbb{1}\left\{\text{p}(mkj) < i \text{ and } \left\|\varphi_{\text{p}(k)}^{mkj}\right\|_{X_{m,\text{p}(mkj)}^{-1}} < 2(\beta\omega dH)^{-1} \text{ and } \left\langle\varphi_{\text{p}(k)}^{mkj}, \bar{\theta}_{\text{p}(mkj)}\right\rangle \ge 0\right\} \tag{20}$$

$$\text{average\_uncertainty} = \sum_{k=1}^{H} \bar{\sigma}_k^m$$

$$\text{uncertainty\_threshold} = \varepsilon/(dH^2\beta\omega)$$

$$\text{discrepancy\_threshold} = \bar{\sigma}_k^m \beta\omega + 3\frac{\varepsilon}{dH^2}$$

Assumption on the maximum discrepancy:

$$\eta \le \frac{\alpha}{10d_0} \min\left\{\varepsilon/(dH^3\omega), 1/\sqrt{m'_{\max}nH}\right\} = \tilde{O}\left(\frac{\varepsilon^2}{d^6 H^8}\right) \tag{21}$$

# C Proof of Proposition 3.4

***Proof of Proposition 3.4, and the MDP conversion argument.*** First, for *(i)*, we show the linearity of rewards with $\theta_1, \ldots, \theta_H$. For this take any $h \in [H]$. Fix any policy $\pi \in \Pi$ and let $\bar{\theta}_h \in \mathcal{B}(L_2)$ be such that for all $(s, a) \in \mathcal{S}_h \times [\mathcal{A}]$, $q^\pi(s, a) \approx_\eta \langle \varphi(s, a), \bar{\theta}_h \rangle$ (the existence of such a $\bar{\theta}$ follows from Definition 3.2). If $h = H$, $\mathbb{E}_{R \sim \mathcal{R}(s,a)}[R] = q^\pi(s, a)$, so $\theta_H = \bar{\theta}_H$ satisfies Definition 3.1. For $h < H$, let $f : \mathcal{S}_{h+1} \rightarrow \mathbb{R}$ be defined as $f(s) = v^\pi(s)$. Fix an arbitrary $Q \in \text{PD}^H$, e.g., $Q = (I, \ldots, I)$. Since $v^\pi(s) \in [0, H]$ and $\text{range}_Q(s) \geq \text{range}(s)/\sqrt{2d} \geq \alpha/\sqrt{2d}$ by Proposition 4.5, $f$ is $\alpha/(\sqrt{2d}H)$-admissible, and therefore by Lemma 4.7 we can take $\tilde{\theta}_h \in \mathcal{B}(4Hd_0\sqrt{2d}L_2/\alpha)$ such that for all $(s, a) \in \mathcal{S}_h \times [\mathcal{A}]$,

$$\mathbb{E}(v^\pi(S_{h+1}) \mid s, a) \approx_{\sqrt{2d}H\eta_0} \langle \varphi(s, a), \tilde{\theta}_h \rangle \, ,$$

where, as before, $\eta_0 = 5d_0\eta/\alpha$. Since

$$\mathop{\mathbb{E}}_{R \sim \mathcal{R}(s,a)} (R) = q^\pi(s, a) - \mathbb{E}(v^\pi(S_{h+1}) \mid s, a) \, ,$$

letting $\theta_h = \bar{\theta}_h - \tilde{\theta}_h$ satisfies *(i)* of Definition 3.1 with $\kappa = \eta + \sqrt{2d}H\eta_0 = \eta + 5H\sqrt{2d}d_0\eta/\alpha$.

To show *(ii)*, take any $f : \mathcal{S} \rightarrow [0, H]$ and $h \in [H - 1]$. As before, $f$ is $\alpha/(\sqrt{2d}H)$-admissible, therefore Lemma 4.7 immediately provides $\theta'_h$ satisfying the required conditions.

Therefore, the MDP is shown to be linear with misspecification $\eta + \sqrt{2d}H\eta_0$, and parameter bound $L_2(4Hd_0\sqrt{2d}/\alpha + 1)$. $\qquad\qquad\qquad\qquad\qquad\qquad\qquad\qquad\qquad\qquad\qquad\qquad\qquad\qquad\qquad\square$

***Sketch of the $q^\pi$-to-linear MDP conversion argument.*** We elaborate on the conversion to linear MDP mechanism presented in Section 3. As the basis of this argument is that an idealistic range-determining oracle is present, we note that this argument only serves as intuition and is otherwise tangential to our proof. Instead of a direct approach of learning this oracle, our proof argues that learning about this oracle happens whenever there is a need (performance shortfall) for it. A formal reduction to linear MDPs given this oracle however is fairly straightforward but cumbersome, with the caveat that the linear MDP will end up with $dH$ (instead of $d$) dimensional features. One would proceed by copying the features of each state $s$ in stage $h$ into the $h^{\text{th}}$ chunk of size $d$ of this vector of size $dH$ (the rest of the vector remains zero). A similar transformation is applied to all $\theta_h(\pi)$. Then, $H$ copies are made of each high-enough-range state, with all possible stages (but keeping the feature vectors). These will be the states of the new MDP we construct. When a transition from state $s$ leads to skipped states, the linear MDP returns with the copy of the first non-skipped state that has a stage counter of $\text{stage}(s) + 1$, so that in this linear MDP the stage numbers are consecutive (as required by our definitions). $q^\pi$-realizability of this modified MDP is easy to show, and (as it has no low-range states) Proposition 3.4 can be used to show that the modified MDP is linear. To account for the fact that this new MDP may finish an episode in fewer than $H$ steps due to the skips, we add a special, zero-reward, self-transitioning state called "episode-over". To ensure that the MDP stays linear, we extend the feature vectors of each state by a scalar 1, and a scalar indicator of being in this state, with all original features of the "episode-over" state defined to be zero. It is easy to see that this construction leads to a linear MDP with the desired action-value functions.

# D Intuition behind our method and proof strategy from the perspective of ELEANOR [Zanette et al., 2020]

The starting point of our method is the ELEANOR algorithm, which is designed for linear MDPs. Similarly to SKIPPYELEANOR, ELEANOR solves an optimistic optimization problem inside a loop. The optimization problem computes optimistic estimates $\bar{\theta}_t$ of the parameters of the MDP simultaneously for all $t \in [H]$, and in each iteration of the loop, more data is collected according to the policy that is optimal for the MDP defined by the estimated parameters. Initial estimates $\hat{\theta}_t$ are computed via solving least-squares problems whose covariates are the features corresponding to state-action pairs $(S_t, A_t)$ from all the data collected so far, while the corresponding least-squares targets are computed as the sum of the immediate reward $R_t$ and the estimated value for $S_{t+1}$, computed from $\bar{\theta}_{t+1}$. $\bar{\theta}_t$ is then optimistically chosen as the solution of the optimization problem, in the neighborhood (confidence ellipsoid) of $\hat{\theta}_t$, the solution to this least-squares problem. It is shown that this

optimistic choice of estimates results in an optimistic estimate of the value of $v^\star$ of the initial state, and the regret is upper bounded in terms of the sum of elliptic potentials of the covariates.

This argument appears in our analysis too, with minor modifications due to our PAC-like setting (instead of aiming to bound the regret), leading to our final-iteration condition of Line 16 in Algorithm 1. Our Optimization Problem 4.10 is similar to ELEANOR's, and the parameters $\bar{\theta}_t$ and $\hat{\theta}_t$ have the same meaning. A key difference between the optimization problems of ELEANOR and SKIPPYELEANOR are how the least-squares targets are determined. For ELEANOR, it is the sum of the immediate reward $R_t$ and its estimated value for $S_{t+1}$); with this target, only one on-policy rollout is required for each episode in order to get the least-squares parameter estimate for all $H$ stages. In contrast, our least-squares targets are formed as the sum of $R_t + \ldots + R_{t+i}$ and the estimated value for $S_{t+i+1}$, where $i$, the number of stages "skipped", depends on the guess $\hat{G}$. The guess $\hat{G}$ is selected only in Optimization Problem 4.10, and we do not know its value at the time of data collection, so we cannot know which stages will have to be skipped for each rollout. Therefore, (i) we need access to the rewards of the current policy at any stage (similarly to ELEANOR), and hence we run the current policy to any stage (including the last one); and (ii) perform rollouts with the fixed policy $\pi^0$ (from any stage) to be able to estimate the reward $R_t + \ldots + R_{t+i}$ collected while skipping over $i$ stages (for any $i$). To ensure this happens for every stage, we start Phase II from every stage $k$, resulting in the additional for loop in Line 6 of Algorithm 1 compared to ELEANOR. Finally, the randomization in Phase I is applied to make the optimization problem smooth, as described in Section 4.

One could analyze this algorithm similarly to the analysis of ELEANOR if it were not for the fact that the least-squares targets we just introduced are not realizable in general. We can, however, prove the realizability of certain components of the matrix-valued version of these targets, $F$ (Lemma 4.9 and Corollary 4.11). This enables us to detect when the realizability of our least-squares targets fail, measure the direction (component) of the largest error, and learn from that. This is the job of Optimization Problem 4.12: $\hat{F}_{\hat{G}\bar{\theta}}^{ki}$ corresponds to the matrix-valued empirical measurements of $F$, while the $y_{\hat{G}\bar{\theta}}^{ki}$ are the average predictions of the same quantities. If the targets are realizable, which happens if we manage to skip the right number of stages), these matrices are very close; if not, the direction of their largest discrepancy tells us something about $\perp(Q, i)$, and allows us to learn.

Optimism ties all this together: either there is no shortfall between predicted and measured $q$-values (and we are done) or we grow the elliptical potential of $X$ (the two cases present in the analysis of ELEANOR, Zanette et al. [2020]), or we grow the elliptical potential of $Q$ (the new case due to the lack of realizability guarantees).

# E    Proof of Theorem 4.1

In this section we present the proof of Theorem 4.1. Recall that some quantities are defined in Appendix B.

## E.1    Checking consistency

We introduce some lemmas to establish the required guarantees of the consistency checker. Their proofs, which rely on the usual least squares analysis techniques and covering arguments, are presented in Appendix H.

**Lemma E.1.** *Let $(k, i, v)$ be the outcome of Optimization Problem 4.12 any time during the execution of* SKIPPYELEANOR, *and let $w = \mathrm{Proj}_{Z(Q,i)} v$ as in the algorithm. Then,*

$$w^\top \left( y_{\hat{G}\bar{\theta}}^{ki} - \hat{F}_{\hat{G}\bar{\theta}}^{ki} \right) w \geq v^\top \left( y_{\hat{G}\bar{\theta}}^{ki} - \hat{F}_{\hat{G}\bar{\theta}}^{ki} \right) v - \frac{\varepsilon}{dH^2\omega}$$

**Lemma E.2.** *There is an event $\mathcal{E}_1$ that happens with probability at least $1 - \zeta$, such that under $\mathcal{E}_1$, during the execution of* SKIPPYELEANOR, *when the beginning of any iteration (Line 5) is executed, for any $t \in [H-1]$, $i \in [t+1 : H]$, for any $\hat{G} \in \mathbf{G}$, $\bar{\theta} \in \bar{\Theta}$, and $v, w \in \mathcal{B}(1)$, for all $(s, a) \in \mathcal{S}_t \times [\mathcal{A}]$,*

$$\left| v_\parallel^\top \left( \varphi(s,a)^\top \hat{\theta}_{\hat{G}\bar{\theta}}^{ti} - \mathbb{E}_{\pi^0,s,a} \bar{F}_{\hat{G}\bar{\theta}}(S_i) \right) w \right| \leq \|\varphi(s,a)\|_{X_{mt}^{-1}} \beta + \frac{\varepsilon}{dH^2\omega} ,$$

*where $\bullet_\parallel$ denotes $\bullet_{\parallel(Q,i)}$.*

The next lemma uses the average least-squares predictions' (capped) uncertainty term $\bar{\sigma}_k^m$ (defined in Eq. (18)), where the average is taken over predictions from the state-action pair where Phase I of SKIPPYPOLICY$(\cdot, \cdot, k)$ ends.

**Lemma E.3.** *There is an event $\mathcal{E}_2$ with probability at least $1 - \zeta$, such that under $\mathcal{E}_1 \cap \mathcal{E}_2$, during the execution of* SKIPPYELEANOR, *when Optimization Problem 4.12 is solved (Line 10), for $(\hat{G}, \bar{\theta})$ as recorded in Line 5 for all $k \in [H-1]$, $i \in [k+1:H]$, and $v, w \in \mathcal{B}(1)$,*

$$\left| v_{\parallel}^{\top} \left( y_{\hat{G}\bar{\theta}}^{ki} - \hat{F}_{\hat{G}\bar{\theta}}^{ki} \right) w \right| \le \bar{\sigma}_k^m \beta + 3 \frac{\varepsilon}{dH^2\omega}$$

*where $\bullet_{\parallel}$ denotes $\bullet_{\parallel(Q,i)}$.*

Together, these lemmas can be used to show that the vector $w$ derived from Line 10 in SKIPPYE-LEANOR is sufficiently aligned with both $Z(Q, \cdot)$ and the subspace $\text{Proj}_{\perp(Q,\cdot)}$ projects to, which leads to the following important result:

**Lemma E.4.** *Under the $\mathcal{E}_1 \cap \mathcal{E}_2$, if Line 13 is executed any time during the execution of* SKIP-PYELEANOR *(i.e., when the consistency check fails), then the resulting $Q$ continues to be a valid preconditioning.*

From now on, our lemmas assume the high-probability events of Lemmas E.2 and E.3 hold, and therefore $Q$ is a valid preconditioning at any time during the execution by Lemma E.4.

## E.2 Sample complexity bounds

We bound the number of iterations of $m$ that SKIPPYELEANOR can execute. The proof of the following lemma is presented in Appendix I:

**Lemma E.5.** *Throughout the execution of* SKIPPYELEANOR, $m \le m_{max}$.

Note that throughout the execution of SKIPPYELEANOR, $m' \le m'_{\max}$. As $m' - m$ equals the number of times Line 13 is executed, i.e., the sum of sequence lengths corresponding to $Q$, by Lemma 4.4,

**Corollary E.6.** *Under $\mathcal{E}_1 \cap \mathcal{E}_2$,* SKIPPYELEANOR *returns with a policy before exiting the while loop of Line 3, and as each iteration executes $Hn$ trajectories in Line 8, the number of interactions of* SKIPPYELEANOR *with the MDP is bounded by $\tilde{O}\left(H^{11}d^7/\varepsilon^2\right)$.*

## E.3 Performance guarantee

We next consider the $m^{\text{th}}$ iteration of SKIPPYELEANOR under the assumption that the consistency check passes, that is, Line 16 is executed. We intend to guarantee the performance of $\pi^{mH}$ in terms of $\sum_{t=1}^{H} \bar{\sigma}_k^m$, given that the optimization value $x$ satisfies $x \le \bar{\sigma}_k^m \beta \omega + 3 \frac{\varepsilon}{dH^2}$ (which follows from the execution reaching Line 16). Next we introduce variants of $c_{ki}^j$ and $\tilde{c}_{ki}^j$ (Eq. (20)) which act, instead of the data collected during the execution of the algorithm, on a trajectory $(S_h, A_h, R_h)_{h \in [H]}$ and corresponding stage mapping $p$ obtained by an independent run of SKIPPYPOLICY, which will be clear from the context: $\tilde{c}_{ki} = \mathbb{1}\{p(k) < i\}$, and

$$c_{ki} = \mathbb{1}\left\{p(k) < i \text{ and } \left\|\varphi(S_{p(k)}, A_{p(k)})\right\|_{X_{m,p(k)}^{-1}} < 2(\beta\omega dH)^{-1} \text{ and } \left\langle \varphi(S_{p(k)}, A_{p(k)}), \bar{\theta}_{p(k)} \right\rangle \ge 0 \right\}.$$

**Remark E.7.** *In our analysis we rely on the obvious fact that the laws of the trajectories of* SKIPPYPOLICY$(\hat{G}, \bar{\theta}, k)$ *and* SKIPPYPOLICY$(\hat{G}, \bar{\theta}, k+1)$ *are the same until stage $p(k+1)$ (as the policies are the same until then), for any parameters $\hat{G}$ and $\bar{\theta}$. This includes $S_{p(k+1)}$ but not $A_{p(k+1)}$ if $p(k+1) \le H$, and includes the whole trajectory ending with $R_H$ otherwise.*

We prove the following using induction on $k = H, \dots, 1$ in Appendix J:

**Lemma E.8.** *There is an event $\mathcal{E}_3$ with probability at least $1 - 3\zeta$, such that under $\mathcal{E}_1 \cap \mathcal{E}_2 \cap \mathcal{E}_3$, during the execution of* SKIPPYELEANOR, *whenever Line 16 is executed, for $(\hat{G}, \bar{\theta})$ as recorded in Line 5 of the current iteration, for $k \in [H]$,*

$$\bar{C}^k := \mathop{\mathbb{E}}_{\pi^{mk}, s_1} \tilde{c}_{k,H+1} C_{\hat{G}\bar{\theta}}(S_{p(k)}) \le \mathop{\mathbb{E}}_{\pi^{mH}, s_1} \sum_{u=p(k)}^{H} R_u + 2 \sum_{i=k}^{H} \bar{\sigma}_k^m \beta \omega dH + 4(H-k+1)\frac{\varepsilon}{H}. \quad (22)$$

As $S_1 = s_1$ is fixed and $\tau(s_1) = 1$, we get the following corollary, which shows that the value $C_{\hat{G}\bar{\theta}}$ of the solution $(\hat{G}, \bar{\theta})$ of Optimization Problem 4.10 can be used as a lower bound on the value of the policy $\pi^{mH}$ up to the uncertainty and some $\varepsilon$ terms:

**Corollary E.9.** *Under $\mathcal{E}_1 \cap \mathcal{E}_2 \cap \mathcal{E}_3$, the value of Optimization Problem 4.10 with the solution $(\hat{G}, \bar{\theta})$ satisfies*

$$C_{\hat{G}\bar{\theta}}(s_1) = \bar{C}^1 \leq \underset{\pi^{mH}, s_1}{\mathbb{E}} \sum_{u=1}^{H} R_u + 2 \sum_{i=1}^{H} \bar{\sigma}_k^m \beta \omega dH^2 + 4\varepsilon = v^{\pi^{mH}}(s_1) + 2 \sum_{i=1}^{H} \bar{\sigma}_k^m \beta \omega dH^2 + 4\varepsilon .$$

### E.4 Optimism of Optimization Problem 4.10

The following establishes the optimistic property, that is, that the value of Optimization Problem 4.10 competes with $v^{\star}(s_1)$. The proof relies on the fact that the correct guess $\hat{G}$ and a good choice of $\bar{\theta}$ are feasible for the optimization problem, combined with the fact that this $\bar{\theta}$ induces a policy $\pi = \text{SKIPPYPOLICY}(\hat{G}, \bar{\theta}, H)$ that takes action-value maximizing actions according to a very accurate approximation of action-values almost everywhere. In fact, it only skips states whose range is at most $\varepsilon/H$. The proof is presented in Appendix K.

**Lemma E.10.** *There is an event $\mathcal{E}_4$ with probability at least $1 - \zeta$, such that under $\mathcal{E}_1 \cap \mathcal{E}_2 \cap \mathcal{E}_4$, throughout the execution of* SKIPPYELEANOR*, the value of Optimization Problem 4.10 is at least $v^{\star}(s_1) - 2\varepsilon$.*

*Proof of Theorem 4.1.* We combine Lemma E.10 with Corollary E.9, Corollary E.6, and the fact that the condition of Line 16 is satisfied when SKIPPYELEANOR returns with a policy, to get that under $\mathcal{E}_1 \cap \mathcal{E}_2 \cap \mathcal{E}_3 \cap \mathcal{E}_4$, that is, with probability at least $1 - 6\zeta$, SKIPPYELEANOR interacts with the MDP for at most $\tilde{O}\left(H^{11}d^7/\varepsilon^2\right)$ many steps, before returning with the policy $\pi^{mH}$ that satisfies

$$v^{\star}(s_1) \leq C_{\hat{G}\bar{\theta}}(s_1) + 2\varepsilon \leq v^{\pi^{mH}}(s_1) + 2 \sum_{i=1}^{H} \bar{\sigma}_k^m \beta \omega dH^2 + 6\varepsilon \leq v^{\pi^{mH}}(s_1) + 8\varepsilon ,$$

where the final inequality follows from the fact that when SKIPPYELEANOR returns in Line 17, $\sum_{k=1}^{H} \bar{\sigma}_k^m \leq \varepsilon/(\beta \omega dH^2)$. By scaling the parameters, this finishes the proof of Theorem 4.1. $\qquad\square$

## F   Deferred definitions and proofs for Section 4.1

***Proof of Lemma 4.3.*** For any $\theta \in \Theta_h^Q$, it holds that $\theta = Q_h^{-1}\hat{\theta}$ for some $\hat{\theta} \in \Theta_h$. Since $\left\|\hat{\theta}\right\|_2 \leq L_2$, and writing $Q_h$ as in Definition 4.2,

$$\|\theta\|_2^2 = \hat{\theta}^\top \left(L_2^{-2}I + \sum_{v \in C_h} vv^\top\right) \hat{\theta} \leq L_2^{-2}L_2^2 + |C_h| \leq 1 + d_1 ,$$

where we used Definition 4.2 and Lemma 4.4. Finally, we conclude that $\|\theta\|_2 \leq \sqrt{d_1 + 1}$. $\qquad\square$

**Definition F.1.** $(G_h^Q, \rho_h^Q)$ *is a near-optimal design for* $\Theta_h^Q$*, if for any* $\theta \in \Theta_h^Q$*,*

$$\langle v, \theta \rangle = 0 \quad \text{for all } v \in \text{Ker}(V(G_h^Q, \rho_h^Q)), \text{ and} \tag{23}$$

$$\|\theta\|_{V(G_h^Q, \rho_h^Q)^\dagger}^2 \leq 2d, \tag{24}$$

$$\text{where } V(G_h^Q, \rho_h^Q) = \sum_{\pi \in G_h^Q} \rho_h^Q(\pi)(\theta_h^Q(\pi))(\theta_h^Q(\pi))^\top . \tag{25}$$

An important corollary of the above definition is that if $M = \text{Proj}_{\text{Im}(V(G_h^Q, \rho_h^Q))}$, then $V(G_h^Q, \rho_h^Q)^{\dagger\frac{1}{2}}V(G_h^Q, \rho_h^Q)^{\frac{1}{2}}Mv = Mv$, and $\langle \theta, Mv \rangle = \langle \theta, v \rangle$ due to Eq. (23), and so

$$\theta^\top v = \theta^\top V(G_h^Q, \rho_h^Q)^{\dagger\frac{1}{2}}V(G_h^Q, \rho_h^Q)^{\frac{1}{2}}v \quad \text{for all } \theta \in \Theta_h^Q \text{ and } v \in \mathbb{R}^d. \tag{26}$$

***Proof of Proposition 4.5.*** Take any $h \in [H]$, $s \in \mathcal{S}_h$, and $Q \in \mathrm{PD}^H$. Take $i, j \in [\mathcal{A}]$ such that $\mathrm{range}(s) = \sup_{\theta \in \Theta_h} \langle \varphi(s, i, j), \theta \rangle$. Then,

$$
\begin{aligned}
\mathrm{range}(s)^2 &= \sup_{\theta \in \Theta_h} \langle \varphi(s, i, j), \theta \rangle^2 = \sup_{\theta \in \Theta_h^Q} \langle \varphi_Q(s, i, j), \theta \rangle^2 \\
&\leq \sup_{\theta \in \Theta_h^Q} \|\theta\|_{V(G_h^Q, \rho_h^Q)^\dagger}^2 \left\|\varphi_Q(s, i, j)\right\|_{V(G_h^Q, \rho_h^Q)}^2 \\
&\leq 2d \varphi_Q(s, i, j)^\top \left( \sum_{\pi \in G_h^Q} \rho_h^Q(\pi) (\theta_h^Q(\pi)) (\theta_h^Q(\pi))^\top \right) \varphi_Q(s, i, j) \\
&= 2d \varphi(s, i, j)^\top \left( \sum_{\pi \in G_h^Q} \rho_h^Q(\pi) (\theta_h(\pi)) (\theta_h(\pi))^\top \right) \varphi(s, i, j) \\
&\leq 2d \max_{\pi \in G_h^Q} \left\langle \varphi(s, i, j)^\top, \theta_h(\pi) \right\rangle^2 \leq 2d \, \mathrm{range}_Q(s)^2,
\end{aligned}
$$

where the first inequality uses Eq. (26) and the Cauchy-Schwarz inequality, and the second inequality follows by substituting the definition of $V(G_h^Q, \rho_h^Q)$ and using Eq. (24). Finally, the first inequality in the last line holds as we replace the weighted sum from the previous line with the maximum operator. We therefore get that $\mathrm{range}(s) \leq \sqrt{2d} \, \mathrm{range}_Q(s)$, finishing the proof. □

***Proof of Lemma 4.4.*** Take any $h \in [H]$ and the sequence $C_h$ corresponding to $Q$. Assume that this sequence is of length $l$, and let $\Sigma_{h,i} = L_2^{-2} I + \sum_{j=1}^i v_j v_j^\top$ for $i \in [l]$. By the second part of Eq. (5),

$$
l = \sum_{i=1}^l \min \left\{ 1, 2 \left\| \left( L_2^{-2} I + \sum_{j=1}^{i-1} v_j v_j^\top \right)^{-\frac{1}{2}} v_i \right\|_2^2 \right\} \leq 2 \sum_{i=1}^l \min \left\{ 1, \|v_i\|_{\Sigma_{h,i-1}^{-1}}^2 \right\} .
$$

Applying the elliptical potential lemma (Lemma L.1),

$$
l \leq 2 \sum_{i=1}^l \min \left\{ 1, \|v_i\|_{\Sigma_{h,i-1}^{-1}}^2 \right\} \leq 4d \log \left( \frac{\mathrm{Tr}(\Sigma_{h,0}) + l L_3^2}{d \det(\Sigma_{h,0})^{1/d}} \right) = 4d \log \left( 1 + \frac{l L_3^2}{L_2^{-2} d} \right) .
$$

where $\Sigma_{h,0} = L_2^{-2} I$ by definition. Using that $\log(1 + x) \leq \sqrt{x}$ for $x \geq 0$, we have $l \leq 4d\sqrt{l L_3^2 L_2^2 / d}$, which implies $l \leq 16 d L_3^2 L_2^2$. Substituting this into the previous bound yields

$$
l \leq 4d \log \left( 1 + \frac{(16 d L_3^2 L_2^2) L_3^2}{L_2^{-2} d} \right) = 4d \log \left( 1 + 16 L_3^4 L_2^4 \right) = d_1 . \qquad \square
$$

# G    Deferred proofs for Section 4.2

For any vector $v \in \mathbb{R}^d$, we define $\bar{v} = v / \|v\|_2$ as the unit vector in the direction of $v$ if $v \neq \mathbf{0}$ and $\bar{\mathbf{0}} = 0$ otherwise. For any $h \in [2 : H]$, $s \in \mathcal{S}_h$, the normalized version of the largest preconditioned feature difference is denoted by

$$
\bar{\varphi}_Q(s) = \overline{\varphi_Q(s, i, j)} \quad \text{where } (i, j) = \arg\max_{i', j' \in [\mathcal{A}]} \left\| \varphi_Q(s, i', j') \right\|_2 . \tag{27}
$$

***Proof of Lemma 4.7.*** Fix $h \in [H]$, $\alpha$-admissible $f : \mathcal{S}_h \to \mathbb{R}$, $t \in [h-1]$, and $\pi \in \Pi$. Our aim is to construct policies $\pi_k^+, \pi_k^- \in \Pi$ for $k \in [d_0]$, such that for all $(s, a) \in \mathcal{S}_t \times [\mathcal{A}]$, $\sum_{k \in [d_0]} (q^{\pi_k^+}(s, a) - q^{\pi_k^-}(s, a))$ is approximately proportional to the desired $\mathbb{E}_{\pi, s, a} f(S_h)$. Let $G_{h,1}^Q, G_{h,2}^Q, \ldots$ denote the policies in $G_h^Q$ underlying the near-optimal design of $\Theta_h^Q$, and for any $s \in \mathcal{S}_h$, denote by $\mathrm{ord}(s) \in$

$[d_0]$ the index of the policy maximizing the range of the action-value function in state $s$, that is, $G_{h,\mathrm{ord}(s)}^{Q} = \arg\max_{\pi \in G_h^Q} \max_{i,j \in [\mathcal{A}]} (q^\pi(s,i) - q^\pi(s,j))$; to simplify notation, we define $\tilde{G}(s) = G_{h,\mathrm{ord}(s)}^{Q}$. For $s \in \mathcal{S}_h$ let

$$(a^+(s), a^-(s)) = \begin{cases} \arg\max_{i,j \in [\mathcal{A}]} \hat{q}^{\tilde{G}(s)}(s,i) - \hat{q}^{\tilde{G}(s)}(s,j) & \text{if } f(s) \geq 0 \\ \arg\min_{i,j \in [\mathcal{A}]} \hat{q}^{\tilde{G}(s)}(s,i) - \hat{q}^{\tilde{G}(s)}(s,j) & \text{otherwise.} \end{cases}$$

By Eq. (7) and Definition 4.6 have that

$$\left| \hat{q}^{\tilde{G}(s)}(s, a^+(s)) - \hat{q}^{\tilde{G}(s)}(s, a^-(s)) \right| = \mathrm{range}_Q(s) \geq \alpha |f(s)| \geq 0.$$

Since $q^{\tilde{G}(s)}(s, a^+(s)) - q^{\tilde{G}(s)}(s, a^-(s)) \approx_{2\eta} \hat{q}^{\tilde{G}(s)}(s, a^+(s)) - \hat{q}^{\tilde{G}(s)}(s, a^-(s))$, if $\alpha |f(s)| \geq 4\eta$, we have

$$\begin{aligned} q^{\tilde{G}(s)}(s, a^+(s)) - q^{\tilde{G}(s)}(s, a^-(s)) &\geq \alpha f(s) - 2\eta \geq \frac{\alpha}{2} f(s) > 0 & \text{if } f(s) \geq 0 \\ q^{\tilde{G}(s)}(s, a^+(s)) - q^{\tilde{G}(s)}(s, a^-(s)) &\leq \alpha f(s) + 2\eta \leq \frac{\alpha}{2} f(s) < 0 & \text{otherwise.} \end{aligned} \tag{28}$$

Let us define $f' : \mathcal{S}_h \to \mathbb{R}$ as

$$f'(s) = \begin{cases} \frac{\alpha f(s)/2}{q^{\tilde{G}(s)}(s, a^+(s)) - q^{\tilde{G}(s)}(s, a^-(s))} & \text{if } \alpha |f(s)| \geq 4\eta \\ 0 & \text{otherwise.} \end{cases}$$

By Eq. (28), there can be no division by zero in the above definition, and $0 \leq f'(s) \leq 1$.

Now we are ready to define $\pi_k^+$ and $\pi_k^-$. Both policies follow $\pi$ up to stage $h-1$, when they switch to $G_{h,k}^Q$, except if at stage $h$ a state $s \in \mathcal{S}_h$ is such that $G_{h,k}^Q$ has the maximal action-value function range. In this case $\pi_k^+$ selects $a^+(s)$ with probability $f'(s)$ and $a^-(s)$ with probability $1 - f'(s)$, while $\pi_k^-$ always selects $a^-(s)$. Formally, for $k \in [d_0]$, we define for $s \in \mathcal{S}$

$$\pi_k^+(s) = \begin{cases} \pi(s) & \text{if } \mathrm{stage}(s) < h; \\ a^+(s) \text{ w.p. } f'(s), \text{ and } a^-(s) \text{ w.p. } 1 - f'(s) & \text{if } \mathrm{stage}(s) = h \text{ and } \mathrm{ord}(s) = k; \\ G_{h,k}^Q(s), & \text{otherwise,} \end{cases}$$

where w.p. stands for *with probability*. Similarly,

$$\pi_k^-(s) = \begin{cases} \pi(s) & \text{if } \mathrm{stage}(s) < h; \\ a^-(s) \text{ w.p. } 1 & \text{if } \mathrm{stage}(s) = h \text{ and } \mathrm{ord}(s) = k; \\ G_{h,k}^Q(s) & \text{otherwise.} \end{cases}$$

Note that $\pi_k^+ \in \Pi$ and $\pi_k^- \in \Pi$, as desired. Since for all $k \in [d_0]$, the policies follow $G_{h,k}$ for $s \in \mathcal{S}_{t'}$ for $t' > h$, therefore for all $k \in [d_0]$,

$$v^{\pi_k^-}(s) = v^{\pi_k^+}(s) = v^{G_{h,k}^Q}(s) \quad \text{for all } s \in \mathcal{S}_{h+1}, \text{ and} \tag{29}$$

$$q^{\pi_k^-}(s,a) = q^{\pi_k^+}(s,a) = q^{G_{h,k}^Q}(s,a) \quad \text{for all } (s,a) \in \mathcal{S}_h \times [\mathcal{A}]. \tag{30}$$

Also, for any $s \in \mathcal{S}$ with $\mathrm{stage}(s) < h$ and any $a \in [\mathcal{A}]$,

$$\begin{aligned} \sum_{k \in [d_0]} \left( q^{\pi_k^+}(s,a) - q^{\pi_k^-}(s,a) \right) &= \mathop{\mathbb{E}}_{\pi,s,a} \sum_{k \in [d_0]} \left( v^{\pi_k^+}(S_h) - v^{\pi_k^-}(S_h) \right) \\ &= \mathop{\mathbb{E}}_{\pi,s,a} \left( v^{\pi_{\mathrm{ord}(S_h)}^+}(S_h) - v^{\pi_{\mathrm{ord}(S_h)}^-}(S_h) \right) \\ &= \mathop{\mathbb{E}}_{\pi,s,a} \left( q^{\tilde{G}(S_h)}(S_h, a^+(S_h)) f'(S_h) + q^{\tilde{G}(S_h)}(S_h, a^-(S_h))(1 - f'(S_h)) \right. \\ &\qquad\qquad \left. - q^{\tilde{G}(S_h)}(S_h, a^-(S_h)) \right) \\ &= \mathop{\mathbb{E}}_{\pi,s,a} \left( f'(S_h) \left( q^{\tilde{G}(S_h)}(S_h, a^+(S_h)) - q^{\tilde{G}(S_h)}(S_h, a^-(S_h)) \right) \right) \\ &= \mathop{\mathbb{E}}_{\pi,s,a} \mathbb{I}\{\alpha |f(S_h)| \geq 4\eta\} \frac{\alpha}{2} f(S_h) \approx_{2\eta} \frac{\alpha}{2} \mathop{\mathbb{E}}_{\pi,s,a} f(S_h), \end{aligned}$$

where the first line is due to both $q^{\pi_k^+}$ and $q^{\pi_k^-}$ following $\pi$ on states with stage less than $h$, the second line follows from the fact that for any $s \in \mathcal{S}_h$, $\pi_k^+(s) = \pi_k^-(s)$ for any $k \neq \mathrm{ord}(s)$; combining this with Eq. (29) leads to all $k \neq \mathrm{ord}(s)$ terms of the sum to cancel. The third line follows from expanding the definition of the policies and Eq. (30).

Let $\tilde{\theta} = \frac{2}{\alpha} \sum_{k \in [d_0]} \left( \theta_t(\pi_k^+) - \theta_t(\pi_k^-) \right)$. Since $\|\theta_t(\cdot)\|_2 \leq L_2$ by definition, we have $\|\tilde{\theta}\|_2 \leq 4 d_0 L_2 / \alpha$. By Definition 3.2, for all $(s,a) \in \mathcal{S}_t \times [\mathcal{A}]$,

$$\left\langle \varphi(s,a), \frac{\alpha}{2} \tilde{\theta} \right\rangle \approx_{2 d_0 \eta} \sum_{k \in [d_0]} q^{\pi_k^+}(s,a) - q^{\pi_k^-}(s,a) \approx_{2\eta} \frac{\alpha}{2} \mathop{\mathbb{E}}_{\pi,s,a} f(S_h),$$

and hence

$$\left\langle \varphi(s,a), \tilde{\theta} \right\rangle \approx_{4(d_0+1)\eta/\alpha} \mathop{\mathbb{E}}_{\pi,s,a} f(S_h).$$

Since $4(d_0 + 1)\eta/\alpha \leq \eta_0 = 5 d_0 \eta / \alpha$ as $d_0 \geq 4$ by definition, this completes the proof. $\qquad\square$

***Proof of Lemma 4.9.*** Take any $s \in \mathcal{S}_h$. For the correct guess, $\mathrm{range}_Q^{\hat{G}}(s) = \mathrm{range}_Q(s)$. Then, using that $\|\bar{\varphi}_Q(\cdot)\|_2 \leq 1$, $\mathrm{Tr}(\mathbf{f}(s)) \leq \mathrm{range}_Q(s) \frac{\sqrt{2d}H^2}{\varepsilon}$, proving the second claim of the lemma (as $\gamma \leq 1$).

For the first claim, take any $\hat{G} = (\hat{\vartheta}_h^i)_{h \in [2:H], i \in [d_0]} \in \mathbf{G}$. Let $\varphi'$ be the unnormalized version of $\bar{\varphi}_Q(s)$ of Eq. (27), that is, $\varphi' = \varphi_Q(s,i,j)$ for the same $i, j$ as in Eq. (27) (i.e., with the largest $\ell_2$-norm). Then, using that $\hat{G} \in \mathbf{G}$,

$$\mathrm{range}_Q^{\hat{G}}(s) = \max_{k \in [d_0]} \max_{i,j} \left\langle \varphi_Q(s,i,j), \hat{\vartheta}_h^k \right\rangle \leq \|\varphi'\|_2 \max_{k \in [d_0]} \|\hat{\vartheta}_h^k\|_2 \leq \|\varphi'\|_2 \sqrt{d_1 + 1}.$$

Using that above in combination with $|f(s)| \leq H$, $v, w \in \mathcal{B}(1)$, $\|\bar{\varphi}_Q(s)\|_2 \leq 1$, we obtain

$$|v_\parallel^\top \mathbf{f}(s) w| \leq \left| \left\langle \bar{\varphi}_Q(s), v_\parallel \right\rangle \left\langle \bar{\varphi}_Q(s), w \right\rangle \right| \mathrm{range}_Q^{\hat{G}}(s) \frac{\sqrt{2d}H^2}{\varepsilon}$$

$$\leq \|\bar{\varphi}_Q(s)_\parallel\|_2 \|\varphi'\|_2 \sqrt{d_1 + 1} \frac{\sqrt{2d}H^2}{\varepsilon} = \|\varphi'_\parallel\|_2 \sqrt{d_1 + 1} \frac{\sqrt{2d}H^2}{\varepsilon}.$$

As the eigenvalues of $V(G_h^Q, \rho_h^Q) = \sum_{\pi \in G_h^Q} \rho_h^Q(\pi)(\theta_h^Q(\pi))(\theta_h^Q(\pi))^\top$ corresponding to the subspace in which $\varphi'_\parallel$ lies are by definition at least $\gamma$, we can write

$$(\mathrm{range}_Q(s))^2 \geq \max_{\pi \in G_h^Q} \left\langle \varphi', \theta_h^Q(\pi) \right\rangle^2 \geq \varphi'^\top V(G_h^Q, \rho_h^Q) \varphi' \geq \varphi'^\top_\parallel V(G_h^Q, \rho_h^Q) \varphi'_\parallel \geq \|\varphi'_\parallel\|_2^2 \gamma.$$

Combining with the previous result, we get that

$$\mathrm{range}_Q(s) \geq \sqrt{\gamma} \|\varphi'_\parallel\|_2 \geq \frac{\sqrt{\gamma}\varepsilon}{\sqrt{2d}\sqrt{d_1 + 1}H^2} |v_\parallel^\top \mathbf{f}(s) w| = \alpha |v_\parallel^\top \mathbf{f}(s) w|,$$

finishing the proof. $\qquad\square$

# H   Deferred proofs for Appendix E.1

The definitions (Eqs. (9) and (10)) immediately give rise to the following facts:

$$D(s_i \to) \in \left[ -\sum_{u=i}^H r_u, H \right] \subseteq [-H, H] \text{ and } \tau(\cdot) \in [0,1], \text{ implying}$$

$$E^\to(s_i \to) \in \left[ -\sum_{u=i}^H r_u, H \right] \subseteq [-H, H], \text{ implying} \tag{31}$$

$$E(s_i \to) \in [-2\tau(s_i)H, 2\tau(s_i)H] \subseteq [-2H, 2H].$$

Furthermore, since either $\tau(s_i) = 0$ or $\|\bar{\varphi}_Q(s_i)\|_2 = 1$ (as $\|\bar{\varphi}_Q(s_i)\| = 0$ implies that $\mathrm{range}_Q(s_i)$ and hence $\tau(s_i)$ are both zero), we have

$$\mathrm{Tr}(F(s_i \to)) = E(s_i \to), \tag{32}$$

which was used to establish the last part of Corollary 4.11.

**_Proof of Lemma E.1._** We drop the subscripts $(\hat{G}, \bar{\theta})$. Let $(\hat{\vartheta}_h^i)_{h \in [2:H], i \in [d_0]} = \hat{G} \in \mathbf{G}$. Let $z = v - w$ be the projection of $v$ to the subspace orthogonal to $Z(Q, i)$, denoted by $Z(Q, i)^\perp$. In other words, $z = \mathrm{Proj}_{Z(Q,i)^\perp} v$. Let $\mathbf{M} = y^{ki} - \hat{F}^{ki}$. By the symmetry of $\mathbf{M}$,

$$v^\top \mathbf{M} v = z^\top \mathbf{M}(v + w) + w^\top \mathbf{M} w \,.$$

It is enough to prove therefore that

$$\frac{\varepsilon}{dH^2\omega} \geq z^\top \mathbf{M}(v + w) \,.$$

As $\|v\|_2 \leq 1$ and $\|v + w\|_2 \leq 2$, and using the definitions and Eq. (31), for any input $(s_i \rightarrow)$,

$$\left| z^\top F(s_i \rightarrow)(v + w) \right| = \left| \langle z, \bar{\varphi}_Q(s_i) \rangle \, \langle v + w, \bar{\varphi}_Q(s_i) \rangle \, E(s_i \rightarrow) \right|$$

$$\leq 4H\tau(s_i) \left| \langle z, \bar{\varphi}_Q(s_i) \rangle \right| \leq 4 \, \mathrm{range}_Q^{\hat{G}}(s_i) \frac{\sqrt{2d}H^2}{\varepsilon} \left| \langle z, \bar{\varphi}_Q(s_i) \rangle \right|$$

$$\leq 4 \left| \langle z, \bar{\varphi}_Q(s_i) \rangle \right| \max_{a,b,k \in [d_0]} \langle \varphi_Q(s, a, b), \hat{\vartheta}_h^k \rangle \frac{\sqrt{2d}H^2}{\varepsilon}$$

$$\leq 4 \left\| \mathrm{Proj}_{Z(Q,i)^\perp} \bar{\varphi}_Q(s) \right\|_2 \|\varphi'\|_2 \max_{k \in [d_0]} \left\| \hat{\vartheta}_h^k \right\|_2 \frac{\sqrt{2d}H^2}{\varepsilon}$$

$$\leq 4 \left\| \mathrm{Proj}_{Z(Q,i)^\perp} \varphi' \right\|_2 \sqrt{d_1 + 1} \frac{\sqrt{2d}H^2}{\varepsilon} \,,$$

where $\varphi'$ is the unnormalized version of $\bar{\varphi}_Q(s_i)$ of Eq. (27), that is, $\varphi' = \varphi_Q(s_i, a, b)$ for the same $a, b$ as in Eq. (27) (i.e., with the largest $\ell_2$-norm).

As $\mathrm{Proj}_{Z(Q,i)^\perp} \varphi' = \mathrm{Proj}_{Z(Q,i)^\perp}(\varphi_Q(s, a) - \varphi_Q(s, b)) = \mathrm{Proj}_{Z(Q,i)^\perp} Q_i(\varphi(s, a) - \varphi(s, b))$ for some $s \in \mathcal{S}_i, a, b \in [\mathcal{A}]$, and by definition $\mathrm{Proj}_{Z(Q,i)^\perp} Q_i \preceq L_3^{-2} I$, $\left\| \mathrm{Proj}_{Z(Q,i)^\perp} \varphi' \right\|_2 \leq L_3^{-2} \|\varphi(s, a) - \varphi(s, b)\|_2 \leq 2L_3^{-2} L_1$, so

$$\left| z^\top F(s_i \rightarrow)(v + w) \right| \leq 8L_3^{-2} L_1 \sqrt{d_1 + 1} \frac{\sqrt{2d}H^2}{\varepsilon}, \tag{33}$$

and hence

$$\left| z^\top \hat{F}^{ki}(v + w) \right| \leq 8L_3^{-2} L_1 \sqrt{d_1 + 1} \frac{\sqrt{2d}H^2}{\varepsilon} \,. \tag{34}$$

To bound $\left| z^\top y^{ki}(v + w) \right|$, note that by the definition $y^{ki}$,

$$z^\top y^{ki}(v + w) = \frac{1}{n} \sum_{j \in [n]} c_{ki}^j \left\langle \varphi_{p(k)}^{mkj}, \check{\theta}^{p(mkj),i} \right\rangle$$

$$\text{where} \qquad \check{\theta}^{ti} = X_{mt}^{-1} \sum_{lkj \in \mathbf{I}^m(t)} \varphi_t^{lkj} \left( z^\top F(S_i^{lkj}, \ldots, R_H^{lkj})(v + w) \right) \qquad \text{for } t \in [i - 1]$$

Therefore

$$\left| z^\top y^{ki}(v + w) \right| \leq \max_{t \in [i-1], s \in \mathcal{S}_t, a \in [\mathcal{A}]} \left\langle \varphi(s, a), \check{\theta}^{ti} \right\rangle \,.$$

Fix any $t \in [i - 1], s \in \mathcal{S}_t, a \in [\mathcal{A}]$. By repeated application of the Cauchy-Schwarz inequality, the fact that $X_{mt} \succeq \lambda I$, the triangle inequality, and using Eq. (33),

$$\left| \left\langle \varphi(s, a), \check{\theta}^{ti} \right\rangle \right| \leq \|\varphi(s, a)\|_{X_{mt}^{-1}} \left\| \sum_{lkj \in \mathbf{I}^m(t)} \varphi_t^{lkj} \left( z^\top F(S_i^{lkj}, \ldots, R_H^{lkj})(v + w) \right) \right\|_{X_{mt}^{-1}}$$

$$\leq \|\varphi(s, a)\|_2 \, \lambda^{-1/2} \cdot 8L_3^{-2} L_1 \sqrt{d_1 + 1} \frac{\sqrt{2d}H^2}{\varepsilon} \sum_{lkj \in \mathbf{I}^m(t)} \left\| \varphi_t^{lkj} \right\|_{X_{mt}^{-1}}$$

$$\leq 8L_3^{-2} L_1^2 \lambda^{-1/2} \sqrt{d_1 + 1} \frac{\sqrt{2d}H^2}{\varepsilon} \sqrt{|\mathbf{I}^m(t)|} \sqrt{\sum_{lkj \in \mathbf{I}^m(t)} \left\| \varphi_t^{lkj} \right\|_{X_{mt}^{-1}}^2}$$

$$\leq 8L_3^{-2} L_1^2 \lambda^{-1/2} \sqrt{d_1 + 1} \frac{\sqrt{2d}H^2}{\varepsilon} \sqrt{m_{\max} nHd} \,,$$

where we use that $|\mathbf{I}^m(t)| \le mnH$, $m \le m_{\max}$ by Lemma E.5, and that

$$\sqrt{\sum_{lkj \in \mathbf{I}^m(t)} \left\|\varphi_t^{lkj}\right\|_{X_{mt}^{-1}}^2} = \sqrt{\sum_{lkj \in \mathbf{I}^m(t)} \text{Tr}(X_{mt}^{-1}\varphi_t^{lkj}\varphi_t^{lkj\top})} \le \sqrt{\text{Tr}\, X_{mt}^{-1}X_{mt}} = \sqrt{d}\,.$$

Combining with Eq. (34), with an appropriate choice of $L_3$, we obtain

$$\left|z^\top \mathbf{M}(v+w)\right| \le 8L_3^{-2}L_1\sqrt{d_1+1}\frac{\sqrt{2d}H^2}{\varepsilon}\left(1 + L_1\lambda^{-1/2}\sqrt{m_{\max}nHd}\right) \le \frac{\varepsilon}{dH^2\omega} \qquad (35)$$

as desired. $\qquad\square$

***Proof of Lemma E.2.*** Choose $\beta$

$$\beta \le 2 + 2H\sqrt{2dH(d_0+1)\log\frac{12d_0HL_2}{\alpha\xi} + 2\log\frac{m'_{\max}H^2}{\zeta} + d\log\left(\lambda + m'_{\max}nHL_1^2/d\right)}, \qquad (36)$$

satisfying $\beta = \tilde{O}(H^{3/2}d)$ as given in Eq. (15), and define

$$\xi = \frac{\varepsilon}{5\sqrt{2d}(H+1)^3L_1}\left(\min\left\{\varepsilon/(dH^2\omega), 1/\sqrt{m'_{\max}nH}\right\} - \eta_0\right).$$

Note that subtracting $\eta_0$ keeps $\xi$ positive, and of the same order, by our assumption that $\eta$ is small enough: $\eta_0 \le \frac{1}{2}\min\left\{\varepsilon/(dH^2\omega), 1/\sqrt{m'_{\max}nH}\right\}$, which follows from Eq. (21).

We start with a covering argument for the set of functions of the form $v_\|^\top \bar{F}_{\hat{G}\bar{\theta}}w$, for different choices of $\hat{G}$, $\bar{\theta}$, $v$, and $w$. By [Vershynin, 2018, Corollary 4.2.13], there is a set $C_\xi \subset \mathcal{B}(1)$ with $|C_\xi| \le (3/\xi)^d$ such that for all $x \in \mathcal{B}(1)$ there exists a $y \in C_\xi$ with $\|x - y\|_2 \le \xi$. Therefore, there is a set $C_\xi^\times \subset \left(\times_{h\in[2:H],k\in[d_0]}\mathcal{B}(\sqrt{d_1+1})\right) \times \left(\times_{h\in[2:H]}\mathcal{B}(4d_0HL_2/\alpha)\right) \times \mathcal{B}(1) \times \mathcal{B}(1)$ with $|C_\xi^\times| \le (12d_0HL_2/(\alpha\xi))^{dH(d_0+1)}$ such that for any $\hat{G} = (\hat{\vartheta}_h^i)_{h\in[2:H],i\in[d_0]} \in \mathbf{G}$, $\bar{\theta} \in \bar{\Theta}$, and $v,w \in \mathcal{B}(1)$, there exists a $y \in C_\xi^\times$, such that if we let $\tilde{G} = (\tilde{\vartheta}^i)_{h\in[2:H],i\in[d_0]} = (y_{(h-1)d_0+i})_{h\in[2:H],i\in[d_0]}$, $\tilde{\theta} = (\tilde{\theta}_h)_{h\in[2:H]} = (y_{(H-1)d_0+h})_{h\in[2:H]}$, and $a = y_{(H-1)(d_0+1)+1}$, $b = y_{(H-1)(d_0+1)+2}$, then $\tilde{G} \in \mathbf{G}$, $\tilde{\theta} \in \bar{\Theta}$, $a,b \in \mathcal{B}(1)$, and

$$\|a - v\|_2 \le \xi \quad \text{and} \quad \|b - w\|_2 \le \xi, \quad \text{and}$$

$$\left\|\hat{\vartheta}_h^i - \tilde{\vartheta}^i\right\|_2 \le \xi \text{ and } \left\|\bar{\theta}_h - \tilde{\theta}_h\right\|_2 \le \xi \text{ for all } h \in [2:H], i \in [d_0]\,.$$

As a result, for all $s \in \mathcal{S} \setminus \mathcal{S}_1$, $|\text{range}_Q^{\hat{G}}(s) - \text{range}_Q^{\tilde{G}}(s)| \le 2L_1\xi$, and therefore $|\tau_{\hat{G}\bar{\theta}}(s) - \tau_{\tilde{G}\bar{\theta}}(s)| \le 2\sqrt{2d}HL_1\xi/\varepsilon$. Furthermore, $|D_{\hat{G}\bar{\theta}}(s,\ldots,r_H) - D_{\tilde{G}\bar{\theta}}(s,\ldots,r_H)| \le L_1\xi$. Combining these with the facts that in either case, $\tau(\cdot) \in [0,1]$, $D(\cdot) \in [-H,H]$, and $E^\to(\cdot) \in [-H,H]$ (Eq. (31)), and using the definition of $E$ and $E^\to$, we have that for any $i \in [H+1]$ and inputs,

$$|E_{\hat{G}\bar{\theta}}(s_i\to) - E_{\tilde{G}\bar{\theta}}(s_i\to)| \le 4\sqrt{2d}H^2L_1\xi/\varepsilon + L_1\xi + |E_{\hat{G}\bar{\theta}}^\to(s_{i+1}\to) - E_{\tilde{G}\bar{\theta}}^\to(s_{i+1}\to)|$$

$$= 4\sqrt{2d}H^2L_1\xi/\varepsilon + L_1\xi + \sum_{j=i+1}^{H}|E_{\hat{G}\bar{\theta}}(s_j\to) - E_{\tilde{G}\bar{\theta}}(s_j\to)|$$

$$\le (H+1)5\sqrt{2d}H^2L_1\xi/\varepsilon,$$

where the first inequality sums over the contributions of $\tau$, $D$, and $E^\to$, and the second applies induction. By combining this bound with the bounds on $\|v - a\|_2$ and $\|w - b\|_2$, and that $E(\cdot) \in [-2H, 2H]$ (Eq. (31)) implying that $\bar{F}(\cdot) \in [-2H, 2H]$, for all $s \in \mathcal{S} \setminus \mathcal{S}_1$, we have that

$$\left|v_\|^\top \bar{F}_{\hat{G}\bar{\theta}}(s)w - a_\|^\top \bar{F}_{\tilde{G}\bar{\theta}}b\right|(s) \le 6H\xi + (H+1)5\sqrt{2d}H^2L_1\xi/\varepsilon$$

$$\le 5\sqrt{2d}(H+1)^3L_1\xi/\varepsilon = \min\{\varepsilon/(dH^2\omega), 1/\sqrt{m'_{\max}nH}\} - \eta_0 \qquad (37)$$

Take any $m' \in [m'_{\max}]$ (this includes the entire execution of SKIPPYELEANOR). and let the quantities of Section 4.3 (such as $F$) be calculated with the value of $Q$ at the beginning iteration $m'$

(Line 5). Take any $t \in [H-1]$, $i \in [t+1:H]$. Take any $y \in C_{\xi}^{\times}$ and assign values to $a, b, \tilde{G}$, and $\tilde{\theta}$ based on $y$ as above. For any $lkj \in \mathbf{I}^m(t)$, observe that given all the history of SKIPPYELEANOR interacting with the MDP up to (and including) $S_t^{lkj}, A_t^{lkj}$, the trajectory $S_{t+1}^{lkj}, A_{t+1}^{lkj}, \ldots, R_H^{lkj}$ is an independent rollout with policy $\pi^0$, with its law given by $\mathcal{P}_{\pi^0, S_t^{lkj}, A_t^{lkj}}$. The random variable $a_{\parallel}^{\top} F_{\tilde{G}\tilde{\theta}}(S_i^{lkj} \ldots, R_H^{lkj})b$ has range $[-2H, 2H]$ and expectation (conditioned on this history) $\mathbb{E}_{\pi^0, S_t^{lkj}, A_t^{lkj}} a_{\parallel}^{\top} \bar{F}_{\tilde{G}\tilde{\theta}}(S_i)b$. Let $\check{\theta}_{ti}$ be $\tilde{\theta}_{ti}$ from Corollary 4.11, satisfying $\left\| \check{\theta}_{ti} \right\|_2 \le 1/\sqrt{\lambda}$ and Eq. (11) for $a_{\parallel}, b, \tilde{G}$, and $\tilde{\theta}$ instead of $v_{\parallel}, w, \hat{G}$, and $\bar{\theta}$:

$$\mathbb{E}_{\pi^0, s, a} a_{\parallel}^{\top} \bar{F}_{\tilde{G}\tilde{\theta}}(S_i)b \approx_{\eta_0} \left\langle \varphi(s,a), \check{\theta}_{ti} \right\rangle . \tag{38}$$

Take the sequence $A$ formed of $\varphi_t^{lkj}$ (for $lkj \in \mathbf{I}^m(t)$, in the order that these random variables are observed), and the sequence $X$ formed of $v_{\parallel} F_{\hat{G}\bar{\theta}}(S_i^{lkj}, \ldots, R_H^{lkj})w$ (for $lkj \in \mathbf{I}^m(t)$, in the same order), and the sequence $\Delta$ formed of $\mathbb{E}_{\pi^0, S_t^{lkj}, A_t^{lkj}} v_{\parallel} \bar{F}_{\hat{G}\bar{\theta}}(S_i)w - \left\langle \varphi_t^{lkj}, \check{\theta}_{ti} \right\rangle$ (for $lkj \in \mathbf{I}^m(t)$, in the same order, for any $v, w, \hat{G}$, and $\bar{\theta}$ as in the statement of this lemma). Then the sequences $A$, $X$, and $\Delta$ satisfy the conditions of Lemma M.4 with a subgaussianity parameter $\sigma = 2H$. Due to this lemma, with probability at least $1 - \zeta/(m_{\max}'H^2|C_{\xi}^{\times}|)$, for any choice of $v, w, \hat{G}$, and $\bar{\theta}$ (as above),

$$\left\| \tilde{\theta}_{ti} - \check{\theta}_{ti} \right\|_{X_{mt}} < \sqrt{\lambda} \left\| \check{\theta}_{ti} \right\|_2 + \|\Delta\|_{\infty} \sqrt{|\mathbf{I}^m(t)|} + 2H \sqrt{2 \log \left( \frac{m_{\max}'H^2|C_{\xi}^{\times}|}{\zeta} \right) + \log \left( \frac{\det X_{mt}}{\lambda^d} \right)} \tag{39}$$

$$\text{where} \qquad \tilde{\theta}_{ti} = X_{mt}^{-1} \sum_{lkj \in \mathbf{I}^m(t)} \varphi_t^{lkj} v_{\parallel}^{\top} F_{\hat{G}\bar{\theta}}(S_i^{lkj}, \ldots, R_H^{lkj})w$$

A union bound over all $m' \in [m_{\max}']$, $t, i$, and $y \in C_{\xi}^{\times}$ guarantees with probability at least $1 - \zeta$, the above holds for all choice of these variables, any time beginning of any iteration (Line 5) is executed. Note that we need the union bound over $m$ because the value of $Q$ underlying the targets of least-squares estimations can potentially change between iterations.

To finish the proof, under this high-probability event, take any $m, t, i, \hat{G}$, and $\bar{\theta}$ as in the statement of this lemma, and choose $y \in C_{\xi}^{\times}$ as before, to satisfy Eq. (37). Combined with Eq. (38), this immediately implies that the sequence $\Delta$ formed of quantities with absolute value

$$\left| \mathbb{E}_{\pi^0, S_t^{lkj}, A_t^{lkj}} v_{\parallel} \bar{F}_{\hat{G}\bar{\theta}}(S_i)w - \left\langle \varphi_t^{lkj}, \check{\theta}_{ti} \right\rangle \right|$$

$$\le \left| \mathbb{E}_{\pi^0, S_t^{lkj}, A_t^{lkj}} v_{\parallel} \bar{F}_{\hat{G}\bar{\theta}}(S_i)w - a_{\parallel} \bar{F}_{\tilde{G}\tilde{\theta}}(S_i)b \right| + \left| a_{\parallel} \bar{F}_{\tilde{G}\tilde{\theta}}(S_i)b - \left\langle \varphi_t^{lkj}, \check{\theta}_{ti} \right\rangle \right| \tag{40}$$

$$\le \min\{\varepsilon/(dH^2\omega), 1/\sqrt{m_{\max}'nH}\} - \eta_0 + \eta_0$$

satisfies $\|\Delta\|_{\infty} \le \min\{\varepsilon/(dH^2\omega), 1/\sqrt{m_{\max}'nH}\}$. Take any $(s,a) \in \mathcal{S}_t \times [\mathcal{A}]$, and let $\tilde{\theta}_{ti}$ and $\check{\theta}_{ti}$ be as above (in Eq. (39)) for $v_{\parallel}, w, \hat{G}$, and $\bar{\theta}$. Note that

$$v_{\parallel}^{\top} \varphi(s,a)^{\top} \hat{\theta}_{\hat{G}\bar{\theta}}^{ti} w = \left\langle \varphi(s,a), \tilde{\theta}_{ti} \right\rangle ,$$

By the triangle inequality, using Cauchy-Schwarz, and Eqs. (39) and (40),

$$
\left| v_{\parallel}^{\top} \left( \varphi(s,a)^{\top} \hat{\theta}_{\hat{G}\bar{\theta}}^{ti} - \underset{\pi^0,s,a}{\mathbb{E}} \bar{F}_{\hat{G}\bar{\theta}}(S_i) \right) w \right|
$$

$$
\leq \left| \langle \varphi(s,a), \tilde{\theta}_{ti} - \check{\theta}_{ti} \rangle \right| + \left| \underset{\pi^0,s,a}{\mathbb{E}} v_{\parallel}^{\top} \bar{F}_{\hat{G}\bar{\theta}}(S_i) w - \langle\langle \varphi(s,a), \check{\theta}_{ti} \rangle\rangle \right|
$$

$$
\leq \|\varphi(s,a)\|_{X_{mt}^{-1}} \left( \sqrt{\lambda} \left\| \check{\theta}_{ti} \right\|_2 + \frac{\sqrt{|\mathbf{I}^m(t)|}}{\sqrt{m'_{\max}nH}} + 2H \sqrt{2\log\left(\frac{m'_{\max}H^2 |C_{\xi}^{\times}|}{\zeta}\right) + \log\left(\frac{\det X_{mt}}{\lambda^d}\right)} \right) + \frac{\varepsilon}{dH^2\omega}
$$

$$
\leq \|\varphi(s,a)\|_{X_{mt}^{-1}} \left( 2 + 2H \sqrt{2dH(d_0+1) \log \frac{12 d_0 HL_2}{\alpha\xi} + 2\log \frac{m'_{\max}H^2}{\zeta} + d\log\left(\lambda + m'_{\max}nHL_1^2/d\right)} \right) + \frac{\varepsilon}{dH^2\omega}
$$

$$
\leq \|\varphi(s,a)\|_{X_{mt}^{-1}} \beta + \frac{\varepsilon}{dH^2\omega} \,,
$$

(41)

where in the fourth line we used that $|\mathbf{I}^m(t)| \leq m'_{\max}nH$, $|C_{\xi}^{\times}| \leq (12 d_0 HL_2/(\alpha\xi))^{dH(d_0+1)}$, and we used the inequality of arithmetic and geometric means to bound $\det X_{mt} \leq \left(\frac{1}{d} \operatorname{Tr} X_{mt}\right)^d \leq \left(\frac{\operatorname{Tr} \lambda I + |\mathbf{I}^m(t)| L_1^2}{d}\right)^d$. $\qquad\square$

***Proof of Lemma E.3.*** Choose $n$ to satisfy

$$
n = \left\lceil 64 \frac{(dH^2\omega)^2}{\varepsilon^2} H^2 \left( 2d \log \frac{18 dH^3}{\varepsilon} + \log \frac{2m'_{\max}H^2}{\zeta} \right) \right\rceil .
$$

(42)

This leads to $n = \tilde{O}(d^5 H^6/\varepsilon^2)$.

Similarly to the proof of Lemma E.2, we start with a covering argument. This time, as $\hat{G}$ and $\bar{\theta}$ are fixed, we only consider $v$ and $w$, to cover $v_{\parallel}^{\top} \bar{F}_{t'}^{(j)} w$ and $v_{\parallel} \hat{F}_{t'}^{(j)} w$. Let $\xi' = \frac{\varepsilon}{12 dH^3}$. There is a set $C_{\xi'}^{+} \subset \mathcal{B}(1) \times \mathcal{B}(1)$ with $|C_{\xi'}| \leq (3/\xi')^{2d}$ such that for all $v,w \in \mathcal{B}(1)$, there exists an $(a,b) \in C_{\xi'}^{+}$ with $\|v - a\|_2 \leq \xi'$ (and therefore $\left\| v_{\parallel} - a_{\parallel} \right\|_2 \leq \xi'$), and $\|w - b\|_2 \leq \xi'$. Take such a choice of $(a,b)$ for any $(v,w)$. As $E(\cdot) \in [-2H, 2H]$ by Eq. (31), and $\left\| \bar{\varphi}_Q(\cdot) \right\|_2 \leq 1$, For $i \in [2:H]$ and any input,

$$
\left| v_{\parallel}^{\top} F(s_i\rightarrow)w - a_{\parallel}^{\top} F(s_i\rightarrow)b \right| \leq 6H\xi' = \frac{\varepsilon}{2dH^2} \,,
$$

and therefore for any $s \in \mathcal{S} \setminus \mathcal{S}_1$, $\left| v_{\parallel}^{\top} \bar{F}(s)w - a_{\parallel}^{\top} \bar{F}(s)b \right| \leq \varepsilon/(2dH^2)$. For $j \in [n]$ let

$$
\tilde{F}_j^{ki} = \underset{\pi^0, S_{\mathrm{p}(k)}^{mkj}, A_{\mathrm{p}(k)}^{mkj}}{\mathbb{E}} F_{\hat{G}\bar{\theta}}(S_i^{mkj}, \ldots, R_H^{mkj}) = \underset{\pi^0, S_{\mathrm{p}(k)}^{mkj}, A_{\mathrm{p}(k)}^{mkj}}{\mathbb{E}} \bar{F}_{\hat{G}\bar{\theta}}(S_i^{mkj})
$$

By the triangle inequality, for any $k \in [H-1]$, $i \in [k+1:H]$,

$$
\left| v_{\parallel}^{\top} \left( y_{\hat{G}\bar{\theta}}^{ki} - \hat{F}_{\hat{G}\bar{\theta}}^{ki} \right) w \right|
$$

$$
\leq \left| \frac{1}{n} \sum_{j\in[n]} c_{ki}^{j} v^{\top} \left( \varphi_{\mathrm{p}(k)}^{mkj\top} \hat{\theta}_{\hat{G}\bar{\theta}}^{\mathrm{p}(mkj),i} - \tilde{F}_j^{ki} \right) w \right| + \left| \frac{1}{n} \sum_{j\in[n]} c_{ki}^{j} v^{\top} \left( \tilde{F}_j^{ki} - F_{\hat{G}\bar{\theta}}(S_i^{mkj}, \ldots, R_H^{mkj}) \right) w \right|
$$

$$
\leq \frac{1}{n} \sum_{j\in[n]} c_{ki}^{j} \left\| \varphi_{\mathrm{p}(k)}^{mkj} \right\|_{X_{m,\mathrm{p}(mkj)}^{-1}} \beta + \frac{\varepsilon}{dH^2\omega} + \frac{\varepsilon}{dH^2\omega} + \left| \frac{1}{n} \sum_{j\in[n]} c_{ki}^{j} a^{\top} \left( \tilde{F}_j^{ki} - F_{\hat{G}\bar{\theta}}(S_i^{mkj}, \ldots, R_H^{mkj}) \right) b \right| \,,
$$

(43)

where the second inequality uses Lemma E.2 and applies the triangle inequality twice again. Observe that for all $j \in [n]$, given all the history of SKIPPYELEANOR interacting with the MDP

up to (and including) $S_{p(k)}^{mkj}, A_{p(k)}^{mkj}$ (which also includes the value of $c_{ki}^j$ for $i \in [H+1]$), the trajectory $S_{p(k)+1}^{mkj}, A_{p(k)+1}^{mkj}, \ldots, R_H^{mkj}$ is an independent rollout with policy $\pi^0$, with its law given by $\mathcal{P}_{\pi^0, S_{p(k)}^{mkj}, A_{p(k)}^{mkj}}$. Therefore, for any fixed $(a, b) \in C_{\xi'}^+$, $c_{ki}^j a^\top \left( \tilde{F}_j^{ki} - F_{\hat{G}\bar{\theta}}(S_i^{mkj}, \ldots, R_H^{mkj}) \right) b$ are independent zero-mean random variables with range $[-4H, 4H]$. Applying Hoeffding's inequality with a union bound over $m', k, i, a$, and $b$, with probability at least $1 - \zeta$, for any of the $m' \in [m'_{\max}]$ times the beginning of the iteration (Line 5) is executed (this includes the entire execution of SKIP-PYELEANOR),

$$\left| \frac{1}{n} \sum_{j \in [n]} c_{ki}^j a^\top \left( \tilde{F}_j^{ki} - F_{\hat{G}\bar{\theta}}(S_i^{mkj}, \ldots, R_H^{mkj}) \right) b \right| \leq \frac{8H}{\sqrt{n}} \sqrt{\log \frac{2m'_{\max}H^2 |C_{\xi'}^+|}{\zeta}}$$

$$= \frac{8H}{\sqrt{n}} \sqrt{2d \log \frac{18dH^3}{\varepsilon} + \log \frac{2m'_{\max}H^2}{\zeta}} \leq \frac{\varepsilon}{dH^2\omega} ,$$

where we used Eq. (42). To finish, note that unless $c_{ki}^j = 0$, $\left\| \varphi_{p(k)}^{mkj} \right\|_{X_{m,p(mkj)}^{-1}} < 2(\beta\omega dH)^{-1}$, so we can continue from Eq. (43) by bounding the average feature-norm by $\bar{\sigma}_k^m$ as

$$\left| v_\parallel^\top \left( y_{\hat{G}\bar{\theta}}^{ki} - \hat{F}_{\hat{G}\bar{\theta}}^{ki} \right) w \right| \leq \bar{\sigma}_k^m \beta + 3 \frac{\varepsilon}{dH^2\omega} . \qquad \square$$

***Proof of Lemma E.4.*** Recall that $(k, i, v)$ are the arguments and $x$ the value of Optimization Problem 4.12. Throughout the proof we write $Q$ to refer to its value *just before* Line 13 is executed. We write $\bullet_\parallel$ for $\bullet_{\parallel(Q,i)}$, and $\bullet_\perp$ for $\bullet_{\perp(Q,i)}$. Let $\mathbf{M} = y_{\hat{G}\bar{\theta}}^{ki} - \hat{F}_{\hat{G}\bar{\theta}}^{ki}$. Therefore, $v^\top \mathbf{M} v = x > \bar{\sigma}_k^m \beta\omega + 3 \frac{\varepsilon}{dH^2}$, and by Lemma E.1, $w^\top \mathbf{M} w > \bar{\sigma}_k^m \beta\omega + 2 \frac{\varepsilon}{dH^2}$.

Line 13 changes $Q_i$ by appending $Q_i^{-1}w$ to the sequence $C_i$ of vectors from which $Q$ is calculated according to Eq. (4). Eq. (5) lists the conditions on the new sequence $C_i$ that need to be satisfied for $Q$ to stay a valid preconditioning. Consider the third condition, i.e., $\left\| Q_i^{-1}w \right\|_2 \leq L_3$. Observe that $Q_i^{-1} \text{Proj}_{Z(Q,i)} \preceq L_3^2 I$ and $\|v\|_2 = 1$, therefore $\left\| Q_i^{-1}w \right\|_2 = \left\| Q_i^{-1} \text{Proj}_{Z(Q,i)} v \right\|_2 \leq L_3$.

Now consider the second condition. To prove that it holds, we need to show that $\left\| Q_i Q_i^{-1}w \right\|_2 = \|w\|_2 \geq \frac{1}{2}$. Let $x = \|w\|_2^{-1}$. Since $v$ was the argument of the optimization problem, and using Lemma E.1,

$$x^2 w^\top \mathbf{M} w \leq v^\top \mathbf{M} v \leq w^\top \mathbf{M} w + \frac{\varepsilon}{dH^2\omega} \leq w^\top \mathbf{M} w(1 + 1/2)$$

Therefore, $\|w\|_2^2 \geq \frac{2}{3}$. We immediately get that

$$\left\| Q_i Q_i^{-1}w \right\|_2^2 \geq \frac{2}{3} ,$$

satisfying the second condition.

It remains to prove that the first condition also holds. First, noting that $\mathbf{M}$ is symmetric, we can decompose $w^\top \mathbf{M} w$ as

$$w^\top \mathbf{M} w = w_\parallel^\top \mathbf{M} w + w_\parallel^\top \mathbf{M} w_\perp + w_\perp^\top \mathbf{M} w_\perp .$$

Applying Lemma E.3 on the first two terms,

$$w^\top \mathbf{M} w \leq 2\bar{\sigma}_k^m \beta + 6 \frac{\varepsilon}{dH^2\omega} + w_\perp^\top \mathbf{M} w_\perp .$$

Due to $\omega > 3$ and $w^\top \mathbf{M} w > \bar{\sigma}_k^m \beta\omega + 2 \frac{\varepsilon}{dH^2}$ and the above, $w_\perp \neq \mathbf{0}$. Let $w' = w_\perp / \|w_\perp\|_2$. Since $v$ was the argument of the optimization problem, have that $v^\top \mathbf{M} v \geq w'^\top \mathbf{M} w'$. Putting this together,

$$\|w_\perp\|_2^{-2} w_\perp^\top \mathbf{M} w_\perp = w'^\top \mathbf{M} w' \leq v^\top \mathbf{M} v \leq w^\top \mathbf{M} w + \frac{\varepsilon}{dH^2\omega} \leq 2\bar{\sigma}_k^m \beta + 7 \frac{\varepsilon}{dH^2\omega} + w_\perp^\top \mathbf{M} w_\perp ,$$

Since $v^\top \mathbf{M} v > \bar\sigma_k^m \beta\omega + 3\frac{\varepsilon}{dH^2}$, $w_\perp^\top \mathbf{M} w_\perp \geq (\omega - 7/3)\left(\bar\sigma_k^m \beta + 3\frac{\varepsilon}{dH^2\omega}\right) > 0$ and therefore dividing the above by $w_\perp^\top \mathbf{M} w_\perp$,

$$\|w_\perp\|_2^{-2} \leq \frac{7/3}{\omega - 7/3} + 1$$

$$\|w_\perp\|_2^2 \geq \frac{1}{1+c} \qquad \text{for } c = \frac{7/3}{\omega - 7/3}$$

$$\|w_\parallel\|_2^2 \leq 1 - \frac{1}{1+c} \qquad \text{as } \|w\|_2 \leq 1.$$

Now to prove that the first condition also holds,

$$\sup_{\theta \in \Theta_i} \left|\langle \theta, Q_i^{-1} w\rangle\right| = \sup_{\theta \in \Theta_i^Q} |\langle \theta, w\rangle| \leq \sup_{\theta \in \Theta_i^Q} \|\theta\|_2 \|w_\parallel\|_2 + \sup_{\theta \in \Theta_i^Q} |\langle \theta, w_\perp\rangle|$$

$$\leq \sqrt{d_1 + 1}\sqrt{1 - \frac{1}{1+c}} + \sup_{\theta \in \Theta_i^Q} \|\theta\|_{V(G_h^Q, \rho_h^Q)^\dagger} \|w_\perp\|_{V(G_h^Q, \rho_h^Q)}$$

$$\leq \sqrt{d_1 + 1}\sqrt{1 - \frac{1}{1+c}} + \sqrt{2d w_\perp^\top (\gamma I) w_\perp}$$

$$\leq \sqrt{d_1 + 1}\sqrt{1 - \frac{1}{1+c}} + \sqrt{2d\gamma} = \sqrt{d_1 + 1}\sqrt{1 - \frac{1}{1+c}} + \frac{1}{2},$$

where in the second line we used Lemma 4.3 to bound $\sup_{\theta \in \Theta_i^Q} \|\theta\|_2$, and for the second term we used Eq. (26) with Cauchy-Schwarz. In the third line we used Eq. (24), and the definition of $\text{Proj}_\perp$. Finally in the last line we use that $w_\perp$ is perpendicular to $a_i$ for $i \leq d'$ (by definition) and that $\lambda_i \leq \gamma$ for $i > d'$. It is left to prove that $\sqrt{d_1 + 1}\sqrt{1 - \frac{1}{1+c}} \leq \frac{1}{2}$. This holds if $c \geq 1/(4(d_1 + 1) - 1)$, which is satisfied as $c = 1/(3(d_1 + 1))$, due to $\omega = 7(d_1 + 1) + 7/3$ (Eq. (13)). □

# I  Deferred proofs for Appendix E.2

***Proof of Lemma E.5.*** The features $\varphi_{\text{p}(k)}^{lkj}$ are observed by SKIPPYELEANOR in the order of increasing $l$, within that increasing $k$, and within that, increasing $j$. Each time the next $\varphi_{\text{p}(k)}^{lkj}$ is observed, we sum the elliptic potential as follows.

For $i \in [m], r \in [H], u \in [n], t \in [H]$, let the set of indices observed before $\varphi_{\text{p}(r)}^{iru}$ whose Phase II (rollout phase) starts at some stage $t$ be:

$$\mathbf{I}^{iru}(t) = \{l \in [i], k \in [H], j \in [n] : lHn + kn + j < iHn + rn + u \text{ and } \text{p}(lkj) = t\}$$

Let a version of this where only the whole iteration $i$'s data is included be

$$\mathbf{J}^i(t) = \{l = i, k \in [H], j \in [n] : \text{p}(lkj) = t\}$$

Let

$$X_{iru}(t) = \lambda I + \sum_{lkj \in \mathbf{I}^{iru}(t)} \varphi_{\text{p}(k)}^{lkj} \varphi_{\text{p}(k)}^{lkj}{}^\top$$

Observe that $X_{it}$, defined in Optimization Problem 4.10, is the version of this that only updates at the start of each iteration $i$, that is,

$$X_{it} = X_{i11}(t).$$

The total elliptic potential, observed by the end of iteration $m$ is, writing $k = \text{p}(iru)$ on the left hand side:

$$\sum_{i \in [m], r \in [H], u \in [n]} \mathbb{1}\{k < H + 1\} \min\left\{1, \left\|\varphi_k^{iru}\right\|_{X_{iru}(k)^{-1}}^2\right\} = \sum_{i \in [m], t \in [H]} \sum_{lkj \in \mathbf{J}^i(t)} \min\left\{1, \left\|\varphi_t^{lkj}\right\|_{X_{lkj}(t)^{-1}}^2\right\}.$$

Applying the elliptical potential lemma (Lemma L.1) $H$ times for $t \in [H]$, this can be bounded as

$$\sum_{t \in [H], i \in [m]} \sum_{lkj \in \mathbf{J}^i(t)} \min\left\{1, \left\|\varphi_t^{lkj}\right\|_{X_{lkj}(t)^{-1}}^2\right\} \leq 2dH \log\left(1 + \frac{HmnL_1^2}{d\lambda}\right)$$

On the other hand, by Lemma L.2, then switching to an $\ell_1$-bound, then observing that by definition, $\sum \bar{\sigma}_k^i$ sums the same quantities but caps them by some threshold,

$$
\sum_{t \in [H], i \in [m]} \sum_{lkj \in \mathbf{J}^i(t)} \min \left\{ 1, \left\| \varphi_t^{lkj} \right\|_{X_{lkj}(t)^{-1}}^2 \right\} \geq \sum_{t \in [H], i \in [m]} \min \left\{ 1, \frac{1}{2} \sum_{lkj \in \mathbf{J}^i(t)} \left\| \varphi_t^{lkj} \right\|_{X_{it}^{-1}}^2 \right\}
$$

$$
\geq \sum_{i \in [m]} \min \left\{ 1, \frac{1}{2} \sum_{t \in [H]} \sum_{lkj \in \mathbf{J}^i(t)} \left\| \varphi_t^{lkj} \right\|_{X_{it}^{-1}}^2 \right\}
$$

$$
\geq \sum_{i \in [m]} \min \left\{ 1, \frac{1}{2Hn} \left( \sum_{t \in [H]} \sum_{lkj \in \mathbf{J}^i(t)} \left\| \varphi_t^{lkj} \right\|_{X_{it}^{-1}} \right)^2 \right\}
$$

$$
\geq \sum_{i \in [m]} \min \left\{ 1, \frac{1}{2Hn} \left( n \sum_{k \in [H]} \bar{\sigma}_k^i \right)^2 \right\}
$$

Whenever an iteration finishes without returning in Line 17, $\sum_{k \in [H]} \bar{\sigma}_k^m > \varepsilon / (dH^2 \beta \omega)$. Therefore,

$$
2dH \log \left( 1 + \frac{HmnL_1^2}{d\lambda} \right) \geq \sum_{i \in [m]} \min \left\{ 1, \frac{1}{2Hn} \left( n \sum_{k \in [H]} \bar{\sigma}_k^i \right)^2 \right\}
$$

$$
\geq \sum_{i \in [m]} \min \left\{ 1, \frac{1}{2H} n \left( \frac{\varepsilon}{dH^2 \beta \omega} \right)^2 \right\}
$$

$$
\geq \sum_{i \in [m]} \min \left\{ 1, Hd/\beta^2 \right\} = mHd/\beta^2 ,
$$

Therefore, even for the iteration that returns in Line 17,

$$
m \leq \beta^2 \log \left( 1 + \frac{HmnL_1^2}{d\lambda} \right) + 1 = m_{\max} . \qquad \square
$$

## J   Deferred proofs for Appendix E.3

***Proof of Lemma E.8.*** For notational simplicity we drop the subscripts $(\hat{G}, \bar{\theta})$. We first use the usual high-probability bounds on the least squares predictor and Hoeffding's inequality on the empirical mean quantities, to prove that with probability at least $1 - 3\zeta$, during the execution of SKIPPYE-LEANOR whenever Line 16 is executed, for all $k \in [H]$,

$$
\mathbb{E}_{\pi^{mk}, s_1} \tilde{c}_{k,H+1} C(S_{p(k)}) \leq \mathbb{E}_{\pi^{mk}, s_1} \sum_{u=p(k)}^{H} R_u + \tilde{c}_{k,H+1} E^{\rightarrow}(S_{p(k)+1}, \dots, R_H) + 2\bar{\sigma}_k^m \beta \omega dH + 4\frac{\varepsilon}{H} .
$$

$$
(44)
$$

The proof of this is presented as Lemma J.1.

Next, to prove the statement for $k \in [H]$, assume by induction that Eq. (22) holds for $i \in [k+1 : H]$.

Observe that SKIPPYPOLICY performs a rollout with policy $\pi^0$ for the rest of the episode starting from stage $p(k) + 1$, that is, $1 = A_{p(k)+1} = \cdots = A_H$. Therefore, the law of the random variables $S_{p(k)+1}, \dots, R_H$, given $(S_{p(k)}, A_{p(k)})$ is fully determined by the dynamics of the MDP, and is independent of the values of $p(k+1), \dots, p(H)$. Therefore,

$$
\mathbb{E}_{\pi^{mk}, s_1} \tilde{c}_{k+1,H+1} C(S_{p(k+1)}) = \mathbb{E}_{\pi^{mk}, s_1} \tilde{c}_{k+1,H+1} D(S_{p(k+1)}, \dots, R_H) + \sum_{u=p(k+1)}^{H} R_u
$$

$$
= \mathbb{E}_{\pi^{mk}, s_1} \tilde{c}_{k,H+1} E^{\rightarrow}(S_{p(k)+1}, \dots, R_H) + \sum_{u=p(k+1)}^{H} R_u ,
$$

$$
(45)
$$

where we use Eq. (9), and that $\pi^{mk}$ (SKIPPYPOLICY) is in phase II after stage $p(k)$, but defines the the mapping $p(\cdot)$ independently of whether the policy is in phase I or phase II, in such a way that for any $H \geq j > p(k)$,

$$
\mathop{\mathcal{P}}_{\pi^{mk},s_1} \left[ p(k+1) = j \mid p(k), S_{p(k)}, A_{p(k)} \right] = \mathop{\mathcal{P}}_{\pi^{mk},s_1} \left[ \tau(S_j) \prod_{j'=p(k)+1}^{j-1} (1 - \tau(S_{j'})) \mid p(k), S_{p(k)}, A_{p(k)} \right] .
$$

Combining Eq. (45) with Eq. (44),

$$
\mathop{\mathbb{E}}_{\pi^{mk},s_1} \tilde{c}_{k,H+1} C(S_{p(k)}) \leq \mathop{\mathbb{E}}_{\pi^{mk},s_1} \sum_{u=p(k)}^{p(k+1)-1} R_u + \tilde{c}_{k+1,H+1} C(S_{p(k+1)}) + 2\bar{\sigma}_k^m \beta \omega dH + 4\frac{\varepsilon}{H} .
$$

By Remark E.7, $\mathbb{E}_{\pi^{mk},s_1} \tilde{c}_{k+1,H+1} C(S_{p(k+1)}) = \mathbb{E}_{\pi^{m,k+1},s_1} \tilde{c}_{k+1,H+1} C(S_{p(k+1)}) = \bar{C}^{k+1}$ . Therefore, combining with the inductive hypothesis,

$$
\mathop{\mathbb{E}}_{\pi^{mk},s_1} \tilde{c}_{k,H+1} C(S_{p(k)}) \leq \mathop{\mathbb{E}}_{\pi^{mk},s_1} \sum_{u=p(k)}^{p(k+1)-1} R_u + \bar{C}^{k+1} + 2\bar{\sigma}_k^m \beta \omega dH + 4\frac{\varepsilon}{H}
$$

$$
\leq \mathop{\mathbb{E}}_{\pi^{mk},s_1} \sum_{u=p(k)}^{p(k+1)-1} R_u + \mathop{\mathbb{E}}_{\pi^{mH},s_1} \sum_{u=p(k+1)}^{H} R_u + 2 \sum_{i=k}^{H} \bar{\sigma}_k^m \beta \omega dH + 4(H-k+1)\frac{\varepsilon}{H}
$$

$$
= \mathop{\mathbb{E}}_{\pi^{mH},s_1} \sum_{u=p(k)}^{H} R_u + 2 \sum_{i=k}^{H} \bar{\sigma}_k^m \beta \omega dH + 4(H-k+1)\frac{\varepsilon}{H}
$$

where the last equation uses Remark E.7 again, finishing the induction. □

**Lemma J.1.** *Adopt the notation of Lemma E.8. With probability at least $1-3\zeta$, during the execution of* SKIPPYELEANOR*, whenever Line 16 is executed, for all $k \in [H]$,*

$$
\mathop{\mathbb{E}}_{\pi^{mk},s_1} \tilde{c}_{k,H+1} C(S_{p(k)}) \leq \mathop{\mathbb{E}}_{\pi^{mk},s_1} \sum_{u=p(k)}^{H} R_u + \tilde{c}_{k,H+1} E^{\rightarrow}(S_{p(k)+1}, \ldots, R_H) + 2\bar{\sigma}_k^m \beta \omega dH + 4\frac{\varepsilon}{H} .
$$

*Proof.* We refer as $\hat{\theta}$ to the value of the argument of Optimization Problem 4.10 recorded in Line 5. For $k \in [H]$, recall the definition of $\bar{\sigma}_k^m$ (Eq. (18)), along with the fact that unless $c_{k,H+1}^j = 0$, $\left\| \varphi_{p(k)}^{mkj} \right\|_{X_{m,p(mkj)}^{-1}} < 2(\beta \omega dH)^{-1}$, we get a useful bound on the average norm of the features under consideration:

$$
\frac{1}{n} \sum_{j \in [n]} c_{k,H+1}^j \left\| \varphi_{p(k)}^{mkj} \right\|_{X_{m,p(mkj)}^{-1}} \leq \bar{\sigma}_k^m . \tag{46}
$$

If Line 16 is executed, the consistency check passed, and therefore for all $k \in [H-1], i \in [k+1 : H]$,

$$
\mathrm{Tr}\left( y^{ki} - \hat{F}^{ki} \right) \leq \bar{\sigma}_k^m \beta \omega d + 3\frac{\varepsilon}{H^2} \tag{47}
$$

For $t \in [H]$ let the least-squares predictor of rewards sums under the policy $\pi^0$ be

$$
\breve{\theta}^{t,H+1} = X_{mt}^{-1} \sum_{lkj \in \mathbf{I}^m(t)} \varphi_t^{lkj} \sum_{u=t}^{H} R_u^{lkj} .
$$

For $k \in [H]$ and $j \in [n]$ let us introduce the shorthand

$$
R_{k\rightarrow}^{mkj} = \sum_{u=p(mkj)}^{H} R_u^{mkj} ,
$$

and similarly when the trajectory is clear from context: $R_{k\to} = \sum_{u=p(k)}^{H} R_u$. For $k \in [H]$ let

$$\hat{E}^k = \frac{1}{n} \sum_{j \in [n]} c_{k,H+1}^j \left( E^{\to}(S_{p(k)+1}^{mkj}, \ldots, R_H^{mkj}) + R_{k\to}^{mkj} \right)$$

$$\hat{C}^k = \frac{1}{n} \sum_{j \in [n]} c_{k,H+1}^j C(S_{p(k)}^{mkj})$$

$$y^{k,H+1} = \frac{1}{n} \sum_{j \in [n]} c_{k,H+1}^j \left\langle \varphi_{p(k)}^{mkj}, \breve{\theta}^{p(mkj),H+1} \right\rangle$$

$$z^{k,H+1} = \frac{1}{n} \sum_{j \in [n]} c_{k,H+1}^j R_{k\to}^{mkj}$$

For $t \in [H-1]$, $i \in [t+1:H]$, along with $\breve{\theta}^{t,H+1}$, let

$$\breve{\theta}^{ti} = X_{mt}^{-1} \sum_{lkj \in \mathbf{I}^m(t)} \varphi_t^{lkj} \operatorname{Tr}(F(S_i^{lkj}, \ldots, R_H^{lkj})) = X_{mt}^{-1} \sum_{lkj \in \mathbf{I}^m(t)} \varphi_t^{lkj} E(S_i^{lkj}, \ldots, R_H^{lkj}),$$

where the second equality is by Eq. (32). Observe that for any $v \in \mathbb{R}^d$, $\operatorname{Tr}(v^{\top} \hat{\theta}^{ti}) = \left\langle v, \breve{\theta}^{ti} \right\rangle$. Therefore, for $k \in [H]$,

$$y^{k,H+1} + \sum_{i=k+1}^{H} \operatorname{Tr}(y^{ki}) = \frac{1}{n} \sum_{j \in [n]} \sum_{i=k+1}^{H+1} c_{ki}^j \left\langle \varphi_{p(k)}^{mkj}, \breve{\theta}^{p(mkj),i} \right\rangle = \frac{1}{n} \sum_{j \in [n]} c_{k,H+1}^j \left\langle \varphi_{p(k)}^{mkj}, \sum_{i=p(mkj)+1}^{H+1} \breve{\theta}^{p(mkj),i} \right\rangle$$

For any $t \in [H]$, by the definitions,

$$\sum_{i=t+1}^{H+1} \breve{\theta}^{ti} = X_{mt}^{-1} \sum_{lkj \in \mathbf{I}^m(t)} \varphi_t^{lkj} \left( \sum_{i=t+1}^{H} E(S_i^{lkj}, \ldots, R_H^{lkj}) + \sum_{u=t}^{H} R_u^{mkj} \right)$$

$$= X_{mt}^{-1} \sum_{lkj \in \mathbf{I}^m(t)} \varphi_t^{lkj} \left( E^{\to}(S_{t+1}^{lkj}, \ldots, R_H^{lkj}) + \sum_{u=t}^{H} R_u^{mkj} \right) = \hat{\theta}_t$$

Plugging this into the previous calculation,

$$
\begin{aligned}
y^{k,H+1} + \sum_{i=k+1}^{H} \operatorname{Tr}(y^{ki}) &= \frac{1}{n} \sum_{j \in [n]} c_{k,H+1}^j \left\langle \varphi_{p(k)}^{mkj}, \hat{\theta}_{p(mkj)} \right\rangle \\
&\geq \frac{1}{n} \sum_{j \in [n]} c_{k,H+1}^j \left\langle \varphi_{p(k)}^{mkj}, \bar{\theta}_{p(mkj)} \right\rangle \\
&\quad - \frac{1}{n} \sum_{j \in [n]} c_{k,H+1}^j \left\| \varphi_{p(k)}^{mkj} \right\|_{X_{m,p(mkj)}^{-1}} \left\| \bar{\theta}_{p(mkj)} - \hat{\theta}_{p(mkj)} \right\|_{X_{m,p(mkj)}} \\
&\geq \frac{1}{n} \sum_{j \in [n]} c_{k,H+1}^j \left\langle \varphi_{p(k)}^{mkj}, \bar{\theta}_{p(mkj)} \right\rangle - \bar{\sigma}_k^m \beta H \\
&\geq \frac{1}{n} \sum_{j \in [n]} c_{k,H+1}^j \operatorname{clip}_{[0,H]} \left\langle \varphi_{p(k)}^{mkj}, \bar{\theta}_{p(mkj)} \right\rangle - \bar{\sigma}_k^m \beta H \\
&= \frac{1}{n} \sum_{j \in [n]} c_{k,H+1}^j C(S_{p(k)}^{mkj}) - \bar{\sigma}_k^m \beta H = \hat{C}^k - \bar{\sigma}_k^m \beta H,
\end{aligned}
\tag{48}
$$

where the first inequality uses Cauchy-Schwarz. The second inequality bounds the average of the first norm by Eq. (46), and the bound on the second norm (for any $j$) is by definition of Optimization Problem 4.10. The third inequality relies on the fact that $c_{k,H+1}^j = 0$ if the clipped inner product is negative, and the final equality is due to the definition of $C$ along with the fact that $A_{p(k)}^{mkj} = \pi^+(S_{p(k)}^{mkj})$, as this is the last state in the trajectory where SKIPPYPOLICY takes the inner-product maximizing action ($\pi^+$) before rolling out with $\pi^0$.

By Eqs. (12) and (32), we have that

$$z^{k,H+1} + \sum_{i\in[k+1:H]} \text{Tr}(\hat{F}^{ki}) = \frac{1}{n}\sum_{j\in[n]} c^j_{k,H+1}\left(\sum_{i=\text{p}(mkj)+1}^{H+1} E(S_i^{mkj},\ldots,R_H^{mkj}) + R_{k\to}^{mkj}\right)$$
$$= \frac{1}{n}\sum_{j\in[n]} c^j_{k,H+1}\left(E^\to(S_{\text{p}(k)+1}^{mkj},\ldots,R_H^{mkj}) + R_{k\to}^{mkj}\right) = \hat{E}^k \tag{49}$$

Combining Eqs. (48) and (49),

$$\hat{C}^k - \hat{E}^k \le \bar{\sigma}_k^m \beta H + \left(y^{k,H+1} - z^{k,H+1}\right) + \sum_{i\in[k+1:H]} \text{Tr}(y^{ki} - \hat{F}^{ki})$$
$$\le \bar{\sigma}_k^m \beta H + \left(\left|\mathbb{E}_{\pi^{mk},s_1} c_{k,H+1} R_{k\to} - z^{k,H+1}\right| + \left|y^{k,H+1} - \mathbb{E}_{\pi^{mk},s_1} c_{k,H+1} R_{k\to}\right|\right) + \bar{\sigma}_k^m \beta\omega dH + 3\frac{\varepsilon}{H} \tag{50}$$

where the sum (last term) is bounded by Eq. (47), and we apply a triangle inequality on the second term. To continue bounding this term, we apply Hoeffding's inequality on the independent random variables $c^j_{k,H+1} R_{k\to}$ (for $j \in [n]$) that have range $[0, H]$, along with a union bound over the iteration $m' \in [m'_{\max}]$ and $k \in [H]$, to get that with probability at least $1 - \zeta$,

$$\left|\mathbb{E}_{\pi^{mk},s_1} c_{k,H+1} R_{k\to} - z^{k,H+1}\right| \le \frac{H}{\sqrt{n}}\sqrt{\log\frac{2m'_{\max}H}{\zeta}} \le \frac{\varepsilon}{dH^2\omega}. \tag{51}$$

The remaining term $\left|y^{k,H+1} - \mathbb{E}_{\pi^{mk},s_1} c_{k,H+1} R_{k\to}\right|$ is bounded using the realizability of $q^{\pi^0}$ (Definition 3.2) as follows. Take any $t \in [H]$. By definition there exists $\theta_t^\star \in \Theta_t^Q \subseteq \mathcal{B}(L_2)$, such that for all $s \in \mathcal{S}_t$ and $a \in [\mathcal{A}]$, $q^{\pi^0}(s,a) \approx_\eta \langle\varphi(s,a),\theta_t^\star\rangle$. Take the sequence $A$ formed of $\varphi_t^{lkj}$ (for $lkj \in \mathbf{I}^m(t)$, in the order that these random variables are observed), and the sequence $X$ formed of $R_{k\to}^{mkj}$ (for $lkj \in \mathbf{I}^m(t)$, in the same order), and the sequence $\Delta$ formed of $q^{\pi^0}(S_t^{lkj},A_t^{lkj}) - \langle\varphi_t^{lkj},\theta_t^\star\rangle$ (for $lkj \in \mathbf{I}^m(t)$, in the same order). Then the sequences $A$, $X$, and $\Delta$ satisfy the conditions of Lemma M.4 with a subgaussianity parameter $\sigma = H$. Due to this lemma, applied with a union bound over $m' \in [m'_{\max}]$ and $t \in [H]$, with probability at least $1 - \zeta$,

$$\left\|\breve{\theta}^{t,H+1} - \theta_t^\star\right\|_{X_{mt}} < \sqrt{\lambda}\left\|\theta_t^\star\right\|_2 + \|\Delta\|_\infty\sqrt{|\mathbf{I}^m(t)|} + H\sqrt{2\log\left(\frac{m'_{\max}H}{\zeta}\right) + \log\left(\frac{\det X_{mt}}{\lambda^d}\right)}$$
$$\le 2 + H\sqrt{2\log\frac{m'_{\max}H}{\zeta} + \log\left(\frac{\det X_{mt}}{\lambda^d}\right)} \le \beta,$$

by Eq. (41). Therefore by Cauchy-Schwarz and Eq. (46),

$$\left|y^{k,H+1} - \mathbb{E}_{\pi^{mk},s_1} c_{k,H+1} R_{k\to}\right| \le \frac{1}{n}\sum_{j\in[n]} c^j_{k,H+1}\left(\left\|\varphi_{\text{p}(k)}^{mkj}\right\|_{X_{m,\text{p}(mkj)}^{-1}}\left\|\breve{\theta}^{\text{p}(mkj),H+1} - \theta_{\text{p}(mkj)}^\star\right\|_{X_{mt}} + \eta\right)$$
$$\le \bar{\sigma}_k^m\beta + \eta \le \bar{\sigma}_k^m\beta + \frac{\varepsilon}{dH^2\omega}.$$

Combining this with Eqs. (50) and (51),

$$\hat{C}^k - \hat{E}^k \le 1.5\bar{\sigma}_k^m\beta\omega dH + 3\frac{\varepsilon}{H} + 2\frac{\varepsilon}{dH^2\omega}. \tag{52}$$

We introduce the following notation for $j \in [n]$, $k \in [H+1]$, $i \in [H+1]$:

$$\bar{c}_{ki}^j = \mathbb{1}\left\{\text{p}(mkj) < i \text{ and } \left\|\varphi_{\text{p}(k)}^{mkj}\right\|_{X_{m,\text{p}(mkj)}^{-1}} \ge 2(\beta\omega dH)^{-1} \text{ and } \left\langle\varphi_{\text{p}(k)}^{mkj},\bar{\theta}_{\text{p}(mkj)}\right\rangle \ge 0\right\}$$
$$\hat{c}_{ki}^j = \mathbb{1}\left\{\text{p}(mkj) < i \text{ and } \left\langle\varphi_{\text{p}(k)}^{mkj},\bar{\theta}_{\text{p}(mkj)}\right\rangle < 0\right\},$$

such that for all $j$,

$$\tilde{c}_{ki}^j = c_{ki}^j + \bar{c}_{ki}^j + \hat{c}_{ki}^j . \tag{53}$$

Continuing from Eq. (52), as $E^{\rightarrow}(s_i\rightarrow) + \sum_{u=i}^H r_u \geq 0$ by Eq. (31), and if $\hat{c}_{k,H+1}^j = 1$ then $C(S_{p(k)}^{mkj}) = 0$, we have that

$$\frac{1}{n} \sum_{j \in [n]} (c_{k,H+1}^j + \hat{c}_{k,H+1}^j) \left( C(S_{p(k)}^{mkj}) - \left( E^{\rightarrow}(S_{p(k)+1}^{mkj}, \ldots, R_H^{mkj}) + R_{k\rightarrow}^{mkj} \right) \right) \leq 1.5\bar{\sigma}_k^m \beta \omega dH + 3\frac{\varepsilon}{H} + 2\frac{\varepsilon}{dH^2\omega} .$$

As (even if $\bar{c}_{k,H+1}^j = 1$) $C(S_{p(k)}^{mkj}) \leq H$,

$$\frac{1}{n} \sum_{j \in [n]} \bar{c}_{k,H+1}^j C(S_{p(k)}) \leq H\bar{\sigma}_k^m / (2(\beta\omega dH)^{-1}) = \frac{1}{2}\bar{\sigma}_k^m \beta \omega dH ,$$

which combined with the previous inequality and Eq. (53) yields

$$\frac{1}{n} \sum_{j \in [n]} \tilde{c}_{k,H+1}^j \left( C(S_{p(k)}^{mkj}) - \left( E^{\rightarrow}(S_{p(k)+1}^{mkj}, \ldots, R_H^{mkj}) + R_{k\rightarrow}^{mkj} \right) \right) \leq 2\bar{\sigma}_k^m \beta \omega dH + 3\frac{\varepsilon}{H} + 2\frac{\varepsilon}{dH^2\omega} .$$

Observe that the random variables $\tilde{c}_{k,H+1}^j \left( C(S_{p(k)}^{mkj}) - \left( E^{\rightarrow}(S_{p(k)+1}^{mkj}, \ldots, R_H^{mkj}) + R_{k\rightarrow}^{mkj} \right) \right)$ are independent (for $j \in [n]$) with range $[-2H, H]$ (Eq. (31)). By Hoeffding's inequality, with probability at least $1 - \zeta$, for all iteration $m' \in [m'_{\max}]$ (this includes the entire execution of SKIPPYELEANOR) and $k \in [H]$,

$$\left| \mathop{\mathbb{E}}_{\pi^{mk}, s_1} \tilde{c}_{k,H+1} \left( C(S_{p(k)}) - \left( E^{\rightarrow}(S_{p(k)+1}, \ldots, R_H) + R_{k\rightarrow} \right) \right) \right.$$

$$\left. - \frac{1}{n} \sum_{j \in [n]} \tilde{c}_{k,H+1}^j \left( C(S_{p(k)}^{mkj}) - \left( E^{\rightarrow}(S_{p(k)+1}^{mkj}, \ldots, R_H^{mkj}) + R_{k\rightarrow}^{mkj} \right) \right) \right|$$

$$\leq \frac{4H}{\sqrt{n}} \sqrt{\log \frac{2m'_{\max}H}{\zeta}} \leq \frac{\varepsilon}{dH^2\omega} .$$

Combining with the previous bound, under the intersection of the high-probability events referred to above, which by a union bound has a probability of at least $1 - 3\zeta$, we have that for all $k \in [H]$,

$$\mathop{\mathbb{E}}_{\pi^{mk}, s_1} \tilde{c}_{k,H+1} C(S_{p(k)}) \leq \mathop{\mathbb{E}}_{\pi^{mk}, s_1} \sum_{u=p(k)}^H R_u + \tilde{c}_{k,H+1} E^{\rightarrow}(S_{p(k)+1}, \ldots, R_H) + 2\bar{\sigma}_k^m \beta \omega dH + 4\frac{\varepsilon}{H} . \quad \square$$

## K   Deferred proofs for Appendix E.4

***Proof of Lemma E.10.*** Let $m$ be the current iteration. Unlike in previous lemmas, here we introduce $(\hat{G}, \bar{\theta})$ that does *not* refer to the outcome of Optimization Problem 4.10. Instead, let $\hat{G} = (\vartheta_h^i)_{h \in [2:H], i \in [d_0]} \in \mathbf{G}$ be the correct guess. For $h = H, \ldots, 1$, $\bar{\theta}_h$ is defined in sequence along with the behavior of a policy $\pi$ on stage $h$.

For $h = H, \ldots, 1$, assuming that this process already defined $\bar{\theta}_{h+1}, \ldots, \bar{\theta}_H$ (in Eq. (55)), let $\pi$ be the policy that, for any $t > h$ and $s \in \mathcal{S}_t$, takes action on $s$ as $\pi_{\hat{G}\bar{\theta}}^+(s)$ with probability $\tau_{\hat{G}\bar{\theta}}(s)$, and action 1 with probability $1 - \tau_{\hat{G}\bar{\theta}}(s)$ ($\tau$ is defined in Eq. (8)). Simultaneously, using the second part of Corollary 4.11, define $\tilde{\theta}_{hi} \in \mathcal{B}(4d_0L_2/\alpha)$ for $i \in [h+1:H]$ to satisfy for all $s \in \mathcal{S}_h, a \in [\mathcal{A}]$:

$$\mathop{\mathbb{E}}_{\pi^0, s, a} \text{Tr}(\bar{F}_{\hat{G}\bar{\theta}}(S_i)) \approx_{\eta_0} \langle \varphi(s, a), \tilde{\theta}_{hi} \rangle .$$

We also define $\tilde{\theta}_{h,H+1} \in \mathcal{B}(L_2)$ to satisfy for all $s \in \mathcal{S}_h, a \in [\mathcal{A}]$:

$$\mathop{\mathbb{E}}_{\pi^0, s, a} \sum_{u=h}^H R_u \approx_{\eta} \langle \varphi(s, a), \tilde{\theta}_{h,H+1} \rangle .$$

By Eq. (32),

$$\mathop{\mathbb{E}}_{\pi^0,s,a} \sum_{i\in[h+1:H]} \operatorname{Tr}(\bar{F}_{\hat{G}\bar\theta}(S_i)) + \sum_{u=h}^{H} R_u = \mathop{\mathbb{E}}_{\pi^0,s,a} E^{\rightarrow}_{\hat{G}\bar\theta}(S_{h+1},\ldots,R_H) + \sum_{u=h}^{H} R_u \approx_{H\eta_0} \left\langle \varphi(s,a),\bar\theta_h \right\rangle, \tag{54}$$

where we define

$$\bar\theta_h = \sum_{i\in[h+1:H+1]} \tilde\theta_{hi}. \tag{55}$$

We first show that $(\hat{G},\bar\theta)$ is feasible for Optimization Problem 4.10. Clearly, $\left\|\bar\theta_h\right\|_2 \le 4d_0 H L_2/\alpha$. For any $i\in[h+1:H]$, let

$$\hat\theta_{hi} = X_{mh}^{-1} \sum_{lkj\in\mathbf{I}^m(h)} \varphi_h^{lkj} \operatorname{Tr}(F_{\hat{G}\bar\theta}(S_{h+1}^{lkj},\ldots,R_H^{lkj})),$$

and let

$$\hat\theta_{h,H+1} = X_{mh}^{-1} \sum_{lkj\in\mathbf{I}^m(h)} \varphi_h^{lkj} \sum_{u=h}^{H} R_u^{lkj}.$$

Then, $\hat\theta$ of Optimization Problem 4.10 satisfies for all $h\in[H]$, by Eq. (32),

$$\hat\theta_h = \sum_{i\in[h+1:H+1]} \hat\theta_{hi}.$$

To show that $(\hat{G},\bar\theta)$ is feasible, it thus suffices to show for all $h\in[H]$, $i\in[h+1:H+1]$, that $\left\|\tilde\theta_{hi} - \hat\theta_{hi}\right\|_{X_{mh}} \le \beta$.

Fix any $h\in[H]$ and $i\in[h+1:H+1]$. Take the sequence $A$ formed of $\varphi_t^{lkj}$ (for $lkj\in\mathbf{I}^m(h)$, in the order that these random variables are observed). For $i<H+1$ take the sequence $X$ formed of $\operatorname{Tr}(F_{\hat{G}\bar\theta}(S_i^{lkj},\ldots,R_H^{lkj}))$ (for $lkj\in\mathbf{I}^m(h)$, in the same order), and the sequence $\Delta$ formed of $\mathbb{E}_{\pi^0,S_h^{lkj},A_h^{lkj}}\operatorname{Tr}(\bar{F}_{\hat{G}\bar\theta}(S_i)) - \left\langle\varphi_h^{lkj},\tilde\theta_{hi}\right\rangle$ (for $lkj\in\mathbf{I}^m(h)$, in the same order). For $i=H+1$, the sequence $X$ is formed of $\sum_{u=h}^{H} R_u^{lkj}$, and $\Delta$ is formed of $q^{\pi^0}(S_h^{lkj},A_h^{lkj}) - \left\langle\varphi_h^{lkj},\tilde\theta_{hi}\right\rangle$. Then the sequences $A$, $X$, and $\Delta$ satisfy the conditions of Lemma M.4 with a subgaussianity parameter $\sigma = H$. Due to this lemma, applied with a union bound over $m'\in[m'_{\max}]$, $t$, and $i$, with probability at least $1-\zeta$,

$$\left\|\hat\theta_{hi}-\tilde\theta_{hi}\right\|_{X_{mh}} < \sqrt{\lambda}\left\|\tilde\theta_{hi}\right\|_2 + \|\Delta\|_\infty\sqrt{|\mathbf{I}^m(t)|} + H\sqrt{2\log\left(\frac{m'_{\max}H^2}{\zeta}\right) + \log\left(\frac{\det X_{mt}}{\lambda^d}\right)}$$

$$\le 2 + H\sqrt{2\log\frac{m'_{\max}H^2}{\zeta} + \log\left(\frac{\det X_{mt}}{\lambda^d}\right)} \le \beta,$$

by Eq. (41).

Next, we show that the resulting policy $\pi$ is near-optimal. Assume by induction on $h=H,\ldots,1$, that for all $t\in[h+1:H]$, all $s\in\mathcal{S}_t$ and $a\in[\mathcal{A}]$,

$$v^\pi(s) \ge v^\star(s) - (H-t+1)(\varepsilon/H + 2H^2\eta_0) \quad\text{and} \tag{56}$$

$$\left\langle\varphi(s,a),\bar\theta_t\right\rangle \approx_{(H-t+1)H\eta_0} q^\pi(s,a). \tag{57}$$

To prove the above for $t=h$ as well, take any $s\in\mathcal{S}_h$, $a\in[\mathcal{A}]$. Introduce the random variable $P$ that, for a trajectory following $\mathcal{P}_{\pi^0,s,a}$, takes as its value the index of the first Bernoulli draw of 1 (starting from index $h+1$), when the Bernoullis have means $\tau_{\hat{G}\bar\theta}(S_j)$ for $j\in[h+1:H]$, and takes the value $H+1$ if all of these Bernoullis have outcome 0. Write $\mathbb{E}_{\pi^0,s,a,P}[\cdot]$ for $\mathbb{E}_{\pi^0,s,a}\mathbb{E}_P[\cdot\,|\,S_{h+1},\ldots,R_H]$.

Then,

$$\mathbb{E}_{\pi^0,s,a} E_{\hat{G}\bar{\theta}}^{\rightarrow}(S_{h+1},\ldots,R_H) + \sum_{u=h}^{H} R_u = \mathbb{E}_{\pi^0,s,a,P} D_{\hat{G}\bar{\theta}}(S_P,\ldots,R_H) + \sum_{u=h}^{H} R_u$$

$$= \mathbb{E}_{\pi^0,s,a,P} \sum_{u=h}^{P-1} R_u + \mathbb{1}\{P < H+1\} C_{\hat{G}\bar{\theta}}(S_P)$$

where we use Eq. (9). Combining with Eq. (54),

$$\left\langle \varphi(s,a), \bar{\theta}_h \right\rangle \approx_{H\eta_0} \mathbb{E}_{\pi^0,s,a,P} \sum_{u=h}^{P-1} R_u + \mathbb{1}\{P < H+1\} C_{\hat{G}\bar{\theta}}(S_P)$$

$$= \mathbb{E}_{\pi^0,s,a,P} \sum_{u=h}^{P-1} R_u + \mathbb{1}\{P < H+1\} \mathrm{clip}_{[0,H]} \left\langle \varphi(S_P, \pi_{\hat{G}\bar{\theta}}^+(S_P)), \bar{\theta}_P \right\rangle$$

$$\approx_{(H-h)H\eta_0} \mathbb{E}_{\pi^0,s,a,P} \sum_{u=h}^{P-1} R_u + \mathbb{1}\{P < H+1\} q^{\pi}(S_P, \pi_{\hat{G}\bar{\theta}}^+(S_P)),$$

where we used the inductive assumption along with the fact that action-values are bounded in $[0,H]$. Observe also that

$$q^{\pi}(s,a) = \mathbb{E}_{\pi^0,s,a,P} \sum_{u=h}^{P-1} R_u + \mathbb{1}\{P < H+1\} q^{\pi}(S_P, \pi_{\hat{G}\bar{\theta}}^+(S_P)),$$

and therefore

$$\left\langle \varphi(s,a), \bar{\theta}_h \right\rangle \approx_{(H-h+1)H\eta_0} q^{\pi}(s,a),$$

proving Eq. (57) of the inductive assumption for $t = h$.

To show Eq. (56) for $t = h$, by Eq. (57) for $t = h$ and the inductive assumption for $t > h$,

$$\left\langle \varphi(s,a), \bar{\theta}_h \right\rangle \approx_{H^2\eta_0} q^{\pi}(s,a) \geq q^{\star}(s,a) - (H-h)(\varepsilon/H + 2H^2\eta_0).$$

Either $\pi$ chooses the action $a'$ maximizing the inner product above, for which

$$q^{\pi}(s,a') \geq \max_{a\in[\mathcal{A}]} q^{\star}(s,a) - (H-h)(\varepsilon/H + 2H^2\eta_0) - 2H^2\eta_0 \geq v^{\star}(s) - (H-h+1)(\varepsilon/H + 2H^2\eta_0),$$

or it chooses action 1. This can only happen with non-zero probability if $\tau_{\hat{G}\bar{\theta}}(s) < 1$, in which case we have by definition that $\mathrm{range}_Q^{\hat{G}}(s) = \mathrm{range}_Q(s) \leq \frac{\varepsilon}{\sqrt{2dH}}$. Combining with Eq. (3) and Proposition 4.5, $\mathrm{range}(s) \leq \frac{\varepsilon}{H}$, and therefore, using Eq. (56) for $t = h+1$, in this case

$$q^{\pi}(s,1) \geq q^{\star}(s,1) - (H-h)(\varepsilon/H + 2H^2\eta_0)$$

$$\geq v^{\star}(s) - \frac{\varepsilon}{H} - 2\eta - (H-h)(\varepsilon/H + 2H^2\eta_0) \geq v^{\star}(s) - (H-h+1)(\varepsilon/H + 2H^2\eta_0).$$

Therefore for any choice of action $a'$ of policy $\pi$ in state $s$, $q^{\pi}(s,a') \geq v^{\star}(s) - (H-h+1)(\varepsilon/H + 2H^2\eta_0)$. Therefore

$$v^{\pi}(s) \geq v^{\star}(s) - (H-h+1)(\varepsilon/H + 2H^2\eta_0),$$

finishing the induction.

We thus conclude that

$$v^{\pi}(s_1) \geq v^{\star}(s_1) - \varepsilon - 2H^3\eta_0.$$

Combined with Eq. (57) of the inductive assumption, the value of Optimization Problem 4.10 can be bounded as

$$C_{\hat{G}\bar{\theta}}(s_1) = \mathrm{clip}_{[0,H]} \left\langle \varphi(s_1, \pi(s_1)), \bar{\theta}_1 \right\rangle \geq H^2\eta_0 + v^{\pi}(s_1) \geq v^{\star}(s_1) - 2\varepsilon,$$

by assumption on $\eta$ being relatively small (Eq. (21)). $\qquad\square$

## L  Deferred lemmas

**Lemma L.1** (Elliptical potential, Lemma 19.4 from Lattimore and Szepesvári [2020])**.** *Let $V_0 \in \mathbb{R}^{d \times d}$ be positive definite and $a_1 \ldots, a_n \in \mathbb{R}^d$ be a sequence of vectors with $\|a_t\|_2 \leq L < \infty$ for all $t \in [n]$, $V_t = V_0 + \sum_{s \leq t} a_s a_s^\top$. Then,*

$$\sum_{t=1}^n \min\left\{1, \|a_t\|_{V_{t-1}^{-1}}^2\right\} \leq 2 \log\left(\frac{\det V_n}{\det V_0}\right) \leq 2d \log\left(\frac{\operatorname{Tr} V_0 + nL^2}{d \det(V_0)^{1/d}}\right) .$$

**Lemma L.2.** *Let $V \in \mathbb{R}^{d \times d}$ be a symmetric positive definite matrix and $(a_i)_{i \in [n]}$ be a sequence of $n$ $d$-dimensional real vectors. Let $V_i = V + \sum_{j \in [i]} a_j a_j^\top$. Then,*

$$\sum_{i \in [n]} \|a_i\|_{V_i^{-1}}^2 \geq \min\left\{1, \frac{1}{2} \sum_{i \in [n]} \|a_i\|_{V^{-1}}^2\right\}$$

*Proof.* If $\sum_{i \in [n]} a_i a_i^\top \preceq V$, then $V_i \preceq 2V$, and therefore

$$\sum_{i \in [n]} \|a_i\|_{V_i^{-1}}^2 \geq \sum_{i \in [n]} \|a_i\|_{2V^{-1}}^2 = \frac{1}{2} \|a_i\|_{V^{-1}} .$$

Otherwise, $\sum_{i \in [n]} a_i a_i^\top V^{-1}$ has an eigenvalue that is at least 1. As all the other eigenvalues are non-negative (as $V$ is symmetric positive definite), we have that

$$\sum_{i \in [n]} \|a_i\|_{V^{-1}}^2 = \operatorname{Tr}\left(\sum_{i \in [n]} a_i a_i^\top V^{-1}\right) \geq 1 . \qquad \square$$

## M  Estimation error blow-up guarantees

We borrow Assumption M.1 and Theorem M.2 from Lattimore and Szepesvári [2020] and refer the reader to the book for the corresponding proof.

**Assumption M.1** (Prerequisites for Theorem M.2)**.** *Let $\lambda > 0$. For $k \in \mathbb{N}^+$, let $A_k$ be random variables taking values in $\mathbb{R}^d$. For some $\theta_\star \in \mathbb{R}^d$, let $X_k = \langle A_k, \theta_\star \rangle + \eta_k$ for all $k \in \mathbb{N}^+$. Here, $\eta_k$ is a conditionally 1-subgaussian random variable ("noise"), ie. it satisfies:*

$$\text{for all } \alpha \in \mathbb{R} \text{ and } t \geq 1, \qquad \mathbb{E}[\exp(\alpha \eta_k) \mid \mathcal{F}_{k-1}] \leq \exp\left(\frac{\alpha^2}{2}\right) \quad a.s.,$$

*where $\mathcal{F}_{k-1}$ is such that $A_1, X_1, \ldots, A_{k-1}, X_{k-1}, A_k$ are $\mathcal{F}_{k-1}$-measurable.*

**Theorem M.2** (Lattimore and Szepesvári [2020], Theorem 20.5)**.** *Let $\zeta \in (0, 1)$. Under Assumption M.1, with probability at least $1 - \zeta$, it holds that for all $k \in \mathbb{N}$,*

$$\left\|\hat{\theta}_k - \theta_\star\right\|_{V_k(\lambda)} < \sqrt{\lambda} \|\theta_\star\|_2 + \sqrt{2 \log\left(\frac{1}{\zeta}\right) + \log\left(\frac{\det V_k(\lambda)}{\lambda^d}\right)},$$

*where for $k \in \mathbb{N}$,*

$$V_k(\lambda) = \lambda I + \sum_{s=1}^k A_s A_s^\top$$

$$\hat{\theta}_k = V_k(\lambda)^{-1} \sum_{s=1}^k X_s A_s$$

We generalize this theorem to handle non-zero-mean noise with parametrized subgaussianity. To handle non-zero-mean noise, we use [Zanette et al., 2020, Lemma 8]. We state the lemma here and refer the reader to Zanette et al. [2020] for the proof:

**Lemma M.3** (Zanette et al. [2020], Lemma 8). *For $n \in N^+$, let $\{A_i\}_{i=1,\ldots,n}$ be any sequence of vectors in $\mathbb{R}^d$ and $\{\Delta_i\}_{i=1,\ldots,n}$ be any sequence of scalars such that $|\Delta_i| \le \xi \in \mathbb{R}$ with $\xi \ge 0$. For any $\lambda \ge 0$ and $V(\lambda) = \sum_{i=1}^n A_i A_i^\top + \lambda I$ we have:*

$$\left\| \sum_{i=1}^n A_i \Delta_i \right\|_{V(\lambda)^{-1}}^2 \le n\xi^2$$

**Lemma M.4.** *Let $\zeta \in (0,1)$, $\lambda > 0$, $\sigma > 0$, and $\xi \ge 0$. For $k \in \mathbb{N}^+$, let $A_k$ be random variables taking values in $\mathbb{R}^d$. For some $\theta_\star \in \mathbb{R}^d$, let $\tilde{X}_k = \langle A_k, \theta_\star \rangle + \eta_k$ for all $k \in \mathbb{N}^+$. Here, $\eta_k$ is a conditionally $\sigma$-subgaussian random variable, ie. it satisfies:*

$$\text{for all } \alpha \in \mathbb{R} \text{ and } t \ge 1, \qquad \mathbb{E}[\exp(\alpha \eta_k) \,|\, \mathcal{F}_{k-1}] \le \exp\left( \frac{\alpha^2 \sigma^2}{2} \right) \quad a.s.,$$

*where $\mathcal{F}_{k-1}$ is such that $A_1, \tilde{X}_1, \ldots, A_{k-1}, \tilde{X}_{k-1}, A_k$ are $\mathcal{F}_{k-1}$-measurable. With probability at least $1 - \zeta$, it holds that for any sequence $\{\Delta_i\}_{i=1,\ldots}$ such that $|\Delta_i| \le \xi$, for all $k \in \mathbb{N}$,*

$$\left\| \hat{\theta}_k - \theta_\star \right\|_{V_k(\lambda)} < \sqrt{\lambda} \, \|\theta_\star\|_2 + \xi \sqrt{k} + \sigma \sqrt{2 \log\left( \frac{1}{\zeta} \right) + \log\left( \frac{\det V_k(\lambda)}{\lambda^d} \right)}.$$

*where for $k \in \mathbb{N}$,*

$$X_k = \tilde{X}_k + \Delta_k$$

$$V_k(\lambda) = \lambda I + \sum_{s=1}^k A_s A_s^\top$$

$$\hat{\theta}_k = V_k(\lambda)^{-1} \sum_{s=1}^k X_s A_s$$

*Proof.* Let $X'_k = (X_k - \Delta_k)/\sigma_k$, $A'_k = A_k/\sigma_k$, $\lambda' = \lambda/\sigma_k^2$, and $\theta'_\star = \theta_\star$, $V'_k(\lambda') = \lambda' I + \sum_{s=1}^k A'_s A_s'^\top$, and $\hat{\theta}'_k = V'_k(\lambda')^{-1} \sum_{s=1}^k X'_s A'_s$. By assumption, $X'_k, A'_k, \lambda'$ and $\theta'_\star$ then satisfy Assumption M.1. Therefore by applying Theorem M.2, with probability at least $1 - \zeta$, it holds that for all $k \in \mathbb{N}$,

$$\left\| \hat{\theta}'_k - \theta_\star \right\|_{V'_k(\lambda')} < \sqrt{\lambda'} \, \|\theta_\star\|_2 + \sqrt{2 \log\left( \frac{1}{\zeta} \right) + \log\left( \frac{\det V'_k(\lambda')}{\lambda'^d} \right)}.$$

Under this high-probability event, since $V'_k(\lambda') = V_k(\lambda)/\sigma^2$, substituting into the previous display yields

$$\left\| \hat{\theta}'_k - \theta_\star \right\|_{V_k(\lambda)} < \sqrt{\lambda} \, \|\theta_\star\|_2 + \sigma \sqrt{2 \log\left( \frac{1}{\zeta} \right) + \log\left( \frac{\det V_k(\lambda)}{\lambda^d} \right)}. \tag{58}$$

Take any sequence $\{\Delta_i\}_{i=1,\ldots}$ such that $|\Delta_i| \le \xi$ and apply the triangle inequality:

$$\left\| \hat{\theta}_k - \theta_\star \right\|_{V_k(\lambda)} \le \left\| \hat{\theta}'_k - \theta_\star \right\|_{V_k(\lambda)} + \left\| \hat{\theta}'_k - \hat{\theta}_k \right\|_{V_k(\lambda)}, \tag{59}$$

so it remains to bound $\left\| \hat{\theta}'_k - \hat{\theta}_k \right\|_{V_k(\lambda)}$.

$$
\begin{aligned}
\left\| \hat{\theta}'_k - \hat{\theta}_k \right\|_{V_k(\lambda)} &= \left\| V'_k(\lambda')^{-1} \sum_{s=1}^k X'_s A'_s - V_k(\lambda)^{-1} \sum_{s=1}^k X_s A_s \right\|_{V_k(\lambda)} \\
&= \left\| V_k(\lambda)^{-1} \sum_{s=1}^k (X_s - \Delta_s) A_s - V_k(\lambda)^{-1} \sum_{s=1}^k X_s A_s \right\|_{V_k(\lambda)} \\
&= \left\| V_k(\lambda)^{-1} \sum_{s=1}^k \Delta_s A_s \right\|_{V_k(\lambda)} = \left\| \sum_{s=1}^k \Delta_s A_s \right\|_{V_k(\lambda)^{-1}} \\
&\le \sqrt{k} \xi,
\end{aligned}
\tag{60}
$$

where the final inequality uses Lemma M.3. The proof is finished by plugging in the bounds of Eqs. (58) and (60) into the triangle inequality of Eq. (59). □

