# OpenReview forum: "Online RL in Linearly $q^\pi$-Realizable MDPs Is as Easy as in Linear MDPs If You Learn What to Ignore"
_NeurIPS.cc/2023/Conference — NeurIPS 2023 oral_

### Official Review · Reviewer_PFYK · 2023-06-22

**Soundness:** 4 excellent
**Presentation:** 3 good
**Contribution:** 4 excellent
**Rating:** 8
**Confidence:** 3

**Summary:**

This submission provides the first algorithm learning an $\epsilon$-optimal policy with poly many samples obtained through online interactions with a $q^\pi$-realizable MDP. This is an important contribution to an open question in the community. Indeed previous works either adopted a generative model or they required more stringent condition on the MDP class such as Linear MDP.
The core observations that might be useful for futures work are that:
1) $q^\pi$-realizability + strictly positive lower bound on the range $\implies$ Linear MDP.
2) In a $q^\pi$-realizable MDP not only $q^\pi$ are in the features span but also all admissible (see definition 4.6) functions.

Unfortunately, the proposed algorithm is computationally inefficient as the authors mention at line 35 in the introduction.
I think that this limitation should have been better discussed in the paper and mentioned in the abstract as well.

**Strengths:**

Learning an $\epsilon$-optimal policy with poly many samples obtained through online interactions with a $q^\pi$-realizable MDP was an open question before this paper (to the best of my knowledge). This paper solves it with an elegant approach unveiling the fact that $q^\pi$-realizable MDP are not linear MDP due to the existence of some low range states.
However this states are unimportant for the goal of finding an optimal policy and therefore in these states we can fix an arbitrary policy and reduce a $q^\pi$-realizable MDP to a Linear MDP with features norm bound inversely proportional to the range lower bound.
The authors further develop their algorithm to bypass the requirement of knowing the low range states in advance.
I think this reduction to Linear MDP will be important for further advancements in the field.

**Weaknesses:**

I think the authors should have discussed better the fact that their algorithm is not computationally efficient.
In particular, I believe that the authors should explain that the computational barrier comes from Optimization Problem 4.10. This is probably obvious for reader already familiar with ELEANOR but I would still suggest to include this discussion in a Limitations section that the authors may want to add in their final revision.

Some steps in the presentation might also be improved (please see questions in the next Section)

**Questions:**

1) Which is the output of Algorithm 2 ? In Algorithm 1, at line 7 the policy $\pi^{mk}$ is defined as output of Skippy Policy which is not specified. Is it the policy that at state $S_i$ sample an action from $\pi^+(S_i)$ with probability $\tau_i$ and plays action 1 otherwise ?

2) Is the quantity $E^{\rightarrow}(s_i \rightarrow)$ at line 275 be the same as the quantity $E_{G\theta}^{\rightarrow}(S_{t+1}^{lkj}, \dots R_H^{lkj})$ used in the least squares target ?

3) I couldn't follow your reasoning to show that $E_{G\theta}^{\rightarrow}(S_{t+1}^{lkj}, \dots R_H^{lkj})$
  is linearly realizable. In particular, Corollary 4.11 shows that is realizable for the true guess. However the solution to the Optimization Problem 4.10 is different from the true guess so in this case $E_{G\theta}^{\rightarrow}(S_{t+1}^{lkj}, \dots R_H^{lkj})$ can not be guaranteed to be realizable. Could you please elaborate more on this step in the rebuttal ?

4) Given the reduction from $q^\pi$-realizable MDPs to Linear MDPs that you discovered in this paper. What would prevent one from applying efficient algorithms for linear MDPs such as LSVI-UCB to solve $q^\pi$-realizable  MDPs ? Why did you choose to adopt the computationally inefficient Eleanor as base algorithm for solving Linear MDPs while computationally efficient alternatives exist ?

**Limitations:**

The limitations are not well discussed. In particular, I would suggest to add a discussion concerning the computational cost of the algorithm.

**Minor**: typo at line 302 "th matrix" -> "the matrix"

---

> ### Author Rebuttal · Authors · 2023-08-09
>
> We thank the reviewer for their helpful comments. We answer their questions below:
>
> Computational complexity and Q4: Regrettably, we eliminated our discussion on computational complexity when shortening the initial submission -- thank you for pointing this out. We will include it in the final submission, as summarized below (also mentioning the inefficiency of the algorithm in the abstract).
> In our opinion significant challenges remain to determine whether a computationally efficient solution exists, and this remains an interesting open problem.
> Our method (Optimization Problem 4.10) includes a guess of G, which impacts the optimization quantities in a non-linear way. Computationally efficient approximations tend to lead to the approximation error growing exponentially in H. This is the main reason we could not show the computational efficiency of our method. As such, we opted to use an ELEANOR-style algorithm instead of one of LSVI-UCB-style, as the LSVI-UCB-style method would not present any advantages but (with our proof) would in fact result in worse sample complexity and would also make the presentation (even) more complex.
>
> Q1. SkippyPolicy is a policy in the MDP (with no return value). In line 7 of Algorithm 1, $\pi^{mk}$ is the policy defined by SkippyPolicy$(\hat{G},\bar\theta,k)$. We realized this was unclear and changed line 7 in the revised version to “Let $\pi^{mk}$ be the policy defined by $\text{SkippyPolicy}(\hat{G},\bar\theta,k)$”.
>
> Q2. Yes (in the case that $i=t+1$). We drop the subscripts of $E$ when not needed (line 265) and introduce the $s_i \rightarrow$ notation in line 268 to clean the notational clutter. We refer to states in a fixed trajectory with lower-case $s_i$ for the explanation in line 275 (and elsewhere), and denote random variables by upper-case in the least-squares target (and elsewhere). Please let us know if this is not presented well and we will improve it.
>
> Q3. You are right that the realizability of $E_{G\theta}^{\rightarrow}(S_{t+1}^{lkj}, \dots R_H^{lkj})$ does not follow from Corollary 4.11 (and is not even true in general). In fact, we can only prove the realizability of this for the true guess (a fact we use to establish the optimism property of Optimization Problem 4.10). Once the optimism property is established, similarly to ELEANOR’s proof, we only need to learn (and make progress) whenever there is a discrepancy (i.e., a difference in the predicted and measured value for the optimistic policy). For ELEANOR, learning is always implemented in the form of an elliptical potential growth of the covariate-matrix, because the underlying least-squares targets are realizable.
> For us, as we do not have realizability of $E_{G\theta}^{\rightarrow}(S_{t+1}^{lkj}, \dots R_H^{lkj})$, the discrepancy can be either (as with ELEANOR) due to the elliptical potential growth of the covariate-matrix, but it could also be the result of our least-squares targets not being realizable.
> Correspondingly, we need another component of the proof/algorithm to learn from these cases (Section 4.4, Checking consistency). This is why we introduce matrix-form versions of $E_{G\theta}^{\rightarrow}(S_{t+1}^{lkj}, \dots R_H^{lkj})$, denoted by $F$. The guarantee of Corollary 4.11 can be used to show that projections of the matrix-form of the discrepancy (see Optimization Problem 4.12) align well with the $\bot(Q, i)$ subspace. Consequently, we learn (and make progress) from the discrepancies that do not grow the elliptical potential of the covariate-matrix, as these stem from the non-realizability of $E_{G\theta}^{\rightarrow}(S_{t+1}^{lkj}, \dots R_H^{lkj})$, an error we can learn from (Line 12 of Algorithm 1). For more information, please see the [“global” response](https://openreview.net/forum?id=HV85SiyrsV&noteId=26AjXdraOA), which includes an explanation of related ideas that we added to the paper.

---

> > ### Comment · Reviewer_PFYK · 2023-08-13
> > **Thanks**
> >
> > Dear authors,
> >
> > Thanks for your response !
> >
> > From your previous answer, I am still not sure to understand the problem that would have been encountered with a LSVI-UCB type of algorithm.
> >
> > Would it lead to an exponential sample complexity due to the approximation error exponential in the horizon that you mentioned ?
> >
> > Or it would lead to a polynomial sample complexity but again with a computationally inefficient algorithm ?

---

> > > ### Author Response · Authors · 2023-08-14
> > >
> > > We think LSVI-UCB may fail altogether, at least we are unable to prove that it produces a good policy, since the linear MDP assumption is not met and the argument that shows that LSVI-UCB produces a good policy appears to rely on this assumption in a crucial way.
> > > We currently do not have a counterexample that shows that LSVI-UCB will indeed fail. However, we believe the following argument could be used to create a counterexample: one could start from a linear MDP and add states with zero range (motivated by the theory in this paper), and extend the features of such states with arbitrary “nuisance” features. Then one would show that the “nuisance” features can be chosen in a way so that the optimistic targets of LSVI-UCB are not realizable, leading to an estimator that is totally misguided, leading to no learning.
> > >
> > > We originally interpreted the question as whether LSVI-UCB could be modified to work for $q^\pi$-realizable MDPs (the way our paper modified ELEANOR). In summary, we were not able to computationally efficiently do this in a way that leads to a polynomial sample complexity. We expand on this further below.
> > >
> > > We need to introduce a mechanism that iteratively refines the target functions so that they are eventually realizable under the $q^\pi$-realizability assumption. For our method presented in the paper, this mechanism is optimistically guessing (with $\hat G$) and refining the range-estimates (via Optimization Problem 4.12). Without such a mechanism, the LSVI-UCB (or indeed even vanilla ELEANOR) targets may be unrealizable, invalidating a crucial prerequisite of their least-squares analyses. Our attempts to add such a mechanism to an LSVI-UCB style algorithm resulted in either (i) having a global optimization routine over $\hat G$, like in the current version of the paper, or (ii) introducing a computationally efficient approximator, but the approximation error of this was too large and scaled exponentially in $H$. (ii) and (i) lead to the two undesirable outcomes you mentioned, respectively.

---

> > > > ### Comment · Reviewer_PFYK · 2023-08-14
> > > >
> > > > Thanks for this additional clarification.
> > > >
> > > > I think that this discussion would fit the paper well because it is a nice discussion on a natural follow up research question which is designing a computationally efficient algorithm for the same setting.
> > > >
> > > > Knowing which are the problems in adapting LSVI-UCB is certainly interesting.
> > > >
> > > > My assessment of the paper improved after discussion therefore I would like to increase my score to 8.

---

### Official Review · Reviewer_Wpjh · 2023-07-03

**Soundness:** 3 good
**Presentation:** 2 fair
**Contribution:** 4 excellent
**Rating:** 8
**Confidence:** 4

**Summary:**

This paper proves that the standard online episodic RL with the $q^\pi$-realizability assumption can be learned with polynomial samples, without generative models. Technically, this paper proves that the states of any MDPs with $q^\pi$-realizability assumption can be partitioned into two disjoint sets, where the states in the first set have small range (i.e., the advantage function on the state is always small for every policy), and the transitions on the states in the second set can be approximated by a linear MDP. Consequently, this paper designs a novel algorithm called SKIPPYELEANOR that learns to distinguish the states with small range and the linear MDPs simultaneously.

**Strengths:**

This paper solves a long-standing open question of reinforcement learning theory by showing that MDPs with $q^\pi$-realizability assumption can be learned with polynomial samples. The result is significant and beneficial to the community. The observation that the difference between MDPs with $q^\pi$-realizability and linear MDPs is only the set of states with very small range is novel and neat. Conceptually, this observation deepens our understand on the $q^\pi$-realizability assumption and can potentially inspire a new line of research.

**Weaknesses:**

The exposition of the results could be improved. Current Section 4 of this paper is very technical and involves too many symbols and definitions, which makes it less readable, despite the fact that the algorithm is conceptually simple. Is there a simpler setting that demonstrate the core idea of the algorithm while keeps the technical part less involved?

**Questions:**

In the last paragraph in Section 3, this paper argues that a MDP with $q^\pi$-realizability can be converted to linear MDP by skipping the states with low range. Is this a rigorous reduction? How to construct the new MDP? In particular, the states are staged in the original MDP. By skipping some states, is the new MDP also staged, and is the stage of the un-skipped states unchanged?

[Minor] In the caption of Figure 1, should the feature of all other states be 0 instead of 0.5?

[Minor] What’s $S_h$ in the displayed equation after line 222?


**Limitations:**

The authors adequately addressed the limitations and potential negative societal impact.

---

> ### Author Rebuttal · Authors · 2023-08-09
>
> We thank the reviewers for their helpful comments. We address their concern mentioned in “weaknesses” in the [“global” response](https://openreview.net/forum?id=HV85SiyrsV&noteId=26AjXdraOA) and answer their questions below:
>
> Q1. We do not show a formal reduction of $q^\pi$-realizable MDPs to linear MDPs in this work. Though such a reduction is part of the intuition of our method, it is not strictly part of the proof sadly. The reason for this is that the skipping oracle mentioned is hard to learn, and instead of a direct approach we argue that learning about this oracle happens whenever there is a need (performance shortfall) for it.
> A formal reduction, while tangential to our proof, is fairly straight-forward but cumbersome, with the caveat that the linear MDP will end up with $dH$ (instead of $d$) dimensional features to account for the technicality about the stage mapping that you mention.
> One would proceed by copying the features of each state $s$ in stage $h$ into the $h^{th}$ chunk of size $d$ of this vector of size $dH$ (the rest of the vector remains zero). A similar transformation is applied to all $\theta_h(\pi)$. Then, $H$ copies are made of each high-enough-range state, with all possible stages (but keeping the feature vectors). These will be the states of the new MDP we construct. When a transition from state $s$ leads to skipped states, the linear MDP returns with the copy of the first non-skipped state that has a stage counter of stage($s$)+1, so that in this linear MDP the stage numbers are consecutive (as required by our definitions). $q^\pi$-realizability of this modified MDP is easy to show, and – as it has no low-range states – Proposition 3.4, can be used to show that the modified MDP is linear. To account for the fact that this new MDP may finish an episode in fewer than H steps due to the skips, we add a special, zero-reward, self-transitioning state called “episode-over”. To ensure that the MDP stays linear, we extend the feature vectors of each state by a scalar 1, and a scalar indicator of being in this state, with all original features of the “episode-over” state defined to be zero. It is easy to see that this construction leads to a linear MDP with the desired action-value functions.
>
> Q2. Yes, thank you for pointing out this typo.
>
> Q3. $S_h$ is the (random) $h^{th}$-stage state when an MDP trajectory is started from state-action $(s, a)$, and policy $\pi$ is followed. Note the preceding expectation operator; this notation is introduced in Line 76.

---

> > ### Comment · Reviewer_Wpjh · 2023-08-14
> > **Thank you for the comments**
> >
> > Thank you for the response. After reading the responses, my rating remains the same, and I would recommend the authors include the reduction (even informally) in the paper upon revision.

---

> > > ### Author Response · Authors · 2023-08-14
> > >
> > > Thank you, we will include the reduction in the appendix.

---

### Official Review · Reviewer_ap44 · 2023-07-07

**Soundness:** 3 good
**Presentation:** 1 poor
**Contribution:** 4 excellent
**Rating:** 4
**Confidence:** 3

**Summary:**

This paper studies online RL under linear $q^\pi$ realizable MDPs, and proposes an algorithm that learns an epsilon-optimal policy in polynomial time and polynomial sample complexity.  Previous online RL in linear RL either assumes that transitions and rewards are linear individually (e.g., linear MDP), or assumes ergodicity (e.g., Politex), or assumes access to an simulator that can reset to a previously visited states (Confidence-MC-Politex). This work removes all these assumptions and show that as long as $q^\pi$ is realizable for all $\pi$, polynomial-time algorithm is possible. The novel and surprising observation is that linear $q^\pi$ MDP is actually a linear MDP if the "range" of all states are large. The proposed algorithm is inspired by ELEANOR but "skip" low-ranged state in its execution, and at the same time refine the estimation for the range on the fly using some techniques related to optimal design.

**Strengths:**

The observations and techniques in this paper are novel and creative. It answers an open question by Du et al. (2019) on whether sample-efficient RL under linear $q^\pi$ MDP is possible.

[Du et al., 2019] Simon Du, Sham Kakade, Ruosong Wang, Lin F. Yang. Is a Good Representation Sufficient for Sample Efficient Reinforcement Learning?

**Weaknesses:**

While I appreciate the technical contribution, the delivery of the result is far from ideal. The paper is very difficult to read. I have identified the following possible reasons:
1. Highly cluttered notation.  I think a rule of thumb in writing math is to avoid super/subscript of super/subscript as best as we can, and keeping the super/subscript as simple as possible, because they can make the expression difficult to parse. However, this kind of notations is everywhere in this paper, like $\hat{\theta}\_{\hat{G}\bar{\theta}}^{p^{mkj}(k),i}$ or
$||\varphi_{p(k)}^{mkj}||\_{X^{-1}_{m,p^{mkj}(k)}}$ where not only the super/subscript dependence is deep, each layer has many letters. It becomes a significant obstacle when tracing the calculations. Besides, for the notation $p^{mkj}(k)$, I did not find a place where the parameter in the parenthesis does not match the second parameter in the superscript, so maybe one of them can be omitted?
2. The "definition" dependence is deep. This means that the definition of a quantity usually involves another quantity defined somewhere else; and when I visit the "somewhere else", it further requires me to understand other quantities defined in other places, and so on. For example, Optimization Problem 4.12 involves $y^{ki}\_{\hat{G}\bar{\theta}}$ and
$\hat{F}^{ki}\_{\hat{G}\bar{\theta}}$ for which I have to check the complicated Eq. (12) for their definitions. The $c^j_{ki}$
there is further defined by a big quantity in (18) and the definitions of $\varphi^{mkj}\_{p(k)}$ and $\hat{\theta}\_{\hat{G}\bar{\theta}}^{p^{mkj}(k), i}$ are further scattered in the previous page which further requires understanding other quantities first. I think part the difficulty to trace is also because of Reason 1 --- the readers are facing highly cluttered notations in each step of tracing the definitions, and eventually cannot really grasp the overall meaning.
3. No reminders for the place of the definitions.  Since there are a big amount of cluttered notations, the readers easily forget where they are defined, so I think it's better that there could be reminders here and there to let the reader know where the notations are defined previously.  For example, when I first see $C_{\hat{G}\bar{\theta}}$ in Line 279, I don't know where this is defined. After a long time, I realized that it is a result of the statement in Line 256-258 with the $C$ defined in (8).
4. Confusing notation/terminology: In Line 188, $Z(Q,h)$ is defined as some subspace, but $Proj_{Z(Q,h)}$ is defined as projection onto its *orthogonal* space, which is counterintuitive notational-wise. As another example, in Line 266 or Line 4 of Algorithm 2, $\tau$ is called a "skipping probability", but to my understanding, $\tau$ is more like "probability of *not* skipping".
5. Lack of high-level explanation. Though I can see that the authors have used a considerable space for describing the components of the algorithm, there are still many design choices left unexplained. For example, I don't quite understand why an additional index of k in Line 6 of Algorithm 1 is needed, and why we need to call Algorithm 2 with that input k. I also hope to see a more high-level interpretation of the $\mathbf{f}$ defined in Line 240, or what does it implied if $v^\top_{\||(Q,h)} \mathbf{f}w$ is $\alpha$-admissible (Lemma 4.9 and Corollary 4.11). For example, what's the importance of this fact for the checking consistency procedure. The design of the regression target $E^{\rightarrow}(s_i\rightarrow)$ also deserves more explanation, e.g., what's the relation between this and the standard one in ELEANOR.

I understand that the main text space is very limited. But I believe there is way to give a high-level explanation and self-contained story in the main texts, and leaving details to the appendix. Currently, the main text itself is crowded with difficult-to-trace notation and lacks necessary explanations. This also hinders me from making a careful check for the correctness of the proofs. So I suggest that the authors spend time re-thinking about how to present the results, to really make it a strong paper.

Other notations I couldn't find corresponding definitions:
- $B_i$ in Line 5 of Algorithm 2
- $\otimes$ in (12).

**Questions:**

Can you give some high-level ideas of Optimization Problem 4.12? What are the meanings of $y^{ki}\_{\hat{G}\bar{\theta}}$ and $\hat{F}^{ki}\_{\hat{G}\bar{\theta}}$?

Besides, why do we need the for-loop in Line 6 in Algorithm 1?

**Limitations:**

There's no potential societal impact.

---

> ### Author Rebuttal · Authors · 2023-08-09
>
> We thank the reviewer for bringing these presentational issues to our attention, and a special thanks for taking the time to provide useful concrete suggestions for improvement! The proof of our result uses many novel ideas and techniques, and we strived hard to present it well despite its inherent complexity (e.g., by introducing our definitions in a conceptually sensible order, accompanied by high-level descriptions). It was very useful to read where this presentation fell short, and we have correspondingly carried out the following improvements (which will appear in the revised/final version):
>
> - All instances of super/subscript of super/subscript were eliminated with a notational shorthand to remove the clutter (most notably, $p^{...}(.)$ becomes $p(...)$).
> - Clarifications and reminders were added to aid understanding, addressing (among others) all concrete confusion points noted in the review.
> - Added a high-level explanation for the decomposition of $E^\to$ first into $E$, before decomposing $E$ into matrix-valued $F$ functions that will be used for Optimization Problem 4.12: “Since our realizability results in Section 4.2 only apply to functions defined at a given stage (as only memoryless policies are $q^\pi$ -realizable), to be able to show that the least-squares targets are linearly realizable, we first decompose $E^\to$ to directly express the effect of each stage in the trajectory (backwards).”
> - See [“global” response](https://openreview.net/forum?id=HV85SiyrsV&noteId=26AjXdraOA) that adds high-level explanation to the paper and addresses the reviewer’s point 5 and both questions.
>
> minor comments:
> - To point 4 of the weaknesses, the reviewer possibly misread this as $Proj_{Z(Q,h)}$ is defined as the (orthogonal) projection onto $Z(Q,h)$, not its orthogonal space (Definition 4.2). We fixed the confusing “skipping probability” terminology for $\tau$.
> - $B_i$ is a Bernoulli random variable and is defined in Appendix B (in the revised version we added this to Algorithm 2), and $\otimes$, defined in Appendix A, denotes a tensor product. We added the necessary pointers to make this clear.

---

> > ### Comment · Reviewer_ap44 · 2023-08-16
> >
> > I thank the authors for addressing many of my questions, and proposing plans of improvement. I reiterate that I think the contribution is significant, and the techniques are novel and inspiring. All my complaints are about the readability. Still, readability is an important part of a paper, and it is hard to see the overall picture of the paper after the authors implement the proposed modification. Therefore, I'm leaving my score as it is, with the hope that the paper can be sufficiently revised and reviewed again.

---

### Official Review · Reviewer_dNHS · 2023-07-09

**Soundness:** 3 good
**Presentation:** 3 good
**Contribution:** 3 good
**Rating:** 5
**Confidence:** 3

**Summary:**

The paper addresses online RL in episodic MDPs under linear $q^\pi$-realizability, where policy action values are expressed as linear functions of state-action features. This class is shown to be more general than linear MDPs, where only the transition kernel and reward function are linear functions of feature vectors. The key contribution is identifying states in linearly $q^\pi$-realizable MDPs where actions have approximately equal values, enabling transformation into a linear MDP. A novel learning algorithm is derived to determine which states to skip and apply another learning algorithm to the linear MDP. The proposed algorithm achieves an $\varepsilon$-optimal policy after $\operatorname{polylog}(H, d) / \varepsilon^2$ interactions, representing the first online RL algorithm for linearly $q^\pi$-realizable MDPs with polynomial sample complexity. Notably, the results hold in the misspecified case, where sample complexity gracefully degrades with the misspecification error.

**Strengths:**

1. This paper offers a comprehensive exploration of Linearly $q^{\pi}$-Realizable MDPs, making a significant contribution to the field of reinforcement learning with feature learning.

2. The paper exhibits a clear and concise mathematical exposition with a well-structured presentation that is easy to follow.

**Weaknesses:**

1. The current form of the paper lacks a Conclusion section, rendering it incomplete.

2. It would enhance clarity if the authors presented a comparative analysis in tabular form, providing a clearer understanding of the different studies involving MDPs with $q^{\pi}$-realizability and linear structure.

3. The paper has relatively few references, and the discussion of this line of work appears somewhat limited. It would be beneficial for the authors to discuss more related work in this area.

4. The left and right sides of Figure 1 seem to be reversed.

**Questions:**

1. What is the lower bound for MDPs with $q^{\pi}$-realizability? It appears to have significant dependencies on both the time H and d.

2. There seems to be an issue with the citation format. The requirement is to use an unnumbered first-level heading for the references.


**Limitations:**

The suggestions have been claimed in "Weaknesses" and "Questions".

---

> ### Author Rebuttal · Authors · 2023-08-09
>
> We thank the reviewer for their helpful comments, based on which we have added a table to the revised version of our paper containing the summary of how our work fits in the wider literature. In short, (i) for linear MDPs, Jin et al. [2020] provide algorithms with both polynomial sample and computational complexity for the online RL setting, and hence also for the planning with a simulator setting; (ii) for the case of linearly $q^\pi$-realizable MDPs, Yin et al. [2022] provide a computationally efficient algorithm with polynomial sample complexity for planning (with simulator access), but no prior work has provided algorithms with polynomial sample complexity for online RL; (iii) we provide a computationally inefficient method for this case with polynomial sample complexity (the first such result in the literature), but whether this is achievable with an efficient algorithm remains an intriguing open problem. We also added a discussion to conclude the paper clearly pointing out directions for future work. Due to space limitations we focused our literature review to the strictly most relevant works that provide a sufficient context to fully understand our new results. We are more than happy to include discussions on any works we might have missed out; please share any suggestions with us in a reply.
>
> Q1. We are not aware of any lower bounds beyond those that apply to linear MDPs. This lower bound is $d^2H^4/\epsilon^2$ (from converting Remark 9 of Zhou et al., 2021, “Nearly Minimax Optimal Reinforcement Learning for Linear Mixture Markov Decision Processes” to our setting, in the latter of which  parameters are not shared across the stages). This bound matches the upper bound of ELEANOR (Zanette et al., 2020) up to logarithmic factors.
> While we therefore have the optimal dependence on $\epsilon$, our exponents of $d$ and $H$ may be far from optimal, though we were unable to improve upon them. That said, our result is the first one establishing that this problem is polynomial-sample-complexity solvable, resolving an open problem of Du et al., [2019]. In our opinion this is the first and most important question to resolve, as it tends to establish a dividing line between tractable and intractable problems: polynomial-complexity bounds often can (and tend to) be improved upon, while problems shown to be exponentially hard tend to remain intractable even despite technological advances.
>
> Q2. Thank you, we will make sure our citation format complies with the formatting requirements.
>
> Re Figure 1, the sides of the figure are correct: The right MDP is obtained from the left one by removing (skipping) the states marked by red (and showing the cumulative rewards for the corresponding transitions). E.g., the two possible 2-step transition in the left MDP from $s_1$ to $s_4$ through $s_3$ with reward 0.5 for each transition is replaced in the right MDP with a single-step transition from $s_1$ to $s_4$ with reward 1 (for action 2, the action leading from $s_1$ to $s_3$ in the left MDP). Note that, as mentioned in the review of Wpjh, there is a typo in the caption and the feature of all other states should be $(0)$ not $(0.5)$.

---

### Author Rebuttal · Authors · 2023-08-09

Please see our individual responses to the reviewers. We use this space to address a comment made by multiple reviewers that the exposition of the results could be improved (thank you for this feedback). We provide below a high-level explanation of our method and proof compared to ELEANOR (such a description is also added the the revised version of the paper).

When setting up least-squares targets for the state-action pair $(S_t, A_t)$, ELEANOR uses $R_t + predicted \textunderscore value \textunderscore for(S_{t+1})$.
With this target we only require one on-policy rollout for each episode in order to get the least squares parameter estimate for all $H$ stages. In contrast, our least-squares targets are of the form of $R_t + … + R_{t+i} + predicted \textunderscore value \textunderscore for( S_{t+i+1})$, where $i$, the number of stages “skipped”, depends on the guess $\hat G$. The guess $\hat G$ is selected only in Optimization Problem 4.10, and we do not know its value at the time of data collection, so we cannot know which stages will have to be skipped. Therefore, (i) we need access to the rewards of the current policy at any stage (similarly to ELEANOR), and hence we run the current policy to any stage (including the last one); and (ii) perform rollouts with the fixed policy $\pi^0$ (from any stage) to be able to estimate the rewards $R_{t+1},...,R_{t+i}$ collected while skipping over $i$ stages (for any $i$). To ensure this happens for every stage, we start phase II from every stage $k$, resulting in the additional for loop in Algorithm 1 compared to ELEANOR. (Finally, the randomization in Phase I is applied to make the optimization problem smooth, as described from line 160.)

One could analyze this algorithm similarly to the analysis of ELEANOR if it were not for the fact that the least-squares targets we just introduced are not realizable in general. We can, however, prove the realizability of certain components of the matrix-valued version of these targets, $F$ (Lemma 4.9 and Corollary 4.11). This enables us to detect when the realizability of our least-squares targets fail, measure the direction (component) of the largest error, and learn from that. This is the job of Optimization Problem 4.12:
$\hat{F}^{ki}_{\hat{G}\bar{\theta}}$ corresponds to the matrix-valued empirical measurements of $F$,

while $y^{ki}_{\hat{G}\bar{\theta}}$
are the average predictions of the same quantities. If realizability were to hold, these matrices would be very close; if not, the direction of their largest discrepancy tells us something about $\bot(Q, i)$, and allows us to learn.

Optimism ties all this together: either there is no shortfall between predicted and measured $q$-values (and we are done) or we grow the elliptical potential of $X$ (the two cases present in ELEANOR), or we grow the elliptical potential of $Q$ (the new case due to the lack of realizability guarantees).

---

### Decision · Program_Chairs · 2023-09-21

**Decision:**

Accept (oral)

**Comment:**

This paper resolves one of the major open problems in RL with function approximation: Linearly $q^\pi$-Realizable MDPs in the online setting. The paper gives structural lemmas by connecting $q^\pi$-Realizable MDPs with a seemingly significant broader class: linear MDP. All reviewers believe this paper makes significant contributions to the community. The AC agrees and recommends acceptance.
Please revise the writing according the reviewers' suggestions in the final version.